# Learning Orthogonal Multi-Index Models: A Fine-Grained Information Exponent Analysis

## Abstract

The information exponent (Ben Arous et al. (2021)) — which is equivalent to the lowest degree in the Hermite expansion of the link function for Gaussian single-index models — has played an important role in predicting the sample complexity of online stochastic gradient descent (SGD) in various learning tasks. In this work, we demonstrate that, for multi-index models, focusing solely on the lowest degree can miss key structural details of the model and result in suboptimal rates.

Specifically, we consider the task of learning target functions of form $f_*(\boldsymbol{x}) = \sum_{k=1}^{P} \phi(\boldsymbol{v}_k^* \cdot \boldsymbol{x})$, where $P \ll d$, the ground-truth directions $\{\boldsymbol{v}_k^*\}_{k=1}^{P}$ are orthonormal, and only the second and $2L$-th Hermite coefficients of the link function $\phi$ can be nonzero. Based on the theory of information exponent, when the lowest degree is $2L$, recovering the directions requires $d^{2L-1} \operatorname{poly}(P)$ samples, and when the lowest degree is 2, only the relevant subspace (not the exact directions) can be recovered due to the rotational invariance of the second-order terms. In contrast, we show that by considering both second- and higher-order terms, we can first learn the relevant space via the second-order terms, and then the exact directions using the higher-order terms, and the overall sample and complexity of online SGD is $d \operatorname{poly}(P)$.

## 1 Introduction

In many learning problems, the target function exhibits or is assumed to exhibit a low-dimensional structure. A classical model of this type is the multi-index model, where the target function depends only on a $P$-dimensional subspace of the ambient space $\mathbb{R}^d$, with $P$ typically much smaller than $d$. When the relevant dimension $P = 1$, the model is known as the single-index model, which dates back to at least Ichimura (1993). Both single- and multi-index models have been widely studied, especially in the context of neural network and stochastic gradient descent (SGD) in recent years, sometimes under the name "feature learning"(Ben Arous et al. (2021); Bietti et al. (2022); Damian et al. (2022); Abbe et al. (2022; 2023); Damian et al. (2024); Oko et al. (2024); Dandi et al. (2024)).

In Ben Arous et al. (2021), the authors show that for single-index models, the behavior of online SGD can be split into two phases: an initial "searching" phase, where most of the samples are used boost the correlation with the relevant (one-dimensional) subspace to a constant, and a subsequent "descending" phase, where the correlation further increases to 1. They introduce the concept of the information exponent (IE), defined as the index of the first nonzero coefficient in the Taylor expansion of the population loss around 0, which corresponds to the lowest degree in the Hermite expansion of the link function in Gaussian single-index models. They prove that the sample complexity of online SGD is $\tilde{O}(d)$ when IE $= 2$ and $\tilde{O}(d^{k-1})$ when IE $= k \geq 3$. After that, various lower and upper bounds have been established for single-index models in Bietti et al. (2022); Damian et al. (2023; 2024). Similar results for certain multi-index models have also been derived in Abbe et al. (2022; 2023); Bietti et al. (2023); Oko et al. (2024). In all cases, the sample complexity of online SGD scales with $d^{\mathrm{IE}-1}$ when IE $\geq 3$.[1]

---

[1]The sample complexity can be significantly improved with non-gradient-based methods (Chen & Meka (2020); Troiani et al. (2024); Barbier et al. (2019)), or if we reuse the batches or preprocess the labels (Arnaboldi et al. (2024); Dandi et al. (2024); Lee et al. (2024); Damian et al. (2024)). The latter leads to the notion of generative exponent (Damian et al. (2024)). However, note that our next example is valid for the generative

For multi-index models of form $f_*(\boldsymbol{x}) = \sum_{k=1}^P \phi_k(\boldsymbol{v}_k^* \cdot \boldsymbol{x})$, another layer of complexity arises. In this setting, there are two types of recovery: recovering each direction $\boldsymbol{v}_k^*$ (strong recovery) and recovering the subspace spanned by $\{\boldsymbol{v}_k^*\}_k$. The former notion is stronger, because once the directions are known, the learning task essentially reduces to learning the one-dimensional $\phi_k$ : $\mathbb{R} \to \mathbb{R}$ for each $k \in [P]$. However, strong recovery is not always possible. To see this, consider the case $\phi_k(z) = h_2(z)$, where $h_L$ is the $L$-th (normalized) Hermite polynomial. One can show that this corresponds to decomposing the projection matrix (a second-order tensor) of the subspace $\mathrm{span}\{\boldsymbol{v}_k^*\}_k$. If the model is isotropic in the relevant subspace, recovering the directions is impossible due to the rotational invariance (see Section 3.1 for more discussion). In contrast, when $\phi_k(z) = h_2(z) + h_4(z)$, the identifiability property of the fourth-order tensor decomposition problem allows strong recovery via tensor power method or (stochastic) gradient descent (Ge et al. (2018); Li et al. (2020); Ge et al. (2021)). Note that in both examples, the information exponent is 2, indicating that information exponent alone does not distinguish between these two scenarios.

This leads to a natural question: Can we combine the above results for orthogonal multi-index models by first using the second-order terms to recover the subspace and then using the higher-order terms to learn the directions? Ideally, the first stage would require at most $\tilde{O}(d\,\mathrm{poly}(P))$ samples, consistent with the case IE $= 2$, and once the subspace is recovered, later steps would also cost at most $d\,\mathrm{poly}(P)$ samples.[2] This would yield an overall $\tilde{O}(d\,\mathrm{poly}(P))$ sample (and also time) complexity for strong recovery of the ground-truth directions. Note that the $d$-denpendence matches the IE $= 2$ case and the strong recovery guarantee aligns with the results for IE $> 2$. In this work, we prove the following theorem, providing a positive answer to this question.

**Theorem 1.1** (Informal version of Theorem 2.1)**.** *Suppose that the target function is* $f_*(\boldsymbol{x}) = \sum_{k=1}^P \phi(\boldsymbol{v}_k^* \cdot \boldsymbol{x})$ *where* $\phi = h_2 + h_{2L}$ ($L \geq 2$) *and* $\{\boldsymbol{v}_k^*\}_{k=1}^P$ *are orthonormal, and the input* $\boldsymbol{x}$ *follows the standard Gaussian distribution* $\mathcal{N}(0, \boldsymbol{I}_d)$*. Then, we can use online SGD (followed by a ridge regression step) to train a two-layer network of width* $\mathrm{poly}(P)$ *to learn (with high probability) this target function using* $\tilde{O}(d\,\mathrm{poly}(P))$ *samples and steps.*

**Remark**. For simplicity, we assume the link function is $\phi = h_2 + h_{2L}$. Our results can be extended to more general even link function, provided their Hermite coefficients decay sufficiently fast. See Section 2 (in particular Lemma 2.1 and Lemma 2.2) for further discussion. ♣

**Organization**   The rest of the paper is organized as follows. First, we review the related works and summarize our contributions. Then, we describe the detailed setting and state the formal version of the main theorem in Section 2. In Section 3, we discuss the easier case where the training algorithm is population gradient flow. Then, in Section 4, we show how to convert the gradient flow analysis to an online SGD one. Finally, we conclude in Section 5. The proofs, simulation results, and a table of contents can be found in the appendix.

## 1.1   RELATED WORK

In this subsection, we discuss works that are directly related to ours or were not covered earlier in the introduction.

Along the line of information exponent, the paper most related to ours is (Oko et al. (2024)). They show that for near orthogonal multi-index models, the sample complexity of recovering all ground-truth directions using online SGD is $\tilde{O}(Pd^{\mathrm{IE}-1})$ when IE $\geq 3$. However, their results do not apply to the case IE $= 2$ for the reason we have discussed earlier. Our result considers the situation where both IE $= 2$ and IE $\geq 3$ terms are present and show that in this case, the sample complexity of online SGD is $\tilde{O}(d\,\mathrm{poly}(P))$.

During the writing of this manuscript, we became aware of the concurrent work (Ben Arous et al. (2024)). Our main results are not directly comparable since the settings are different. They run SGD on the Stiefel manifold which automatically prevents collapse but allow the target model to

---

exponent as well with some slight modifications. In other words, the generative exponent is also not sufficient to capture the richer structure of multi-index models.

[2]The $d$ factor in the second stage comes from the fact that the typical squared norm of the noise is $d$, so we have to choose the step size to be $O(d^{-1})$ for the noise to be reasonably small.

have condition number larger than $1$. In addition, only the lowest degree is considered in their work. However, they also show (in a different setting) that when the second order term is isotropic, the initial randomness can be preserved throughout training. A similar idea is used in our analysis of Stage 1.1 (cf. Section 3.1).

Another related line of research is learning two-layer networks in the teacher-student setting (Zhong et al. (2017); Li & Yuan (2017); Tian (2017); Li et al. (2020); Zhou et al. (2021); Ge et al. (2021)). Among them, the ones most relevant to this work are (Li et al. (2020)) and the follow-up (Ge et al. (2021)), both of which consider orthogonal models similar to ours and use similar ideas in the analysis of the population process. However, they do not assume a low-dimensional structure and only provide very crude $\mathrm{poly}(d)$-style sample complexity bounds.

## 1.2 OUR CONTRIBUTIONS

We summarize our contributions as follows:

- We demonstrate that information exponent alone is insufficient to characterize certain structures in the learning task and show that for a specific orthogonal multi-index model, if we consider both the lower- and higher-order terms, the sample complexity of strong recovery using online SGD can be greatly improved over the vanilla information exponent-based analysis.

- In the analysis, we prove that when the second-order term is isotropic, the initial randomness can be preserved during training and the relevant subspace can be recovered using $\tilde{O}(d\,\mathrm{poly}(P))$ samples. To the best of our knowledge, this has only been shown by the concurrent work (Ben Arous et al. (2024)) in a different setting.

- As a by-product, we provide a collection of user-friendly technical lemmas to analyze difference between noisy one-dimensional processes and their deterministic counterparts, which may be of independent interests (see Section 4.1 and Section F.2).

## 2 SETUP AND MAIN RESULT

In this section, we describe the setting of our learning task and the training algorithm. Then we formally state our main result. We will also convert the problem to an orthogonal tensor decomposition task using the standard Hermite argument (Ge et al. (2018)).

**Notations** We use $\left\lVert\cdot\right\rVert_p$ to denote the $p$-norm of a vector. When $p = 2$, we often drop the subscript and simply write $\left\lVert\cdot\right\rVert$. For $a, b, \delta \in \mathbb{R}$, $a = b \pm \delta$ means $|a - b| \leq |\delta|$ and $a \vee b = \max\{a, b\}$ and $a \wedge b = \min\{a, b\}$. Beside the standard asymptotic (big $O$) notations, we also use the notation $f_d = O_L(g_d)$, which means there exists a constant $C_L > 0$ that can depend only on $L$ such that $f_d \leq C_L g_d$ for all large enough $d$. Sometimes we also write $f_d \lesssim_L g_d$ for $f_d = O_L(g_d)$. The actual value of $C_L$ can vary between lines, but we will typically point this out when it does.

## 2.1 INPUT AND TARGET FUNCTION

We assume the input $\boldsymbol{x}$ follows the standard Gaussian distribution $\mathcal{N}(0, \boldsymbol{I}_d)$ and the target function has form $f_*(\boldsymbol{x}) = \sum_{k=1}^{P} \phi(\boldsymbol{v}_k^* \cdot \boldsymbol{x})$, where $\log^C d \leq P \leq d$ for a large universal constant $C > 0$, $\{\boldsymbol{v}_k^*\}_{k=1}^{P}$ are orthonormal and $\phi(z) = h_2(z) + h_{2L}(z)$ with $L \geq 2$ and $h_l : \mathbb{R} \to \mathbb{R}$ being the $l$-th (normalized) Hermite polynomial.

Our target model and algorithm will all be invariant under rotation. Hence, we may assume without loss of generality that $\boldsymbol{v}_k^* = \boldsymbol{e}_k$ where $\{\boldsymbol{e}_k\}_k$ is the standard basis of $\mathbb{R}^d$. For now, we continue writing $\boldsymbol{v}_k^*$ since most of the results in this section do not depend on the orthonormality of $\{\boldsymbol{v}_k^*\}_k$.

## 2.2 LEARNER MODEL, LOSS FUNCTION AND ITS GRADIENT

Our learner model is a width-$m$ two-layer network $f(\boldsymbol{x}) := f(\boldsymbol{x}; \boldsymbol{a}, \boldsymbol{V}) := \sum_{i=1}^{m} a_i \phi(\boldsymbol{v}_i \cdot \boldsymbol{x})$, where $\boldsymbol{a} = (a_1, \ldots, a_m) \in \mathbb{R}^m$ and $\boldsymbol{V} = (\boldsymbol{v}_1, \ldots, \boldsymbol{v}_m) \in (\mathbb{S}^{d-1})^m$ are the trainable parameters. We will call $\{\boldsymbol{v}_i\}_{i \in [m]}$ the first-layer neurons. We measure the difference between the learner and the target

model using the mean-square error (MSE). Given a sample $(\boldsymbol{x}, f_*(\boldsymbol{x}))$, we define the per-sample loss as

$$l(\boldsymbol{x}) := l(\boldsymbol{x}; \boldsymbol{a}, \boldsymbol{V}) := \frac{1}{2} \left(f_*(\boldsymbol{x}) - f(\boldsymbol{x})\right)^2.$$

For convenience, we denote the population MSE loss with $\mathcal{L} := \mathcal{L}(\boldsymbol{a}, \boldsymbol{V}) := \mathbb{E}_{\boldsymbol{x}} \, l(\boldsymbol{x}; \boldsymbol{a}, \boldsymbol{V})$. With Hermite expansion, one can rewrite $\mathcal{L}$ as a tensor decomposition loss as in the following lemma. The proof of this lemma is standard and can be found in, for example, Ge et al. (2018). We also provide a proof in Appendix A for completeness.

**Lemma 2.1** (Population loss). *Consider the setting described above. For $l \in \mathbb{N}_{\geq 0}$, let $\hat{\phi}_l$ denote the $l$-th Hermite coefficient of $\phi$ (with respect to the normalized Hermite polynomials). Then, for the population loss, we have*

$$\mathcal{L} = \text{Const.} - \sum_{l=0}^{\infty} \sum_{k=1}^{P} \sum_{j=1}^{m} a_j \hat{\phi}_l^2 \langle \boldsymbol{v}_k^*, \boldsymbol{v}_j \rangle^l + \frac{1}{2} \sum_{l=0}^{\infty} \sum_{j_1,j_2=1}^{m} a_{j_1} a_{j_2} \hat{\phi}_l^2 \langle \boldsymbol{v}_{j_1}, \boldsymbol{v}_{j_2} \rangle^l, \qquad (1)$$

*where* Const. *is a real number that does not depend on $\boldsymbol{a}$ nor $\boldsymbol{V}$.*

**Remark**. The lemma does not require $\{\boldsymbol{v}_k^*\}_k$ to be orthonormal nor $\phi = h_2 + h_{2L}$. All we need is $\phi \in L^2(\mathcal{N}(0, \boldsymbol{I}_d))$ so that the Hermite expansion is well-defined. ♣

For the per-sample and population gradients, we have the following lemma, the proof of which can also be found in Appendix A.

**Lemma 2.2** (First-layer gradients). *Consider the setting described above. Suppose that $\phi = h_2 + h_{2L}$ and $|a_i| \leq a_0$ for some $a_0 > 0$ and all $i \in [m]$. Then, for each $i \in [m]$, we have*

$$\nabla_{\boldsymbol{v}_i} \mathcal{L} = -2a_i \sum_{k=1}^{P} \langle \boldsymbol{v}_k^*, \boldsymbol{v}_i \rangle \boldsymbol{v}_k^* - 2L a_i \sum_{k=1}^{P} \langle \boldsymbol{v}_k^*, \boldsymbol{v}_i \rangle^{2L-1} \boldsymbol{v}_k^* \pm_2 2L m a_0^2, \qquad (2)$$

*where $\boldsymbol{z} = \boldsymbol{z}' \pm_2 \delta$ means $\|\boldsymbol{z} - \boldsymbol{z}'\|_2 \leq \delta$.*

*Moreover, for $\boldsymbol{x} \sim \mathcal{N}(0, \boldsymbol{I}_d)$ and every direction $\boldsymbol{u} \in \mathbb{S}^{d-1}$ that is independent of $\boldsymbol{x}$, there exists a constant $C_L > 0$ that can depend only on $L$ such that*

$$\mathbb{P}\left(a_0^{-1} |\langle \nabla_{\boldsymbol{v}_i} l(\boldsymbol{x}) - \nabla_{\boldsymbol{v}} \mathcal{L}, \boldsymbol{u} \rangle| \geq s\right) \leq C_L \exp\left(-\frac{1}{C_L} \left(\frac{s}{P}\right)^{1/(2L)}\right),$$

$$\mathbb{P}\left(a_0^{-1} \|\nabla_{\boldsymbol{v}_i} l(\boldsymbol{x}) - \nabla_{\boldsymbol{v}} \mathcal{L}\| \geq s\right) \leq C_L \exp\left(\log d - \frac{1}{C_L} \left(\frac{s}{P\sqrt{d}}\right)^{1/(2L)}\right),$$

$$a_0^{-2} \mathbb{E}_{\boldsymbol{x}} \langle \nabla_{\boldsymbol{v}_i} l(\boldsymbol{x}), \boldsymbol{u} \rangle^2 \leq C_L P^2.$$

**Remark on the population gradient**. Note that (2) implies that when $\boldsymbol{a}$ is small, the dynamics of different neurons are approximately decoupled. This allows us to consider each neuron separately. The same is also true when we consider the per-sample gradient. Hence, we can often drop the subscript $i$ and say $\boldsymbol{v} := \boldsymbol{v}_i$ is an arbitrary first-layer neuron and the (population) gradient with respect to it is given by (2). ♣

**Remark on the tail bounds**. We will choose $m = \text{poly}(P)$. In this case, in order for the RHS of the bounds to be $o(1)$ (after applying the union bound over all $m$ neurons), it suffices to choose $s = \omega(P \log^{2L} P)$ and $s = \omega(P d^{1/2} \log^{2L} d)$. Up to some logarithmic terms, this matches what one should expect when $\nabla_{\boldsymbol{v}_i} l(\boldsymbol{x})$ is a $P^2$-subgaussian random vector. ♣

**Remark on possible extensions**. The formula (2) and the tail and variance bounds in this lemma are essentially all the structures we need (besides the orthonormality) to establish our results. To extend our results to general even link function whose Hermite coefficients decay sufficiently fast, first note that the second-order and then the $2L$-th order (the lowest even order that is larger than 2) terms dominate the gradient. Moreover, since $\{\boldsymbol{v}_k^*\}_k$ are assumed to be orthonormal, for any fixed

even order (that is larger than 4), the minimizer of the corresponding terms matches the ground-truth directions, and the gradient will always push the neurons toward one of the ground-truth directions. In other words, they only help the model recover the directions. We consider only the lowest order since it determines the overall complexity (as in the theory of information exponent).

Our tail bound is based on Theorem 1.3 of Adamczak & Wolff (2015) (cf. Theorem A.1), which deals with polynomials of a fixed degree. Theorem 1.2 of Adamczak & Wolff (2015) deals with general functions with controlled higher-order derivatives and can be used to extend our result to non-polynomial link functions. See Appendix G for an empirical evidence. ♣

### 2.3 TRAINING ALGORITHM

Now, we describe the training algorithm. First, we initialize each output weight $a_i$ to be $a_0$ where $a_0 > 0$ is a hyperparameter to be determined later and $\boldsymbol{v}_i \sim \mathrm{Unif}(\mathbb{S}^{d-1})$ independently. Then, we fix the output weights $\boldsymbol{a}$ and train the first-layer weight $\boldsymbol{v}_i$ using online (spherical) SGD with step size $\eta/a_0$ ($\eta > 0$) for $T$ iterations. Then, we fix the first-layer weights and use ridge regression to train the output weights $\boldsymbol{a}$.

Let $\{(\boldsymbol{x}_t, f_*(\boldsymbol{x}_t))\}_{t \in \mathbb{N}}$ be our samples where $\{\boldsymbol{x}_t\}$ are i.i.d. standard Gaussian vectors, and let $\tilde{\nabla}_{\boldsymbol{v}} = (\boldsymbol{I} - \boldsymbol{v}\boldsymbol{v}^\top)\nabla_{\boldsymbol{v}}$ denote the spherical gradient. Then, we can formally describe the training procedure as follows:

Initialization: $\quad a_{0,i} = a_0, \quad \boldsymbol{v}_{0,i} \overset{\text{i.i.d.}}{\sim} \mathrm{Unif}(\mathbb{S}^{d-1}), \qquad\qquad\qquad \forall i \in [m];$

$$
\text{Stage 1:} \quad
\begin{cases}
\hat{\boldsymbol{v}}_{t+1,i} = \boldsymbol{v}_{t,i} - \dfrac{\eta}{a_0}\tilde{\nabla}_{\boldsymbol{v}_i} l(\boldsymbol{x}_t; a_0, \boldsymbol{V}_t), \\[2mm]
\boldsymbol{v}_{t+1,i} = \dfrac{\hat{\boldsymbol{v}}_{t+1,i}}{\|\hat{\boldsymbol{v}}_{t+1,i}\|},
\end{cases}
\quad \forall i \in [m], t \in [T]; \tag{3}
$$

$$
\text{Stage 2:} \quad \boldsymbol{a} = \underset{\boldsymbol{a}'}{\mathrm{argmin}}\ \frac{1}{2N}\sum_{n=1}^{N} l(\boldsymbol{x}_{T+n}; \boldsymbol{a}', \boldsymbol{V}_T) + \lambda \|\boldsymbol{a}'\|^2.
$$

Here, the hyperparameters are the initialization scale $a_0 > 0$, network width $m > 0$, step size $\eta > 0$, time horizon $T > 0$, the number of samples $N$ in Stage 2, and the regularization strength $\lambda > 0$.

Before move on, we make some remarks here on the training algorithm. As we have seen in Lemma 2.1 and Lemma 2.2, when the second-layer weights are small, the dynamics of the first-layer weights are roughly decoupled. Hence, we choose to initialize each $a_i$ small and fix them at $a_0$ in Stage 1. We rescale the learning rate with $1/a_0$ to compensate the fact that the first-layer gradients are proportional to $a_0$.

We will show that after the first stage, for each ground truth direction $\boldsymbol{v}_k^*$, there will be some neurons $\boldsymbol{v}_i$ that converge to that direction. As a result, in the second stage, we can use ridge regression to pick out those neurons and use them to fit the target function. The analysis of this stage is standard and has been done in (Damian et al. (2022); Abbe et al. (2022); Ba et al. (2022); Lee et al. (2024); Oko et al. (2024)). Hence, we will not further discuss this stage in the main text and defer the proofs for this stage to Appendix D.

### 2.4 MAIN RESULT

The following is our main result. The proof of it can be found in Appendix E.

**Theorem 2.1** (Main Theorem). *Consider the setting and algorithm described above. Let $C > 0$ be a large universal constant. Suppose that $\log^C d \le P \le d$ and $\{\boldsymbol{v}_k^*\}_{k=1}^P$ are orthonormal. Let $\delta_{\mathbb{P}} \in (\exp(-\log^C d), 1)$ and $\varepsilon_* > 0$ be given. Suppose that we choose $a_0, \eta, T, N$ satisfying*

$$
m = \Omega\left(P^8 \log^{1.5}(P \vee 1/\delta_{\mathbb{P}})\right), \quad a_0 = O_L\left(\frac{\varepsilon_*^2}{mdP^{2L+2}\log^3 d \log(1/\varepsilon_*)}\right), \quad N = \Omega_L\left(\frac{Pm}{\varepsilon_*^2\delta_{\mathbb{P}}^2}\right),
$$

$$
\eta = O_L\left(\frac{\varepsilon_*^4\delta_{\mathbb{P}}}{dP^{L+8}\log^{4L+1}(d/\delta_{\mathbb{P}})}\right) = \tilde{O}_L\left(\frac{\varepsilon_*^4\delta_{\mathbb{P}}}{dP^{L+8}}\right),
$$

$$
T = O_L\left(\frac{\log d + P^{L-1} + \log(P/\varepsilon_*)}{\eta}\right) = \tilde{O}_L\left(\frac{dP^{2L+7}}{\delta_{\mathbb{P}}\varepsilon_*^4}\right).
$$

*Then, there exists some $\lambda > 0$ such that at the end of training, we have $\mathcal{L}(\boldsymbol{a}, \boldsymbol{V}) \leq \varepsilon_*$ with probability at least $1 - O(\delta_{\mathbb{P}})$.*

## 3 THE GRADIENT FLOW ANALYSIS

In this section, we consider the situation where the training algorithm in Stage 1 is gradient flow over the population loss instead of online SGD. The discussion here is non-rigorous and our formal proof does not rely on anything in this section. Nevertheless, this gradient flow analysis will provide valuable intuition on the behavior of online SGD and also lead to rough guesses on the time complexity.

For notational simplicity, we will assume without loss of generality that $\boldsymbol{v}_k^* = \boldsymbol{e}_k$. Let $\boldsymbol{v}$ be an arbitrary first-layer neuron. By Lemma 2.2, when we rescale the time by $a_0^{-1}$, the dynamics of $\boldsymbol{v}$ are controlled by[3]

$$\dot{\boldsymbol{v}}_\tau \approx 2 \sum_{k=1}^P v_k (\boldsymbol{I} - \boldsymbol{v}\boldsymbol{v}^\top) \boldsymbol{e}_k + 2L \sum_{k=1}^P v_k^{2L-1} (\boldsymbol{I} - \boldsymbol{v}\boldsymbol{v}^\top) \boldsymbol{e}_k.$$

The second term on the RHS comes from the normalized/projection. For each $k \in [d]$, we have

$$\frac{\mathrm{d}}{\mathrm{d}\tau} v_k^2 \approx 4 \mathbb{1}\{k \leq P\} \left(1 + L v_k^{2L-2}\right) v_k^2 - 4 \left(\|\boldsymbol{v}_{\leq P}\|^2 + L \|\boldsymbol{v}_{\leq P}\|_{2L}^{2L}\right) v_k^2. \tag{4}$$

We further split Stage 1 into two substages. In Stage 1.1, the second-order terms dominate and $\|\boldsymbol{v}_{\leq P}\|^2 / \|\boldsymbol{v}_{>P}\|^2$ grows from $\Theta(P/d)$ to $\Theta(1)$. In Stage 1.2, $\boldsymbol{v}$ converges to one ground-truth direction.

The direction to which $\boldsymbol{v}$ will converge depends on the index of the largest $v_k^2$ at the beginning of Stage 1.2. With some standard concentration/anti-concentration argument, one can show that $\max_{k \in [P]} v_k^2$ is at least $1 + c$ times larger than the second-largest $v_k^2$ for a small constant $c > 0$ with probability at least $1/\operatorname{poly}(P)$ at initialization (of Stage 1.1). Hence, as long as this gap can be preserved throughout Stage 1, we can choose $m = \operatorname{poly}(P)$ to ensure all ground-truth directions can be found after Stage 1.2.

### 3.1 STAGE 1.1: LEARNING THE SUBSPACE AND PRESERVATION OF THE GAP

In this substage, we track $\|\boldsymbol{v}_{\leq P}\|^2 / \|\boldsymbol{v}_{>P}\|^2$ and $v_p^2 / v_q^2$[4] where $p, q \in [P]$ are arbitrary. The goal is to show that $\|\boldsymbol{v}_{\leq P}\|^2 / \|\boldsymbol{v}_{>P}\|^2$ will grow to a constant while $v_p^2 / v_q^2$ stay close to its initial value.

For the norm ratio, by (4), we have

$$\frac{\mathrm{d}}{\mathrm{d}\tau} \frac{\|\boldsymbol{v}_{\leq P}\|^2}{\|\boldsymbol{v}_{>P}\|^2} = \frac{\frac{\mathrm{d}}{\mathrm{d}\tau} \|\boldsymbol{v}_{\leq P}\|^2}{\|\boldsymbol{v}_{>P}\|^2} - \frac{\|\boldsymbol{v}_{\leq P}\|^2}{\|\boldsymbol{v}_{>P}\|^2} \frac{\frac{\mathrm{d}}{\mathrm{d}\tau} \|\boldsymbol{v}_{>P}\|^2}{\|\boldsymbol{v}_{>P}\|^2}$$

$$= \frac{4 \|\boldsymbol{v}_{\leq P}\|^2}{\|\boldsymbol{v}_{>P}\|^2} + \frac{4L \|\boldsymbol{v}_{\leq P}\|_{2L}^{2L}}{\|\boldsymbol{v}_{>P}\|^2} - \frac{4 \left(\|\boldsymbol{v}_{\leq P}\|^2 + L \|\boldsymbol{v}_{\leq P}\|_{2L}^{2L}\right) \|\boldsymbol{v}_{\leq P}\|^2}{\|\boldsymbol{v}_{>P}\|^2}$$

$$+ \frac{\|\boldsymbol{v}_{\leq P}\|^2}{\|\boldsymbol{v}_{>P}\|^2} \frac{4 \left(\|\boldsymbol{v}_{\leq P}\|^2 + L \|\boldsymbol{v}_{\leq P}\|_{2L}^{2L}\right) \|\boldsymbol{v}_{>P}\|^2}{\|\boldsymbol{v}_{>P}\|^2}.$$

In particular, note that the terms coming from normalization cancel with each other. Moreover, this implies $\frac{\mathrm{d}}{\mathrm{d}\tau} \frac{\|\boldsymbol{v}_{\leq P}\|^2}{\|\boldsymbol{v}_{>P}\|^2} \geq 4 \frac{\|\boldsymbol{v}_{\leq P}\|^2}{\|\boldsymbol{v}_{>P}\|^2}$, and therefore, it takes only at most $\frac{1+o(1)}{4} \log(d/P) = \Theta(\log(d/P))$ amount of time for the ratio to grow from $\Theta(P/d)$ to $\Theta(1)$. If we choose a small step size $\eta$ so that online SGD closely tracks the gradient flow, then the number of steps one should expect is $O(\log(d/P)/\eta)$.

---

[3]We use $\tau$ to index the time in this continuous-time process (as $t$ has been used to index the steps in the discrete-time process) and will often omit it when it is clear from the context.

[4]A slightly different quantity will be used in the online SGD analysis, but the intuition remains the same.

Meanwhile, for any $p, q \in [P]$, we have

$$\frac{\mathrm{d}}{\mathrm{d}\tau} \frac{v_p^2}{v_q^2} = 4 \left( 1 + L v_p^{2L-2} \right) \frac{v_p^2}{v_q^2} - 4 \left( \|\boldsymbol{v}_{\leq P}\|^2 + L \|\boldsymbol{v}_{\leq P}\|_{2L}^{2L} \right) \frac{v_p^2}{v_q^2}$$

$$- \frac{v_p^2}{v_q^2} \left( 4 \left( 1 + L v_q^{2L-2} \right) - 4 \left( \|\boldsymbol{v}_{\leq P}\|^2 + L \|\boldsymbol{v}_{\leq P}\|_{2L}^{2L} \right) \right) = 4L \left( v_p^{2L-2} - v_q^{2L-2} \right) \frac{v_p^2}{v_q^2}.$$

Note that not only those terms coming from normalization cancel with each other, but also the second-order terms. In particular, this also implies that we cannot learn the directions using only the second-order terms. At initialization, it is unlikely that some $v_k^2$ are significantly larger than all other $v_l^2$. Hence, if we assume the induction hypothesis $v_p^2/v_q^2 \approx v_{0,p}^2/v_{0,q}^2$, we will have $v_k^2 \leq \tilde{O}(1/P)$ and the above will become $\frac{\mathrm{d}}{\mathrm{d}\tau} v_p^2/v_q^2 \leq \tilde{O}(L/P) v_p^2/v_q^2$. As a result, $v_{t,p}^2/v_{t,q}^2 \leq (1 + o(1)) v_{0,p}^2/v_{0,q}^2$ for any $t \leq \Theta(\log(d/P))$, as long as $P \geq \operatorname{poly} \log d$.

## 3.2 STAGE 1.2: LEARNING THE DIRECTIONS

Let $\boldsymbol{v}$ be a first-layer neuron with $v_1^2 \geq (1+c) \max_{2 \leq k \leq P} v_k^2$ for some small constant $c > 0$ at initialization. By our previous discussion, we know at the end of Stage 1.1, the above bound still holds with a potentially smaller constant $c > 0$. In addition, since $\|\boldsymbol{v}_{\leq P}\|^2 = \Theta(1)$, we also have $v_1^2 \geq \Omega(1/P)$ at the end of Stage 1.1. We claim that $\boldsymbol{v}$ will converge to $\boldsymbol{e}_1$. The argument here is similar to the proofs in Li et al. (2020) and Ge et al. (2021).

Again, by (4), we have

$$\frac{\mathrm{d}}{\mathrm{d}\tau} v_1^2 \approx 4 \left( 1 - \|\boldsymbol{v}_{\leq P}\|^2 + L v_1^{2L-2} - L \|\boldsymbol{v}_{\leq P}\|_{2L}^{2L} \right) v_1^2 \geq 4L \left( v_1^{2L-2} - \|\boldsymbol{v}_{\leq P}\|_{2L}^{2L} \right) v_1^2.$$

Assume the induction hypothesis $v_1^2 \geq (1+c) \max_{2 \leq k \leq P} v_k^2$ and write

$$v_1^{2L-2} - \|\boldsymbol{v}_{\leq P}\|_{2L}^{2L} = v_1^{2L-2} \left( 1 - v_1^2 \right) - \left( \|\boldsymbol{v}_{\leq P}\|^2 - v_1^2 \right) \sum_{k=2}^{P} \frac{v_k^2}{\|\boldsymbol{v}_{\leq P}\|^2 - v_1^2} v_k^{2L-2}.$$

Note that the summation is a weighted average of $\{v_k^{2L-2}\}_{k \geq 2}$ and therefore is upper bounded by $\left( v_1^2/(1+c) \right)^{L-1} \leq (1 - c_L) v_1^{2L-2}$ for some constant $c_L > 0$ that can only depend on $L$. Thus, we have

$$\frac{\mathrm{d}}{\mathrm{d}\tau} v_1^2 \gtrsim 4L \left( 1 - v_1^2 - \left( \|\boldsymbol{v}_{\leq P}\|^2 - v_1^2 \right) (1 - c_L) \right) v_1^{2L} \geq 4 c_L L \left( 1 - v_1^2 \right) v_1^{2L}.$$

When $v_1^2 \leq 3/4$, this implies $\frac{\mathrm{d}}{\mathrm{d}\tau} v_1^2 \geq c_L L v_1^{2L}$. As a result, it takes at most $O_L(P^{L-1})$ amount of time for $v_1^2$ to grow from $\Omega(1/P)$ to $3/4$. It is important that $v_1^2 = \Omega(1/P)$ instead of $\Omega(1/d)$ at the start of Stage 1.2, since otherwise the time needed will be $O_L(d^{L-1})$. After $v_1^2$ reaches $3/4$, we have $\frac{\mathrm{d}}{\mathrm{d}\tau} (1 - v_1^2) \leq -4 c_L L (3/4)^{2L} \left( 1 - v_1^2 \right)$. Thus, $v_1^2$ will converge linearly to 1 afterwards.

## 4 FROM GRADIENT FLOW TO ONLINE SGD

In this section, we discuss how to convert the previous gradient flow analysis to an online SGD one. Our actual proof will be based directly on the online SGD analysis, but the overall idea is still proving that the online SGD dynamics of certain important quantities closely track their population gradient descent (GD) counterparts. Our choice of learning rate $\eta$ will be much smaller than what needed for GD to track GF — the bottleneck comes from the GD-to-SGD conversion, not the GF-to-GD one. In other words, provided that SGD tracks GD well, the number of steps/samples it needs to finish each substage is roughly the amount of time GF needs, divided by the step size $\eta$.

The rest of this section is organized as follows. In Section 4.1, we collect a few useful lemmas for controlling the difference between noisy dynamics and their deterministic counterparts. The idea behind them has appeared in Ben Arous et al. (2021) and is also used in Abbe et al. (2022). Here, we simplify and slightly generalize their argument and provide a user-friendly interface. When used properly, it reduces the GD-to-SGD proof to routine calculus. Then, in Section 4.2, we discuss how to apply those general results to analyze the dynamics of online SGD in our setting.

## 4.1 Technical lemmas for analyzing general noisy dynamics

We start with the lemma that will be used to analyze $\|\boldsymbol{v}_{\leq P}\|^2 / \|\boldsymbol{v}_{>P}\|^2$. The proof of it and all other lemmas in this subsection can be found in Section F.2.

**Lemma 4.1.** *Let* $(\Omega, \mathcal{F}, (\mathcal{F}_t)_{t \in \mathbb{N}}, \mathbb{P})$ *be a filtered probability space. Suppose that* $(X_t)_t$ *is an* $(\mathcal{F}_t)_t$-*adapted real-valued process satisfying*

$$X_{t+1} = X_t + \alpha X_t + \xi_{t+1} + Z_{t+1}, \quad X_0 = x_0 > 0, \tag{5}$$

*where* $\alpha > 0$ *is fixed,* $(\xi_t)_t$ *is an* $(\mathcal{F}_t)_t$-*adapted process, and* $(Z_t)_t$ *is an* $(\mathcal{F}_t)_t$-*adapted martingale difference sequence. Define its deterministic counterpart as* $x_t = (1 + \alpha)^t x_0$.

*Let* $T > 0$ *and* $\delta_{\mathbb{P}} \in (0, 1)$ *be given. Suppose that there exists some* $\delta_{\mathbb{P}, \xi} \in (0, 1)$ *and* $\Xi, \sigma_Z > 0$ *such that for every* $t \leq T$, *if* $X_t = (1 \pm 0.5)x_t$, *then we have* $|\xi_{t+1}| \leq (1 + \alpha)^t \Xi$ *with probability at least* $1 - \delta_{\mathbb{P}, \xi}$ *and* $\mathbb{E}[Z_{t+1}^2 \mid \mathcal{F}_t] \leq (1 + \alpha)^t \sigma_Z^2$. *If*

$$\Xi \leq \frac{x_0}{4T} \quad and \quad \sigma_Z^2 \leq \frac{\delta_{\mathbb{P}} \alpha x_0^2}{16}, \tag{6}$$

*then we have* $X_t = (1 \pm 0.5)x_t$ *for all* $t \in [T]$ *with probability at least* $1 - T\delta_{\mathbb{P}, \xi} - \delta_{\mathbb{P}}$.

**Remark on condition** (6). One may interpret $Z_{t+1}$ as those terms coming from the difference between the population and mini-batch gradients and $\xi_{t+1}$ as the higher-order error terms. $\alpha$ is usually small. In our case, it is proportional to the step size $\eta$. $T$ is usually the time needed for $X_t$ to grow from a small $x_0 > 0$ to $\Theta(1)$, which is roughly $\alpha^{-1} \log(1/x_0)$. In other words, we have $\alpha = \tilde{O}(1/T)$. As a result, in order for (6) to hold, it suffices to have $\Xi = O(x_0/T)$ and $\sigma_Z = O(x_0/\sqrt{T})$. Note that the condition on $\sigma_Z$ is much weaker than the condition on $\Xi$. Meanwhile, since $\xi_{t+1}$ models the higher-order error terms, we should expect it to be able to satisfy the stronger condition $\Xi \leq O(1/T)$. ♣

**Remark on stochastic induction**. One important feature of this lemma is that it only requires the bounds $|\xi_{t+1}| \leq (1 + \alpha)^t \Xi$ and $\mathbb{E}[Z_{t+1}^2 \mid \mathcal{F}_t] \leq (1 + \alpha)^t \sigma_Z^2$ to hold when $X_t = (1 \pm 0.5)x_t$. This can be viewed as a form of induction. This is particularly useful when considering the dynamics of, say, $v_k^2$. Similar to how the RHS of $\frac{\mathrm{d}}{\mathrm{d}\tau} v_{\tau,k}^2 = 2v_{\tau,k} \dot{v}_{\tau,k}$ depends on $v_{\tau,k}$, the size of $\xi_{t+1}$ and $Z_{t+1}$ will usually depend on $X_t$. Hence, we will not be able to bound them without an induction hypothesis on $X_t$. ♣

**Remark on the dependence on** $\delta_{\mathbb{P}}$. The dependence on $\delta_{\mathbb{P}}$ can be improved to $\mathrm{poly} \log(1/\delta_{\mathbb{P}})$ if we have tail bounds on $Z_{t+1}$ similar to the ones in Lemma 2.2. We state this lemma in this simpler form because we will only take union bound over $\mathrm{poly}(P)$ events, and we are not optimizing the dependence on $P$. We include in Section F.2 an example (cf. Lemma F.9 and Lemma F.10) where this improvement is made (though that result will not be used in the proof). ♣

*Proof sketch of Lemma 4.1.* For the ease of presentation, we assume that $|\xi_{t+1}| \leq (1 + \alpha)^t \Xi$ with probability at least $1 - \delta_{\mathbb{P}, \xi}$ and $\mathbb{E}[Z_{t+1}^2 \mid \mathcal{F}_t] \leq (1 + \alpha)^t \sigma_Z^2$ always hold. Recursively expand the RHS of (5), and we obtain

$$X_{t+1} = (1 + \alpha)^{t+1} x_0 + \sum_{s=1}^{t} (1 + \alpha)^{t-s} \xi_{s+1} + \sum_{s=1}^{t} (1 + \alpha)^{t-s} Z_{s+1}.$$

Divide both sides with $(1 + \alpha)^{t+1}$ and replace $t + 1$ with $t$. Then, the above becomes

$$X_t (1 + \alpha)^{-t} = x_0 + \sum_{s=1}^{t} (1 + \alpha)^{-s} \xi_s + \sum_{s=1}^{t} (1 + \alpha)^{-s} Z_s.$$

The second term is bounded by $T\Xi$ (uniformly over $t \leq T$) with probability at least $1 - T\delta_{\mathbb{P}, \xi}$. Note that $(1 + \alpha)^{-s} Z_s$ is still a martingale difference sequence. Hence, by Doob's $L^2$-submartingale inequality, the third term is bounded by $x_0/4$ with probability at least $16\sigma_Z^2/(\alpha x_0^2)$. Thus, when (6) holds, the RHS is $(1 \pm 0.5)x_0$ with probability at least $1 - T\delta_{\mathbb{P}, \xi} - \delta_{\mathbb{P}}$. Multiply both sides with $(1 + \alpha)^t$, and we complete the proof. □

Using the same strategy, one can prove a similar lemma (cf. Lemma F.8) that deals with the case $\alpha = 0$, which will be used to show the preservation of the gap in Stage 1.1. Another interesting case is where the growth is not linear but polynomial. This is the case of Stage 1.2 in our setting. For this case, we have the following lemma.

**Lemma 4.2.** *Suppose that $(X_t)_t$ satisfies*

$$X_{t+1} = X_t + \alpha X_t^p + \xi_{t+1} + Z_{t+1}, \quad X_0 = x_0 > 0,$$

*where $p > 1$, the signal growth rate $\alpha > 0$ and initialization $x_0 > 0$ are given and fixed, $(\xi_t)_t$ is an adapted process, and $(Z_t)_t$ is a martingale difference sequence. Let $\hat{x}_t$ be the solution to the deterministic recurrence relationship $\hat{x}_{t+1} = \hat{x}_t + \alpha \hat{x}_t^p, \hat{x}_0 = x_0/2$.*

*Fix $T > 0, \delta_{\mathbb{P}} \in (0,1)$. Suppose that there exist $\Xi, \sigma_Z > 0$ and $\delta_{\mathbb{P},\xi} \in (0,1)$ such that when $X_t \geq \hat{x}_t$, we have $|\xi_t| \leq \Xi$ with probability at least $1 - \delta_{\mathbb{P},\xi}$ and $\mathbb{E}[Z_{t+1} \mid \mathcal{F}_t] \leq \sigma_Z^2$. Then, if $\Xi \leq \frac{x_0}{4T}$ and $\sigma_Z^2 \leq \frac{x_0^2 \delta_{\mathbb{P}}}{16T}$, we have $X_t \geq \hat{x}_t$ for all $t \leq T$.*

The proof is essentially the same as the previous one, except that we need to replace $(1 + \alpha)^t$ with $\prod_{s=0}^{t-1}(1 + \alpha X_s^{p-1})$. Let $x_t$ be the version of $\hat{x}_t$ with the initial value being $x_0$ instead of $x_0/2$. Unlike the linear case, here it is generally difficult to ensure $X_t \geq x_t/2$ since this type of polynomial systems exhibits sharp transitions and blows up in finite time. In fact, the difference between the deterministic processes $\hat{x}_t$ and $x_t/2$ can be large. However, if one is only interested in the time needed for $X_t$ to grow from a small value to a constant, then results obtained from $\hat{x}_t$ and $x_t$ differ only by a multiplicative constant, and when $\alpha > 0$ is small, both of them can be estimated using their continuous-time counterpart $\dot{x}_\tau = x_\tau^p$ (cf. Lemma F.12).

## 4.2 SAMPLE COMPLEXITY OF ONLINE SGD

In this subsection, we demonstrate how to use the previous results to obtain results for online SGD and discuss why the sample complexity is $\tilde{O}(d \operatorname{poly}(P))$ instead of $\tilde{O}(d^{2L-1})$ even though we are relying on the $2L$-th order terms to learn the directions.

### 4.2.1 A SIMPLIFIED VERSION OF STAGE 1.1

As an example, we consider the dynamics of $P v_p^2/(d v_q^2)$ where $p \leq P$ and $q > P$ and assume both of $v_p$ and $v_q$ are small and $P v_p^2/(d v_q^2) \leq 1$. This can be viewed as a simplified version of the analysis of $\|\boldsymbol{v}_{\leq P}\|^2 / \|\boldsymbol{v}_{>P}\|^2$ in Stage 1.1. The analysis of other quantities/stages is essentially the same — we rewrite the update rule to single out martingale difference terms and the higher-order error terms, and apply a suitable lemma from the previous subsection (or Section F.2) to complete the proof.

For the ease of presentation, in this subsection, we ignore the higher-order terms. In particular, we assume the approximation

$$\hat{v}_{t+1,k} \approx v_{t,k} + 2\eta \left( \mathbb{1}\{k \leq P\} - \|\boldsymbol{v}_{\leq P}\|^2 \right) + \eta Z_{t+1,k}, \qquad \forall k \in [d],$$

where $Z_{t+1,k}$ represents the difference between the population and mini-batch gradients. Then, we compute

$$\hat{v}_{t+1,k}^2 \approx \left( 1 + 4\eta \left( \mathbb{1}\{k \leq P\} - \|\boldsymbol{v}_{\leq P}\|^2 \right) \right) v_k^2 + 2\eta v_k Z_k \pm C_L \eta^2 (1 \vee Z_k^2).$$

Here, the last term is the higher-order term and will eventually be included in $\xi$. For simplicity, we will also ignore them in the following discussion. The second term is the martingale difference term. Its (conditional) variance depend on $v_k$, and this necessitates the induction-style conditions in Lemma F.6. Note that $v_{t+1,p}^2/v_{t+1,q}^2 = \hat{v}_{t+1,p}^2/\hat{v}_{t+1,q}^2$. Hence, we have

$$\frac{v_{t+1,p}^2}{v_{t+1,q}^2} \approx \frac{\left( 1 + 4\eta \left( 1 - \|\boldsymbol{v}_{\leq P}\|^2 \right) \right) v_p^2 + 2\eta v_p Z_p}{\left( 1 - 4\eta \|\boldsymbol{v}_{\leq P}\|^2 \right) v_q^2 + 2\eta v_q Z_q}.$$

For any small $a > 0$ and small $\delta > 0$, we have the following elementary identity: $\frac{1}{a+\delta} = \frac{1}{a}\left(1 - \frac{\delta}{a}\left(1 - \frac{\delta}{a+\delta}\right)\right) \approx \frac{1}{a}\left(1 - \frac{\delta}{a}\right)$. Repeatedly use this identity, and we can rewrite the above equation as

$$\frac{Pv_{t+1,p}^2}{dv_{t+1,q}^2} \approx \frac{P\left(1 + 4\eta\left(1 - \|\boldsymbol{v}_{\leq P}\|^2\right)\right)v_p^2}{d\left(1 - 4\eta\|\boldsymbol{v}_{\leq P}\|^2\right)v_q^2}\left(1 - \frac{2\eta v_q Z_q}{\left(1 - 4\eta\|\boldsymbol{v}_{\leq P}\|^2\right)v_q^2}\right)$$

$$+ \frac{2P\eta v_p Z_p}{d\left(1 - 4\eta\|\boldsymbol{v}_{\leq P}\|^2\right)v_q^2}\left(1 - \frac{2\eta v_q Z_q}{\left(1 - 4\eta\|\boldsymbol{v}_{\leq P}\|^2\right)v_q^2}\right)$$

$$\approx (1 + 4\eta)\frac{Pv_p^2}{dv_q^2} - \frac{Pv_p^2}{dv_q^2}\frac{2\eta v_q Z_q}{v_q^2} + \frac{2P\eta v_p Z_p}{dv_q^2}.$$

Suppose that $v_p^2 \approx v_q^2$ at initialization and assume the induction hypothesis $Pv_p^2/(dv_q^2) = (1 \pm 0.5)(1 + 4\eta)^t Pv_{0,p}^2/(dv_{0q}^2)$. Then, by Lemma 2.2, the conditional variance of the martingale difference terms (the last two terms) is bounded by $O_L((1 + 4\eta)^t \eta^2 P^4/d)$. Using the language of Lemma 4.1, this means $\sigma_Z^2 \leq O_L(\eta^2 P^4/d)$. Hence, in order for (the second condition of) (6) to hold, it suffices to choose $\eta \lesssim_L \delta_{\mathbb{P}}/(dP^2)$. By our gradient flow analysis, the number steps Stage 1.1 needs is roughly $\log d/\eta$. In other words, for Stage 1.1, the sample complexity is $\tilde{O}_L(dP^2/\delta_{\mathbb{P}})$ (if we ignore the higher-order error terms).

### 4.2.2 THE IMPROVED SAMPLE COMPLEXITY FOR STAGE 1.2

To see why the existence of the second-order terms can reduce the sample complexity from $d^{\mathrm{IE}-1}$ to $d\,\mathrm{poly}(P)$, first note that after Stage 1.1, $\max_{p \in [P]} v_p^2$ will be $\Omega(1/P)$. Also note that the conditions in Lemma 4.2 depend on the initial value. With the initial value being $\Omega(1/P)$ instead of $\tilde{O}(1/d)$, the largest possible step size we can choose will be $O(1)/(d\,\mathrm{poly}(P))$, which is much larger than the usual $O(1/d^{L-1})$ requirement from the vanilla information exponent argument. Meanwhile, by our gradient flow analysis, we know the number of iterations needed is $O(P^{L-1}/\eta)$. Combine these and we obtain the $d\,\mathrm{poly}(P)$ sample complexity.

## 5 CONCLUSION AND FUTURE DIRECTIONS

In this work, we study the task of learning multi-index models of form $f_*(\boldsymbol{x}) = \sum_{k=1}^P \phi(\boldsymbol{v}_k^* \cdot \boldsymbol{x})$ with $P \ll d$, $\{\boldsymbol{v}_k^*\}_k$ be orthogonal and $\phi = h_2 + h_{2L}$. By considering both the lower- and higher-order terms, we prove an $\tilde{O}(d\,\mathrm{poly}(P))$ bound on the sample complex for strong recovery of directions using online SGD, which improve the results one can obtain using vanilla information exponent-based analysis.

One possible future direction of our work is to generalize our results to more general link functions and assume the learner model is a generic two-layer network with, say, ReLU activation. Another interesting but more challenging direction is to consider the non-(near)-orthogonal case. We conjecture when the target model has a hierarchical structure across different orders, online SGD can gradually learn the directions using those terms of different order sequentially.

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

TABLE OF CONTENTS

## A   FROM MULTI-INDEX MODEL TO TENSOR DECOMPOSITION

In this section, we show that the task of learning the multi-index target function $f_*(\boldsymbol{x}) = \sum_{k=1}^{P} \phi(\boldsymbol{v}_k^* \cdot \boldsymbol{x})$ can be reduced to tensor decomposition. We will need the following classical result on Hermite polynomials (cf. Chapter 11.2 of O'Donnell (2014)) and correlated Gaussian variables.

**Lemma A.1** (Proposition 11.31 of O'Donnell (2014)). *For $k \in \mathbb{N}_{\geq 0}$ denote the normalized Hermite polynomials. Let $\rho \in [-1, 1]$ and $z, z'$ be $\rho$-correlated standard Gaussian variables. Then, we have*

$$\mathop{\mathbb{E}}_{z,z'} [h_k(z)h_j(z')] = \mathbb{1}\{k = j\}\rho^k.$$

**Lemma 2.1** (Population loss). *Consider the setting described above. For $l \in \mathbb{N}_{\geq 0}$, let $\hat{\phi}_l$ denote the $l$-th Hermite coefficient of $\phi$ (with respect to the normalized Hermite polynomials). Then, for the population loss, we have*

$$\mathcal{L} = \mathrm{Const.} - \sum_{l=0}^{\infty} \sum_{k=1}^{P} \sum_{j=1}^{m} a_j \hat{\phi}_l^2 \langle \boldsymbol{v}_k^*, \boldsymbol{v}_j \rangle^l + \frac{1}{2} \sum_{l=0}^{\infty} \sum_{j_1,j_2=1}^{m} a_{j_1} a_{j_2} \hat{\phi}_l^2 \langle \boldsymbol{v}_{j_1}, \boldsymbol{v}_{j_2} \rangle^l, \tag{1}$$

*where* $\mathrm{Const.}$ *is a real number that does not depend on $\boldsymbol{a}$ nor $\boldsymbol{V}$.*

*Proof.* By definition, we have

$$\mathcal{L} = \frac{1}{2} \mathop{\mathbb{E}}_{\boldsymbol{x} \sim \mathcal{N}(0,\boldsymbol{I}_d)} \left( \sum_{k=1}^{P} \phi(\boldsymbol{v}_k^* \cdot x) - \sum_{j=1}^{m} a_j \phi(\boldsymbol{v}_j \cdot \boldsymbol{x}) \right)^2$$

$$= \frac{1}{2} \sum_{k_1,k_2=1}^{P} \mathop{\mathbb{E}}_{\boldsymbol{x} \sim \mathcal{N}(0,\boldsymbol{I}_d)} \left\{ \phi(\boldsymbol{v}_{k_1}^* \cdot x)\phi(\boldsymbol{v}_{k_2}^* \cdot x) \right\} - \sum_{k=1}^{P} \sum_{j=1}^{m} a_j \mathop{\mathbb{E}}_{\boldsymbol{x} \sim \mathcal{N}(0,\boldsymbol{I}_d)} \left\{ \phi(\boldsymbol{v}_k^* \cdot x)\phi(\boldsymbol{v}_j \cdot \boldsymbol{x}) \right\}$$

$$+ \frac{1}{2} \sum_{j_1,j_2=1}^{m} a_{j_1} a_{j_2} \mathop{\mathbb{E}}_{\boldsymbol{x} \sim \mathcal{N}(0,\boldsymbol{I}_d)} \left\{ \phi(\boldsymbol{v}_{j_1} \cdot x)\phi(\boldsymbol{v}_{j_2} \cdot \boldsymbol{x}) \right\}.$$

The first term is independent of $\boldsymbol{a}$ and $\boldsymbol{V}$. For the other two terms, we now use Lemma A.1 to evaluate the expectation. Let $\phi = \sum_{k=0}^{\infty} \hat{\phi}_k h_k$ be the Hermite expansion of $\phi$ where the convergence is in $L^2$ sense. For any $\rho \in [-1, 1]$ and $\rho$-correlated standard Gaussian variables $z, z'$, we have

$$\mathop{\mathbb{E}}_{z,z'} \{\phi(z)\phi(z')\} = \sum_{k,l=0}^{\infty} \hat{\phi}_k \hat{\phi}_l \mathop{\mathbb{E}}_{z,z'} \{h_k(z)h_l(z')\} = \sum_{k=0}^{\infty} \hat{\phi}_k^2 \rho^k,$$

where the first equality comes from the Dominated Convergence Theorem and the second from Lemma A.1. Note that $\boldsymbol{v}_k^* \cdot \boldsymbol{x}$ and $\boldsymbol{v}_j \cdot \boldsymbol{x}$ are $\langle \boldsymbol{v}_k^*, \boldsymbol{v}_j \rangle$-correlated standard Gaussian variables. Hence, by applying the above identity to the second term, and we obtain

$$\sum_{k=1}^{P} \sum_{j=1}^{m} a_j \mathop{\mathbb{E}}_{\boldsymbol{x} \sim \mathcal{N}(0,\boldsymbol{I}_d)} \{\phi(\boldsymbol{v}_k^* \cdot x)\phi(\boldsymbol{v}_j \cdot \boldsymbol{x})\} = \sum_{l=0}^{\infty} \sum_{k=1}^{P} \sum_{j=1}^{m} a_j \hat{\phi}_l^2 \langle \boldsymbol{v}_k^*, \boldsymbol{v}_j \rangle^l.$$

Similarly, for the last term, we have

$$\frac{1}{2} \sum_{j_1,j_2=1}^{m} a_{j_1} a_{j_2} \mathop{\mathbb{E}}_{\boldsymbol{x} \sim \mathcal{N}(0,\boldsymbol{I}_d)} \{\phi(\boldsymbol{v}_{j_1} \cdot x)\phi(\boldsymbol{v}_{j_2} \cdot \boldsymbol{x})\} = \frac{1}{2} \sum_{l=0}^{\infty} \sum_{j_1,j_2=1}^{m} a_{j_1} a_{j_2} \hat{\phi}_l^2 \langle \boldsymbol{v}_{j_1}, \boldsymbol{v}_{j_2} \rangle^l.$$

□

Then, we consider the population and per-sample gradient. It is well-known that any Lipschitz function of a Gaussian variable is still subgaussian. Similar tail bounds can still be obtained when the function is not Lipschitz but has a bounded higher-order derivative. To estimate the tail of the per-sample gradient, we need the following result from Adamczak & Wolff (2015). As a side note, Theorem 1.2 of Adamczak & Wolff (2015) is a more general result that deals with general non-Lipschitz functions with controlled higher-order derivatives. That result can be used to extend our setting to link functions with infinitely many nonzero higher-order Hermite coefficients, given that they decay sufficiently fast.

**Theorem A.1** (Theorem 1.3 of Adamczak & Wolff (2015))**.** *Let $\boldsymbol{Z} \sim \mathcal{N}(0, \boldsymbol{I}_d)$ and $f : \mathbb{R}^d \to \mathbb{R}$ be a polynomial of degree $Q$. Then, for any $t \geq 0$, we have*

$$\mathbb{P}\left[|f(Z) - \mathbb{E}\,f(\boldsymbol{Z})| \geq t\right] \leq C_Q \exp\left(-C_Q^{-1} \min_{q \in [Q]} \min_{J \in P_q} \left(\frac{t}{\|\mathbb{E}\,\nabla^q f(\boldsymbol{Z})\|_J}\right)^{2/|J|}\right), \quad (7)$$

*where $C_Q > 0$ is a constant that depends only on the degree $Q$, $P_q$ is the collection of partitions of $[q]$, and for any $J \in P_q$ and $\boldsymbol{A} \in (\mathbb{R}^d)^{\otimes q}$,*

$$\|\boldsymbol{A}\|_J := \sup\left\{\sum_{\boldsymbol{i} \in [d]^q} A_{\boldsymbol{i}} \prod_{l=1}^{|J|} X_{\boldsymbol{i}_{J_l}}^{(l)} \; : \; \boldsymbol{X}^{(l)} \in (\mathbb{R}^d)^{\otimes |J_l|}, \left\|\boldsymbol{X}^{(l)}\right\|_F \leq 1, \forall l \in [|J|]\right\}.$$

**Remark on the definition of $\|\cdot\|_J$.** The definition of $\|\boldsymbol{A}\|_J$ might look bizarre, but it has a natural functional interpretation. Given a partition $J \in P_q$, we can treat a tensor $\boldsymbol{A} \in (\mathbb{R}^d)^{\otimes q}$ as a multilinear function by grouping the indices according to $J$ as follows. For each $J_l \in J$, we take $\boldsymbol{X}^{(l)} \in (\mathbb{R}^d)^{|J_l|}$ and feed them into $\boldsymbol{A}$ to obtain a real number. Similar to how the induced norm is defined for matrices, we restrict the norm of each $\boldsymbol{X}^{(l)}$ to be at most 1 to obtain this definition of $\|\boldsymbol{A}\|_J$. As an example, consider $\boldsymbol{A} \in (\mathbb{R}^d)^{\otimes 3}$ and $J = \{\{1, 2\}, \{3\}\}$. In this case, $\boldsymbol{X}^{(1)}$ is a matrix and $\boldsymbol{X}^{(2)}$ is a vector, and we have

$$\|\boldsymbol{A}\|_{\{1,2\},\{3\}} = \sup\left\{\sum_{i,j,k \in [d]} A_{i,j,k} X_{i,j}^{(1)} X_k^{(2)} \; : \; \left\|\boldsymbol{X}^{(1)}\right\|_F \leq 1, \left\|\boldsymbol{X}^{(2)}\right\|_2 \leq 1\right\}.$$

♣

**Remark on the RHS of** (7)**.** Fix $\boldsymbol{z} \in \mathbb{R}^d$ and $f$ be a polynomial with degree at most $Q$. Suppose that the coefficients of monomials of $f$ are all bounded by some constant $A_Q > 0$ that may depend on $Q$. Note that $f$ can contain at most $d^Q$ monomials. Meanwhile, for each $q \in [Q]$ and $\boldsymbol{i} \in [d]^q$, $[\nabla^q f(\boldsymbol{z})]_{\boldsymbol{i}}$ is nonzero only if $[\nabla^q m(\boldsymbol{z})]_{\boldsymbol{i}}$ for some monomial $m : \mathbb{R}^d \to \mathbb{R}$ contained in $f$. Since $m$ has degree at most $Q$, $\nabla^q m(\boldsymbol{z})$ can have at most $Q!$ nonzero entries (across all different $\boldsymbol{z}$). Thus, the total number of possible nonzero entries in $\nabla^q f(\boldsymbol{z})$ is bounded by $Q! d^Q$ and all entries of it are bounded by $Q! A_Q$. Thus, we have $\|\mathbb{E}\,\nabla^q f(\boldsymbol{Z})\|_J \leq C_Q' d^Q$ for some constant $C_Q' > 0$ that can depend only on $Q$. In other words, for the RHS of (7) to be $o(1)$, we need $t = \omega(C_Q' d^Q)$.

The above bound might seem to be bad. Fortunately, in our case, we only need to consider $f : \mathbb{R}^d \to \mathbb{R}$ of form $f(\boldsymbol{x}) = F(\boldsymbol{u}_1 \cdot \boldsymbol{x}, \boldsymbol{u}_2 \cdot \boldsymbol{x}, \boldsymbol{u}_3 \cdot \boldsymbol{x})$ where $F$ is a polynomial and $\boldsymbol{u}_1, \boldsymbol{u}_2, \boldsymbol{u}_3 \in \mathbb{S}^{d-1}$ are three arbitrary directions. Suppose that $\boldsymbol{x} \sim \mathcal{N}(0, \boldsymbol{I}_d)$ and define $\boldsymbol{\Sigma} \in \mathbb{R}^{3 \times 3}$ via $\Sigma_{i,j} = \langle \boldsymbol{u}_i, \boldsymbol{u}_j \rangle$. Then, we have

$$f(\boldsymbol{x}) \stackrel{d}{=} F\left(\boldsymbol{\Sigma}^{1/2} \boldsymbol{z}\right) \quad \text{where} \quad \boldsymbol{z} \sim \mathcal{N}(0, \boldsymbol{I}_3).$$

When $F : \mathbb{R}^3 \to \mathbb{R}$ is a degree-$Q$ polynomial with coefficients being constants that can depend only on $Q$, so $\boldsymbol{z} \mapsto F\left(\boldsymbol{\Sigma}^{1/2} \boldsymbol{z}\right)$. Thus, we can apply this theorem (with dimension being 3) and our previous discussion to obtain

$$\mathbb{P}\left[|f(Z) - \mathbb{E}\,f(\boldsymbol{Z})| \geq t\right] \leq C_Q \exp\left(-\frac{t^{2/Q}}{C_Q}\right),$$

where $C_Q > 0$ is a constant that can depend only on $Q$.

♣

Now, we are ready to prove Lemma 2.2, which we also restate bellow.

**Lemma 2.2** (First-layer gradients). *Consider the setting described above. Suppose that $\phi = h_2 + h_{2L}$ and $|a_i| \leq a_0$ for some $a_0 > 0$ and all $i \in [m]$. Then, for each $i \in [m]$, we have*

$$\nabla_{\boldsymbol{v}_i} \mathcal{L} = -2a_i \sum_{k=1}^{P} \langle \boldsymbol{v}_k^*, \boldsymbol{v}_i \rangle \boldsymbol{v}_k^* - 2La_i \sum_{k=1}^{P} \langle \boldsymbol{v}_k^*, \boldsymbol{v}_i \rangle^{2L-1} \boldsymbol{v}_k^* \pm_2 2Lma_0^2, \tag{2}$$

*where $\boldsymbol{z} = \boldsymbol{z}' \pm_2 \delta$ means $\|\boldsymbol{z} - \boldsymbol{z}'\|_2 \leq \delta$.*

*Moreover, for $\boldsymbol{x} \sim \mathcal{N}(0, \boldsymbol{I}_d)$ and every direction $\boldsymbol{u} \in \mathbb{S}^{d-1}$ that is independent of $\boldsymbol{x}$, there exists a constant $C_L > 0$ that can depend only on $L$ such that*

$$\mathbb{P}\left(a_0^{-1} |\langle \nabla_{\boldsymbol{v}_i} l(\boldsymbol{x}) - \nabla_{\boldsymbol{v}} \mathcal{L}, \boldsymbol{u}\rangle| \geq s\right) \leq C_L \exp\left(-\frac{1}{C_L} \left(\frac{s}{P}\right)^{1/(2L)}\right),$$

$$\mathbb{P}\left(a_0^{-1} \|\nabla_{\boldsymbol{v}_i} l(\boldsymbol{x}) - \nabla_{\boldsymbol{v}} \mathcal{L}\| \geq s\right) \leq C_L \exp\left(\log d - \frac{1}{C_L} \left(\frac{s}{P\sqrt{d}}\right)^{1/(2L)}\right),$$

$$a_0^{-2} \mathop{\mathbb{E}}_{\boldsymbol{x}} \langle \nabla_{\boldsymbol{v}_i} l(\boldsymbol{x}), \boldsymbol{u}\rangle^2 \leq C_L P^2.$$

*Proof.* Fix $i \in [m]$. First, by Lemma 2.1, we have

$$\nabla_{\boldsymbol{v}_i} \mathcal{L} = -\sum_{k=1}^{P} a_i \nabla_{\boldsymbol{v}_i} \langle \boldsymbol{v}_k^*, \boldsymbol{v}_i \rangle^2 - \sum_{k=1}^{P} a_i \nabla_{\boldsymbol{v}_i} \langle \boldsymbol{v}_k^*, \boldsymbol{v}_i \rangle^{2L} + \frac{1}{2} \sum_{l \in \{2, 2L\}} \sum_{j=1}^{m} a_i a_j \nabla_{\boldsymbol{v}_i} \langle \boldsymbol{v}_i, \boldsymbol{v}_j \rangle^l$$

$$= -2a_i \sum_{k=1}^{P} \langle \boldsymbol{v}_k^*, \boldsymbol{v}_i \rangle \boldsymbol{v}_k^* - 2La_i \sum_{k=1}^{P} \langle \boldsymbol{v}_k^*, \boldsymbol{v}_i \rangle^{2L-1} \boldsymbol{v}_k^*$$

$$+ \frac{1}{2} a_i \sum_{l \in \{2, 2L\}} \left( l \sum_{j \in [m] \setminus \{i\}} a_j \langle \boldsymbol{v}_i, \boldsymbol{v}_j \rangle^{l-1} \boldsymbol{v}_j + 2la_i \langle \boldsymbol{v}_i, \boldsymbol{v}_i \rangle^{l-1} \boldsymbol{v}_i \right).$$

Note that the last line is bounded by $2Lma_0^2$. In other words,

$$\nabla_{\boldsymbol{v}_i} \mathcal{L} = -2a_i \sum_{k=1}^{P} \langle \boldsymbol{v}_k^*, \boldsymbol{v}_i \rangle \boldsymbol{v}_k^* - 2La_i \sum_{k=1}^{P} \langle \boldsymbol{v}_k^*, \boldsymbol{v}_i \rangle^{2L-1} \boldsymbol{v}_k^* \pm_2 2Lma_0^2.$$

Now, consider the per-sample gradient. We write

$$\nabla_{\boldsymbol{v}_i} l(\boldsymbol{x}) = -\left(f_*(\boldsymbol{x}) - f(\boldsymbol{x}; \boldsymbol{a}, \boldsymbol{V})\right) \nabla_{\boldsymbol{v}_i} f(\boldsymbol{x}; \boldsymbol{a}, \boldsymbol{V})$$

$$= -a_i \left(f_*(\boldsymbol{x}) - f(\boldsymbol{x}; \boldsymbol{a}, \boldsymbol{V})\right) \phi'(\boldsymbol{v}_i \cdot \boldsymbol{x}) \boldsymbol{x}$$

$$= -a_i \sum_{k=1}^{P} \phi(\boldsymbol{v}_k^* \cdot \boldsymbol{x}) \phi'(\boldsymbol{v}_i \cdot \boldsymbol{x}) \boldsymbol{x} + a_i \sum_{k=1}^{m} a_k \phi(\boldsymbol{v}_k \cdot \boldsymbol{x}) \phi'(\boldsymbol{v}_i \cdot \boldsymbol{x}) \boldsymbol{x}$$

$$=: \boldsymbol{g}_{i,1} + \boldsymbol{g}_{i,2}.$$

Let $\boldsymbol{u} \in \mathbb{S}^{d-1}$ be an arbitrary direction. We now estimate the tail of $\langle \nabla_{\boldsymbol{v}_i} l, \boldsymbol{u} \rangle$. By Theorem A.1 (and the second remark following it), we have

$$\mathbb{P}\left(\left| \phi(\boldsymbol{v}_k^* \cdot \boldsymbol{x}) \phi'(\boldsymbol{v}_i \cdot \boldsymbol{x}) \langle \boldsymbol{x}, \boldsymbol{u} \rangle - \mathop{\mathbb{E}}_{\boldsymbol{x}'} \phi(\boldsymbol{v}_k^* \cdot \boldsymbol{x}') \phi'(\boldsymbol{v}_i \cdot \boldsymbol{x}') \langle \boldsymbol{x}', \boldsymbol{u} \rangle \right| \geq s\right) \leq C_L \exp\left(-\frac{s^{1/(2L)}}{C_L}\right),$$

for some constant $C_L > 0$ that can depend only on $L$. Hence, we have

$$\mathbb{P}\left(a_i^{-1} |\langle \boldsymbol{g}_{i,1}, \boldsymbol{u} \rangle - \mathbb{E}\langle \boldsymbol{g}_{i,1}, \boldsymbol{u} \rangle| \geq s\right) \leq C_L \exp\left(-\frac{(s/P)^{1/(2L)}}{C_L}\right).$$

In particular, this implies that typical value of $a_i^{-1} \boldsymbol{g}_{i,1}$ is bounded by $\Theta(P)$. Similarly, for $\boldsymbol{g}_{i,2}$, we have

$$\mathbb{P}\left(a_0^{-2} |\langle \boldsymbol{g}_{i,1}, \boldsymbol{u} \rangle - \mathbb{E}\langle \boldsymbol{g}_{i,1}, \boldsymbol{u} \rangle| \geq s\right) \leq C_L \exp\left(-\frac{(s/m)^{1/(2L)}}{C_L}\right), \tag{8}$$

or equivalently,

$$\mathbb{P}\left(a_0^{-1}\left|\langle \boldsymbol{g}_{i,1}, \boldsymbol{u}\rangle - \mathbb{E}\langle \boldsymbol{g}_{i,1}, \boldsymbol{u}\rangle\right| \geq s\right) \leq C_L \exp\left(-\frac{(s/(a_0 m))^{1/(2L)}}{C_L}\right).$$

Note that since $a_0 m = o(1) \ll P$, the RHS of this inequality is much smaller than the RHS of (8) when we choose the same $s$. Combine the above bounds together, and we obtain that for each fixed $i \in [m]$,

$$\mathbb{P}\left(a_0^{-1}\left|\langle \nabla_{\boldsymbol{v}_i} l(\boldsymbol{x}), \boldsymbol{u}\rangle - \langle \nabla_{\boldsymbol{v}}\mathcal{L}, \boldsymbol{u}\rangle\right| \geq s\right) \leq C_L \exp\left(-\frac{(s/P)^{1/(2L)}}{C_L}\right),$$

for some constant $C_L > 0$ that can depend only on $L$ and is potentially different from the $C_L$ in (8). As a corollary, we have

$$\mathbb{P}\left(a_0^{-1}\left\|\nabla_{\boldsymbol{v}_i} l(\boldsymbol{x}) - \nabla_{\boldsymbol{v}}\mathcal{L}\right\| \geq s\right)$$

$$\leq C_L \sum_{k=1}^{d} \mathbb{P}\left(a_0^{-1}\left|\langle \nabla_{\boldsymbol{v}_i} l(\boldsymbol{x}), \boldsymbol{e}_k\rangle - \langle \nabla_{\boldsymbol{v}}\mathcal{L}, \boldsymbol{e}_k\rangle\right| \geq s/\sqrt{d}\right)$$

$$\leq C_L \exp\left(\log(d) - \frac{1}{C_L}\left(\frac{s}{P\sqrt{d}}\right)^{1/(2L)}\right).$$

Similarly, one can show that $\mathbb{E}\langle \nabla_{\boldsymbol{v}_i} l(\boldsymbol{x}), \boldsymbol{u}\rangle^2 \leq C_L a_0^2 P^2$ for some constant $C_L > 0$ that can depend only on $L$ and is potentially different from the $C_L$ in (8). $\qquad\square$

## B  TYPICAL STRUCTURE AT INITIALIZATION

In this section, we use the results in Section F.1 to analyze the structure of $\boldsymbol{v}_1, \ldots, \boldsymbol{v}_m$ at initialization. Recall that we initialize $\boldsymbol{v}_i$ with $\mathrm{Unif}(\mathbb{S}^{d-1})$ independently. Meanwhile, note that for $\boldsymbol{v} \sim \mathrm{Unif}(\mathbb{S}^{d-1})$, we have $\boldsymbol{v} \overset{d}{=} \boldsymbol{Z}/\|\boldsymbol{Z}\|$ where $\boldsymbol{Z} \sim \mathcal{N}(0, \boldsymbol{I}_d)$.

We start with a lemma on the largest coordinate. This lemma ensures that $\|\boldsymbol{v}\|_{2L}^{2L}$ is much smaller than the second-order terms at least at initialization.

**Lemma B.1** (Largest coordinate). *Let $\boldsymbol{v} \sim \mathrm{Unif}(\mathbb{S}^{d-1})$. For any $K \geq 1$, we have*

$$\max_{i \in [d]} |v_i| \leq \frac{4\sqrt{2K \log d}}{\sqrt{d}} \quad \text{with probability at least } 1 - \frac{4}{d^K}.$$

*As a corollary, for any $\delta_{\mathbb{P}} \in (0, 1)$, at initialization, we have*

$$\max_{i \in [m]} \|\boldsymbol{v}_i\|_\infty \leq \frac{4\sqrt{2\log(4m/\delta_{\mathbb{P}})}}{\sqrt{d}} \quad \text{with probability at least } 1 - \delta_{\mathbb{P}}.$$

*In particular, this implies that at initialization, at least with the same probability, for any $L \geq 2$,*

$$\max_{i \in [m]} \|\boldsymbol{v}_i\|_{2L}^{2L} \leq d\left(\frac{4\sqrt{2K \log d}}{\sqrt{d}}\right)^{2L} \leq d\left(\frac{32K \log d}{d}\right)^{L}.$$

*Proof.* Let $\boldsymbol{Z} \sim \mathcal{N}(0, \boldsymbol{I}_d)$. Recall that $\boldsymbol{Z}/\|\boldsymbol{Z}\|$ follows the uniform distribution over the sphere. By Lemma F.1 with $s = \sqrt{d}/3$, we have $\|\boldsymbol{Z}\| \geq \sqrt{d}/2$ with probability at least $1 - 2\exp(-d/18)$. Then, by Lemma F.2, with probability at least $1 - 2e^{-d/18} - 2e^{-s^2/2}$, we have

$$\frac{\max_{i \in [d]} |Z_i|}{\|\boldsymbol{Z}\|} \leq \frac{\sqrt{2\log d} + s}{\sqrt{d}/2} = \frac{2\sqrt{2\log d}}{\sqrt{d}} + \frac{2s}{\sqrt{d}}.$$

Let $K \geq 1$ be arbitrary. Choose $s = \sqrt{2K \log d}$ and the above becomes

$$\frac{\max_{i \in [d]} |Z_i|}{\|\boldsymbol{Z}\|} \leq \frac{4\sqrt{2K \log d}}{\sqrt{d}} \quad \text{with probability at least } 1 - \frac{4}{d^K}.$$

For the corollary, use union bound and choose $K = \log(4m/\delta_\mathbb{P})/\log d$, we have

$$\max_{i \in [m]} \|\boldsymbol{v}_i\|_\infty \leq \frac{4\sqrt{2\log(4m/\delta_\mathbb{P})}}{\sqrt{d}} \quad \text{with probability at least } 1 - \frac{4m}{d^K} = 1 - \delta_\mathbb{P}.$$

$\square$

Suppose that we only have higher-order terms. Then, for a neuron $\boldsymbol{v} \in \mathbb{S}^{d-1}$ to converge to a ground-truth direction $\boldsymbol{e}_k$ in a reasonable amount of time, we need $v_k^2$ to be the largest among all $v_i^2$ and there is gap between it and the second largest $v_i^2$. The following lemma ensures that when $m$ is large, for every ground-truth direction $\{\boldsymbol{e}_k\}_{k \in [P]}$, there will be at least one neuron satisfying the above property. Note that in our case, we only need to ensure $v_k^2$ is the largest among all $\{v_i^2\}_{i \in [P]}$ instead of $\{v_i^2\}_{i \in [d]}$, as the second-order term will help us identify the correct subspace.

**Lemma B.2** (Existence of good neurons). *Let $\delta_\mathbb{P} \in (0,1)$ be given and $c \geq 1$ a universal constant. Suppose that the number of neurons $m$ satisfies*

$$m \geq 400c P^{8c^2} \sqrt{\log P} \log\left(P \vee \frac{1}{\delta_\mathbb{P}}\right).$$

*Then, at initialization, with probability at least $1 - \delta_\mathbb{P}$, we have*

$$\forall p \in [P] \, \exists i \in [m] \quad \text{such that} \quad \frac{|v_{i,p}|}{\max_{q \in [P] \setminus \{p\}} |v_{i,q}|} \geq \frac{1 + 2c}{1 + c}.$$

**Remark**. In particular, note that the number of neurons we need is $\text{poly}(P)$ instead of $\text{poly}(d)$. ♣

*Proof.* Let $\boldsymbol{Z} \sim \mathcal{N}(0, \boldsymbol{I}_d)$. Note that $|v_p|/|v_q| \overset{d}{=} |Z_p|/|Z_q|$. Hence, it suffices to consider the largest and the second largest among $\{|Z_i|\}_{i \in [P]}$. Let $|v|_{(1)}$ and $|v|_{(2)}$ denote the largest and second largest among $\{|v_i|\}_{i \in [P]}$. By Lemma F.4 (with $d$ replaced by $P$), for any $c \geq 1$, we have

$$\mathbb{P}\left[\frac{|v|_{(1)}}{|v|_{(2)}} \geq \frac{1+2c}{1+c}\right] \geq \frac{1}{5\pi(1+2c)} \frac{1}{P^{8c^2}\sqrt{\log P}}.$$

Then, for each $p \in [P]$, by symmetry, we have

$$\mathbb{P}\left[\frac{|v_p|}{\max_{q \in [P] \setminus \{p\}} |v_q|} \geq \frac{1+2c}{1+c}\right] \geq \frac{1}{5\pi(1+2c)} \frac{1}{P^{8c^2}\sqrt{\log P}}.$$

Now, define the event $G_p$ as

$$G_p = \left\{\exists i \in [m], \frac{|v_{i,p}|}{\max_{q \in [P] \setminus \{p\}} |v_{i,q}|} \geq \frac{1+2c}{1+c}\right\}.$$

Then, we compute

$$\mathbb{P}[G_p] \geq 1 - \left(\mathbb{P}\left[\frac{|v_p|}{\max_{q \in [P] \setminus \{p\}} |v_q|} < \frac{1+2c}{1+c}\right]\right)^m$$

$$\geq 1 - \left(1 - \frac{1}{5\pi(1+2c)} \frac{1}{P^{8c^2}\sqrt{\log P}}\right)^m$$

$$\geq 1 - \exp\left(-\frac{1}{5\pi(1+2c)} \frac{m}{P^{8c^2}\sqrt{\log P}}\right).$$

By union bound, we have

$$\mathbb{P}\left[\bigwedge_{p=1}^P G_p\right] \geq 1 - \exp\left(\log P - \frac{1}{5\pi(1+2c)} \frac{m}{P^{8c^2}\sqrt{\log P}}\right).$$

Let $\delta_\mathbb{P} \in (0,1)$ be given. Choose

$$m \geq 400c P^{8c^2} \sqrt{\log P} \log\left(P \vee \frac{1}{\delta_\mathbb{P}}\right).$$

Then, the above becomes $\mathbb{P}[\bigwedge_{p=1}^P G_p] \geq 1 - \delta_\mathbb{P}$.

$\square$

**Lemma B.3** (Typical structure at initialization). *Let $\delta_{\mathbb{P}} \in (e^{-\log^C d}, 1)$ be given. Suppose that $\{\boldsymbol{v}_k\}_{k=1}^m \sim \mathrm{Unif}(\mathbb{S}^{d-1})$ independently with*

$$m = 400 P^8 \log^{1.5}(P \vee 1/\delta_{\mathbb{P}}).$$

*Then, with probability at least $1 - 3\delta_{\mathbb{P}}$, we have*

$$\forall p \in [P] \, \exists i \in [m] \quad such \, that \quad \frac{|v_{i,p}|}{\max_{q \in [P] \setminus \{p\}} |v_{i,q}|} \geq \frac{3}{2},$$

$$\forall i \in [m], \quad \|\boldsymbol{v}_i\|_\infty \leq \frac{20\sqrt{\log(P/\delta_{\mathbb{P}})}}{\sqrt{d}},$$

$$\forall i \in [m], \quad \frac{\sqrt{P}}{3\sqrt{d}} \leq \frac{\|\boldsymbol{v}_{\leq P}\|}{\|\boldsymbol{v}\|} \leq \frac{3\sqrt{P}}{\sqrt{d}}.$$

*Proof.* The first two bounds comes directly from Lemma B.1 and Lemma B.2. By Lemma F.1, we have

$$\mathbb{P}\left(|\|\boldsymbol{Z}\| - \mathbb{E}\|\boldsymbol{Z}\|| \geq \sqrt{d}/2\right) \leq 2e^{-d/8},$$

$$\mathbb{P}\left(|\|\boldsymbol{Z}_{\leq P}\| - \mathbb{E}\|\boldsymbol{Z}_{\leq P}\|| \geq \sqrt{P}/2\right) \leq 2e^{-P/8}.$$

As a result, for any $\boldsymbol{v} \sim \mathrm{Unif}(\mathbb{S}^{d-1})$, we have with probability at least $1 - 4e^{-P/8}$ that

$$\frac{\|\boldsymbol{v}_{\leq P}\|}{\|\boldsymbol{v}\|} \stackrel{d}{=} \frac{\|\boldsymbol{Z}_{\leq P}\|}{\|\boldsymbol{Z}\|} = \frac{\mathbb{E}\|\boldsymbol{Z}_{\leq P}\| \pm \sqrt{P}/2}{\mathbb{E}\|\boldsymbol{Z}\| \pm \sqrt{d}/2} = [1/3, 3] \times \sqrt{\frac{P}{d}}.$$

Since we assume $P \geq \log^{C'} d$ for a large $C'$, we have $4e^{-P/8} \leq \delta_{\mathbb{P}}/m$. This gives the third bound. $\qquad\square$

## C  STAGE 1: RECOVERY OF THE SUBSPACE AND DIRECTIONS

In this section, we consider the stage where the second layer is fixed to be a small value and the first layer is trained using online spherical SGD. Let $\boldsymbol{v}$ be an arbitrary first-layer neuron. By Lemma 2.2, we can write its update rule as[5]

$$\hat{\boldsymbol{v}}_{t+1} = \boldsymbol{v}_t + \frac{\eta}{a_0}\left(\tilde{\nabla}_{\boldsymbol{v}}\mathcal{L} + a_0 \boldsymbol{Z}_{t+1}\right), \quad \boldsymbol{v}_{t+1} = \frac{\hat{\boldsymbol{v}}_{t+1}}{\|\hat{\boldsymbol{v}}_{t+1}\|},$$

where $\boldsymbol{Z}_{t+1} = a_0^{-1}(\boldsymbol{I} - \boldsymbol{v}\boldsymbol{v}^\top)(\nabla_{\boldsymbol{v}} l(\boldsymbol{x}) - \nabla_{\boldsymbol{v}}\mathcal{L})$ and

$$-\tilde{\nabla}_{\boldsymbol{v}}\mathcal{L} = -(\boldsymbol{I} - \boldsymbol{v}\boldsymbol{v}^\top)\nabla_{\boldsymbol{v}}\mathcal{L}$$

$$= 2a_0 \sum_{k=1}^P v_k(\boldsymbol{I} - \boldsymbol{v}\boldsymbol{v}^\top)\boldsymbol{e}_k + 2La_0 \sum_{k=1}^P v_k^{2L-1}(\boldsymbol{I} - \boldsymbol{v}\boldsymbol{v}^\top)\boldsymbol{e}_k \pm_2 2Lma_0^2.$$

In particular, for each $k \in [d]$, we have[6]

$$\hat{v}_{t+1,k} = v_{t,k} + \eta\left(\mathbb{1}\{k \leq P\}\left(2 + 2Lv_k^{2L-2}\right) - \rho\right)v_k + \eta Z_{t+1,k} \pm 2\eta Lma_0,$$

where

$$\rho := 2\sum_{i=1}^P v_i^2 + 2L\sum_{i=1}^P v_i^{2L} = 2\|\boldsymbol{v}_{\leq P}\|^2 + 2L\|\boldsymbol{v}_{\leq P}\|_{2L}^{2L}. \tag{9}$$

In addition, we have the following lemma on the dynamics of $v_k^2$. The proof is routine calculation and is deferred to the end of this section.

---

[5]See the remark following Lemma 2.2 for the meaning of an arbitrary first-layer neuron $\boldsymbol{v}$. Also recall that we assume w.l.o.g. that $\boldsymbol{v}_k^* = \boldsymbol{e}_k$.

[6]We will often drop the subscript $t$ when it is clear from the context.

**Lemma C.1** (Dynamics of $v_k^2$). *For any first-layer neuron $\boldsymbol{v}$ and $k \in [d]$, we have*

$$\hat{v}_{t+1,k}^2 = \left(1 + 2\eta \left(\mathbb{1}\{k \leq P\} \left(2 + 2Lv_k^{2L-2}\right) - \rho\right)\right) v_k^2 + 2\eta v_k Z_k$$
$$\pm 300L^3 \eta m a_0 \pm 300L^3 \eta^2 \left(1 \vee Z_k^2\right).$$

To proceed, we split Stage 1 into two substages. In Stage 1.1, we rely on the second-order terms to learn the relevant subspace. We will also show that the gap between largest and second-largest coordinates, which can be guaranteed with certain probability at initialization, is preserved throughout Stage 1.1. These give Stage 1.2 a nice starting point. Then, we show that in Stage 1.2, online spherical SGD can recover the directions using the $2L$-th order terms.

## C.1 STAGE 1.1: RECOVERY OF THE SUBSPACE AND PRESERVATION OF THE GAP

In this subsection, first we show that the ratio $\|\boldsymbol{v}_{\leq P}\|^2 / \|\boldsymbol{v}_{>P}\|^2$ will grow from $\Omega(P/d)$ to $\Theta(1)$ within $\tilde{O}(dP)$ iterations and during this phase. We will rely on the second-order terms and bound the influence of higher-order terms. This leads to the desired complexity. The next goal to show the initial randomness is preserved. In our case, we only to the gap between the largest and the second-largest coordinate to be preserved. This ensures that the neurons will not collapse to one single direction. Formally, we have the following lemma.

**Lemma C.2** (Stage 1.1). *Let $\boldsymbol{v} \in \mathbb{S}^{d-1}$ be an arbitrary first-layer neuron satisfying $\|\boldsymbol{v}\|_\infty \leq \log^2 d / (2d)$ and $\|\boldsymbol{v}_{\leq P}\|^2 / \|\boldsymbol{v}_{>P}\|^2 \geq 0.1P/d$ at initialization. Let $\delta_\mathbb{P} \in (e^{-\log^C d}, 1)$ be given. Suppose that we choose*

$$ma_0 \lesssim_L \frac{1}{d \log^3 d} \quad and \quad \eta \lesssim_L \frac{\delta_\mathbb{P}}{dP^2 \log^{4L+1}(d/\delta_\mathbb{P})} = \tilde{\Theta}_L \left(\frac{\delta_\mathbb{P}}{dP^2}\right).$$

*Then, with probability at least $1 - O(\delta_\mathbb{P})$, we have*

$$\frac{\|\boldsymbol{v}_{\leq P}\|^2}{\|\boldsymbol{v}_{<P}\|^2} \geq 1 \quad within \ T = \frac{1 + o(1)}{4\eta} \log \left(\frac{d}{P}\right) = \tilde{\Theta}(dP^2) \ iterations.$$

*Moreover, if at initialization, $v_p^2$ is the largest among $\{v_k^2\}_{k \in [P]}$ and is $1.5$ times larger than the second-largest $\{v_k^2\}_{k \in [P]}$, then at the end of Stage 1.1, it is still $1.25$ times larger than the second-largest $\{v_k^2\}_{k \in [P]}$.*

**Remark.** To make the above result hold uniformly over all $m = \text{poly}(P)$ neurons, it suffices to replace $\delta_\mathbb{P}$ with $\delta_\mathbb{P}/m$. In addition, by Lemma B.3, the hypotheses of this lemma hold with high probability at initialization. ♣

*Proof.* It suffices to combine Lemma C.4, Lemma C.5 and Lemma C.6. ☐

To prove this lemma, we will use stochastic induction (cf. Section F.2), in particular, Lemma F.6, Lemma F.8, and Lemma F.10. For example, to analyze the dynamics of $\|\boldsymbol{v}_{\leq P}\|^2 / \|\boldsymbol{v}_{>P}\|^2$, it suffices to write down the update rule of $\|\boldsymbol{v}_{\leq P}\|^2 / \|\boldsymbol{v}_{>P}\|^2$ and decompose it into a signal growth term, a higher-order error term, and a martingale difference term as in Lemma F.6. Then, we bound the higher-order error terms, and estimate the covariance of the martingale difference terms, assuming the induction hypotheses.

The induction hypotheses we will maintain in this substage are the following:

$$\frac{\|\boldsymbol{v}_{t,\leq P}\|^2}{\|\boldsymbol{v}_{t,>P}\|^2} = \Theta(1)(1 + 4\eta)^t \frac{\|\boldsymbol{v}_{0,\leq P}\|^2}{\|\boldsymbol{v}_{0,>P}\|^2}, \quad v_p^2 \leq \frac{\log^2 d}{P}.$$

They are established in Lemma C.4, Lemma C.9 and Lemma C.8.

### C.1.1 LEARNING THE SUBSPACE

Now, we derive formulas for the dynamics of the ratio $\|\boldsymbol{v}_{\leq P}\|^2 / \|\boldsymbol{v}_{>P}\|^2$. Since we will use Lemma F.6 to analyze it, the goal here is separate the signal terms, martingale difference terms, and higher-order error terms.

**Lemma C.3** (Dynamics of the norm ratio). *Assume the induction hypotheses. Let $\boldsymbol{v}$ be an arbitrary first-layer neuron. For any $t \leq T$, we have*

$$\frac{\|\boldsymbol{v}_{t+1,\leq P}\|^2}{\|\boldsymbol{v}_{t+1,>P}\|^2} = \frac{\|\boldsymbol{v}_{\leq P}\|^2}{\|\boldsymbol{v}_{>P}\|^2} (1 + 4\eta + \varepsilon_v) + \xi_{t+1}$$

$$- \frac{(1 + 4\eta - 2\eta\rho + \varepsilon_v)\|\boldsymbol{v}_{\leq P}\|^2}{(1 - 2\eta\rho)\|\boldsymbol{v}_{>P}\|^2} \frac{2\eta \langle \boldsymbol{v}_{>P}, \boldsymbol{Z}_{>P} \rangle}{(1 - 2\eta\rho)\|\boldsymbol{v}_{>P}\|^2} + \frac{2\eta \langle \boldsymbol{v}_{\leq P}, \boldsymbol{Z}_{\leq P} \rangle}{(1 - 2\eta\rho)\|\boldsymbol{v}_{>P}\|^2},$$

*where $\varepsilon_v := 4L\eta \|\boldsymbol{v}_{\leq P}\|_{2L}^{2L} / \|\boldsymbol{v}_{\leq P}\|^2$ and for any $\delta_{\mathbb{P}} \in (0,1)$, we have with probability at least $1 - \delta_{\mathbb{P}}$, that*

$$|\xi_{t+1}| \leq C_L (1 + 4\eta)^t \eta P \left( ma_0 \vee \eta P^3 \log^{4L} \left( \frac{1}{\delta_{\mathbb{P}}} \right) \right),$$

*where $C_L > 0$ is a constant that can depend on $L$.*

*Proof.* Recall from Lemma C.1 that

$$\hat{v}_{t+1,k}^2 = \left( 1 + 2\eta \left( \mathbb{1}\{k \leq P\} \left( 2 + 2Lv_k^{2L-2} \right) - \rho \right) \right) v_k^2 + 2\eta v_k Z_k$$

$$\pm 300L^3 \eta ma_0 \pm 300L^3 \eta^2 \left( 1 \vee Z_k^2 \right).$$

Hence, for the norms, we have (the higher order terms are changed; additional $P, d$ factors)

$$\|\hat{\boldsymbol{v}}_{\leq P}\|^2 = (1 + 2\eta (2 - \rho)) \|\boldsymbol{v}_{\leq P}\|^2 + 4L\eta \|\boldsymbol{v}_{\leq P}\|_{2L}^{2L} + 2\eta \langle \boldsymbol{v}_{\leq P}, \boldsymbol{Z}_{\leq P} \rangle$$

$$\underbrace{\pm 300L^3 P\eta ma_0 \pm 300L^3 \eta^2 \left( P \vee \|\boldsymbol{Z}_{\leq P}\|^2 \right)}_{=: \, \xi_{\leq P, t}},$$

$$\hat{v}_{t+1,k}^2 = (1 - 2\eta\rho) \|\boldsymbol{v}_{>P}\|^2 + 2\eta \langle \boldsymbol{v}_{>P}, \boldsymbol{Z}_{>P} \rangle$$

$$\underbrace{\pm 300L^3 d\eta ma_0 \pm 300L^3 \eta^2 \left( d \vee \|\boldsymbol{Z}_{\geq P}^2\| \right)}_{=: \, \xi_{>P, t}}.$$

For notational simplicity, put $\varepsilon_v = 4L\eta \|\boldsymbol{v}_{\leq P}\|_{2L}^{2L} / \|\boldsymbol{v}_{\leq P}\|^2$. Note that $\|\boldsymbol{v}_{\leq P}\| / \|\boldsymbol{v}_{>P}\| = \|\hat{\boldsymbol{v}}_{\leq P}\| / \|\hat{\boldsymbol{v}}_{>P}\|$. Thus, we have

$$\frac{\|\boldsymbol{v}_{t+1,\leq P}\|^2}{\|\boldsymbol{v}_{t+1,>P}\|^2} = \frac{(1 + 2\eta (2 - \rho) + \varepsilon_v) \|\boldsymbol{v}_{\leq P}\|^2 + 2\eta \langle \boldsymbol{v}_{\leq P}, \boldsymbol{Z}_{\leq P} \rangle + \xi_{\leq P}}{(1 - 2\eta\rho) \|\boldsymbol{v}_{>P}\|^2 + 2\eta \langle \boldsymbol{v}_{>P}, \boldsymbol{Z}_{>P} \rangle + \xi_{>P}}$$

$$= \frac{(1 + 4\eta - 2\eta\rho + \varepsilon_v) \|\boldsymbol{v}_{\leq P}\|^2}{(1 - 2\eta\rho) \|\boldsymbol{v}_{>P}\|^2} \left( 1 - \frac{2\eta \langle \boldsymbol{v}_{>P}, \boldsymbol{Z}_{>P} \rangle}{\|\hat{\boldsymbol{v}}_{t+1,>P}\|^2} - \frac{\xi_{>P}}{\|\hat{\boldsymbol{v}}_{t+1,>P}\|^2} \right)$$

$$+ \frac{2\eta \langle \boldsymbol{v}_{\leq P}, \boldsymbol{Z}_{\leq P} \rangle}{\|\hat{\boldsymbol{v}}_{t+1,>P}\|^2} + \frac{\xi_{\leq P}}{\|\hat{\boldsymbol{v}}_{t+1,>P}\|^2}$$

$$= \frac{(1 + 4\eta - 2\eta\rho + \varepsilon_v) \|\boldsymbol{v}_{\leq P}\|^2}{(1 - 2\eta\rho) \|\boldsymbol{v}_{>P}\|^2}$$

$$- \frac{(1 + 4\eta - 2\eta\rho + \varepsilon_v) \|\boldsymbol{v}_{\leq P}\|^2}{(1 - 2\eta\rho) \|\boldsymbol{v}_{>P}\|^2} \frac{2\eta \langle \boldsymbol{v}_{>P}, \boldsymbol{Z}_{>P} \rangle}{\|\hat{\boldsymbol{v}}_{t+1,>P}\|^2} + \frac{2\eta \langle \boldsymbol{v}_{\leq P}, \boldsymbol{Z}_{\leq P} \rangle}{\|\hat{\boldsymbol{v}}_{t+1,>P}\|^2}$$

$$- \frac{(1 + 4\eta - 2\eta\rho + \varepsilon_v) \|\boldsymbol{v}_{\leq P}\|^2}{(1 - 2\eta\rho) \|\boldsymbol{v}_{>P}\|^2} \frac{\xi_{>P}}{\|\hat{\boldsymbol{v}}_{t+1,>P}\|^2} + \frac{\xi_{\leq P}}{\|\hat{\boldsymbol{v}}_{t+1,>P}\|^2}.$$

Note that up to some higher-order terms, the first line contains the signal terms and the second line contains the martingale difference terms. Now, our goal is to factor out those higher-order terms. For the first line, first recall from (13) that $\rho \leq 4L$, and then we use the fact that

$$\frac{1}{1+z} = 1 - z \pm 2z^2, \quad \forall |z| \leq 1/2, \tag{10}$$

to obtain

$$
\frac{(1 + 4\eta - 2\eta\rho + \varepsilon_v)\|\boldsymbol{v}_{\leq P}\|^2}{(1 - 2\eta\rho)\|\boldsymbol{v}_{>P}\|^2} = \frac{\|\boldsymbol{v}_{\leq P}\|^2}{\|\boldsymbol{v}_{>P}\|^2}(1 + 4\eta - 2\eta\rho + \varepsilon_v)(1 + 2\eta\rho \pm 64L^2\eta^2)
$$

$$
= \frac{\|\boldsymbol{v}_{\leq P}\|^2}{\|\boldsymbol{v}_{>P}\|^2}(1 + 4\eta + \varepsilon_v \pm 2000L^3\eta^2).
$$

Similarly, for the second line, we write

$$
\frac{1}{\|\hat{\boldsymbol{v}}_{t+1,>P}\|^2} = \frac{1}{(1 - 2\eta\rho)\|\boldsymbol{v}_{>P}\|^2}\left(1 - \frac{2\eta\langle\boldsymbol{v}_{>P}, \boldsymbol{Z}_{>P}\rangle + \xi_{>P}}{\|\hat{\boldsymbol{v}}_{t+1,>P}\|^2}\right).
$$

By the tail bounds in Lemma 2.2 and the union bound, for any $\delta_{\mathbb{P}} \in (0, 1)$, we have

$$
|\langle\overline{\boldsymbol{v}_{>P}}, \boldsymbol{Z}_{>P}\rangle| \leq C_L^{2L} P \log^{2L}\left(\frac{C_L}{\delta_{\mathbb{P}}}\right), \quad |\langle\overline{\boldsymbol{v}_{\leq P}}, \boldsymbol{Z}_{\leq P}\rangle| \leq C_L^{2L} P \log^{2L}\left(\frac{C_L}{\delta_{\mathbb{P}}}\right),
$$

$$
|Z_k| \leq C_L^{2L} P \log^{2L}\left(\frac{C_L d}{\delta_{\mathbb{P}}}\right), \quad \forall k \in [d],
$$

with probability at least $1 - 2\delta_{\mathbb{P}}$. In particular, note that the second bound also implies, with at least the same probability, we have

$$
|\xi_{\leq P}| \leq 600L^3\eta P\left(ma_0 \vee \eta C_L^{4L} P^2 \log^{4L}\left(\frac{C_L d}{\delta_{\mathbb{P}}}\right)\right),
$$

$$
|\xi_{>P}| \leq 600L^3\eta d\left(ma_0 \vee \eta C_L^{4L} P^2 \log^{4L}\left(\frac{C_L d}{\delta_{\mathbb{P}}}\right)\right).
$$

By our definition of Stage 1.1, we have $\|\hat{\boldsymbol{v}}_{t+1,>P}\|^2 \geq 1/2$. Therefore, with probability at least $1 - 2\delta_{\mathbb{P}}$, we have

$$
\frac{1}{\|\hat{\boldsymbol{v}}_{t+1,>P}\|^2} = \frac{1}{(1 - 2\eta\rho)\|\boldsymbol{v}_{>P}\|^2}\left(1 \pm C_L'\eta P \log^{2L}\left(\frac{1}{\delta_{\mathbb{P}}}\right)\right),
$$

for some constant $C_L' > 0$ that can depend on $L$. Thus, for the ratio of the norms, we have

$$
\frac{\|\boldsymbol{v}_{t+1,\leq P}\|^2}{\|\boldsymbol{v}_{t+1,>P}\|^2} = \frac{\|\boldsymbol{v}_{\leq P}\|^2}{\|\boldsymbol{v}_{>P}\|^2}(1 + 4\eta + \varepsilon_v \pm 2000L^3\eta^2)
$$

$$
- \frac{(1 + 4\eta - 2\eta\rho + \varepsilon_v)\|\boldsymbol{v}_{\leq P}\|^2}{(1 - 2\eta\rho)\|\boldsymbol{v}_{>P}\|^2}\frac{2\eta\langle\boldsymbol{v}_{>P}, \boldsymbol{Z}_{>P}\rangle}{(1 - 2\eta\rho)\|\boldsymbol{v}_{>P}\|^2}\left(1 \pm C_L'\eta P \log^{2L}\left(\frac{1}{\delta_{\mathbb{P}}}\right)\right)
$$

$$
+ \frac{2\eta\langle\boldsymbol{v}_{\leq P}, \boldsymbol{Z}_{\leq P}\rangle}{(1 - 2\eta\rho)\|\boldsymbol{v}_{>P}\|^2}\left(1 \pm C_L'\eta P \log^{2L}\left(\frac{1}{\delta_{\mathbb{P}}}\right)\right)
$$

$$
- \frac{(1 + 4\eta - 2\eta\rho + \varepsilon_v)\|\boldsymbol{v}_{\leq P}\|^2}{(1 - 2\eta\rho)\|\boldsymbol{v}_{>P}\|^2}\frac{\xi_{>P}}{\|\hat{\boldsymbol{v}}_{t+1,>P}\|^2} + \frac{\xi_{\leq P}}{\|\hat{\boldsymbol{v}}_{t+1,>P}\|^2}.
$$

Collect the higher-order terms into $\xi_{t+1}$, so that the above becomes

$$
\frac{\|\boldsymbol{v}_{t+1,\leq P}\|^2}{\|\boldsymbol{v}_{t+1,>P}\|^2} = \frac{\|\boldsymbol{v}_{\leq P}\|^2}{\|\boldsymbol{v}_{>P}\|^2}(1 + 4\eta + \varepsilon_v) + \xi_{t+1}
$$

$$
- \frac{(1 + 4\eta - 2\eta\rho + \varepsilon_v)\|\boldsymbol{v}_{\leq P}\|^2}{(1 - 2\eta\rho)\|\boldsymbol{v}_{>P}\|^2}\frac{2\eta\langle\boldsymbol{v}_{>P}, \boldsymbol{Z}_{>P}\rangle}{(1 - 2\eta\rho)\|\boldsymbol{v}_{>P}\|^2} + \frac{2\eta\langle\boldsymbol{v}_{\leq P}, \boldsymbol{Z}_{\leq P}\rangle}{(1 - 2\eta\rho)\|\boldsymbol{v}_{>P}\|^2}.
$$

For the higher-order terms, we have with probability at least $1 - O(\delta_{\mathbb{P}})$

$$
|\xi_{t+1}| \lesssim_L \frac{\|\boldsymbol{v}_{\leq P}\|^2}{\|\boldsymbol{v}_{>P}\|^2}\eta^2 + \frac{\|\boldsymbol{v}_{\leq P}\|^2}{\|\boldsymbol{v}_{>P}\|^2}\frac{\eta|\langle\boldsymbol{v}_{>P}, \boldsymbol{Z}_{>P}\rangle|}{\|\boldsymbol{v}_{>P}\|^2}\eta P\log^{2L}\left(\frac{1}{\delta_{\mathbb{P}}}\right)
$$

$$
+ \frac{\eta|\langle\boldsymbol{v}_{\leq P}, \boldsymbol{Z}_{\leq P}\rangle|}{\|\boldsymbol{v}_{>P}\|^2}\eta P\log^{2L}\left(\frac{1}{\delta_{\mathbb{P}}}\right) + \frac{\|\boldsymbol{v}_{\leq P}\|^2}{\|\boldsymbol{v}_{>P}\|^2}\frac{|\xi_{>P}|}{\|\boldsymbol{v}_{>P}\|^2} + \frac{|\xi_{\leq P}|}{\|\boldsymbol{v}_{>P}\|^2}
$$

$$
\lesssim_L \frac{\|\boldsymbol{v}_{\leq P}\|^2}{\|\boldsymbol{v}_{>P}\|^2}\eta^2 + \left(\frac{\|\boldsymbol{v}_{\leq P}\|^2}{\|\boldsymbol{v}_{>P}\|^3} + \frac{\|\boldsymbol{v}_{\leq P}\|}{\|\boldsymbol{v}_{>P}\|^2}\right)\eta^2 P^2\log^{4L}\left(\frac{1}{\delta_{\mathbb{P}}}\right)
$$

$$
+ \left(\frac{d\|\boldsymbol{v}_{\leq P}\|^2}{\|\boldsymbol{v}_{>P}\|^4} + \frac{P}{\|\boldsymbol{v}_{>P}\|^2}\right)\eta\left(ma_0 \vee \eta P^2\log^{4L}\left(\frac{d}{\delta_{\mathbb{P}}}\right)\right)
$$

$$
\lesssim_L (1 + 4\eta)^t\eta P\left(ma_0 \vee \eta P^2\log^{4L}\left(\frac{d}{\delta_{\mathbb{P}}}\right)\right),
$$

where we use the induction hypothesis $\|\boldsymbol{v}_{\leq P}\|^2 / \|\boldsymbol{v}_{>P}\|^2 = \Theta\left((1 + 4\eta)^t P/d\right)$ to handle the $d\|\boldsymbol{v}_{\leq P}\|^2 / \|\boldsymbol{v}_{>P}\|^4$ factor in the last line. $\qquad\square$

With the above formula, we can now use Lemma F.6 to analyze the dynamics of ratio of the norms.

**Lemma C.4** (Learning the subspace). *Let $\boldsymbol{v}$ be an arbitrary fixed first-layer neuron. Suppose that*

$$
ma_0 \lesssim_L \frac{1}{d\log d} \quad \text{and} \quad \eta \lesssim_L \frac{\delta_{\mathbb{P}}}{dP^2\log^{4L+1}(d/\delta_{\mathbb{P}})} = \tilde{\Theta}_L\left(\frac{\delta_{\mathbb{P}}}{dP^2}\right),
$$

*Then, throughout Stage 1.1, we have*

$$
\frac{(1 + 4\eta)^t}{2}\frac{\|\boldsymbol{v}_{0, \leq P}\|^2}{\|\boldsymbol{v}_{0, >P}\|^2} \leq \frac{\|\boldsymbol{v}_{\leq P}\|^2}{\|\boldsymbol{v}_{>P}\|^2} \leq \frac{3(1 + 4\eta)^t}{2}\frac{\|\boldsymbol{v}_{0, \leq P}\|^2}{\|\boldsymbol{v}_{0, >P}\|^2},
$$

*and Stage 1.1 takes at most $(1 + o(1))(4\eta)^{-1}\log(d/P) = \tilde{O}_L\left(dP^2/\delta_{\mathbb{P}}\right)$ iterations. To obtain estimates that uniformly hold for all neurons, it suffices to replace $\delta_{\mathbb{P}}$ with $\delta_{\mathbb{P}}/m$.*

*Proof.* By Lemma C.3, we have

$$
\frac{\|\boldsymbol{v}_{t+1, \leq P}\|^2}{\|\boldsymbol{v}_{t+1, >P}\|^2} = \frac{\|\boldsymbol{v}_{\leq P}\|^2}{\|\boldsymbol{v}_{>P}\|^2}(1 + 4\eta + \varepsilon_v) + \xi_{t+1}
$$

$$
\underbrace{-\frac{(1 + 4\eta - 2\eta\rho + \varepsilon_v)\|\boldsymbol{v}_{\leq P}\|^2}{(1 - 2\eta\rho)\|\boldsymbol{v}_{>P}\|^2}\frac{2\eta\langle\boldsymbol{v}_{>P}, \boldsymbol{Z}_{>P}\rangle}{(1 - 2\eta\rho)\|\boldsymbol{v}_{>P}\|^2}}_{=: H_{t+1}^{(1)}} + \underbrace{\frac{2\eta\langle\boldsymbol{v}_{\leq P}, \boldsymbol{Z}_{\leq P}\rangle}{(1 - 2\eta\rho)\|\boldsymbol{v}_{>P}\|^2}}_{=: H_{t+1}^{(2)}},
$$

where $\varepsilon_v := 4L\eta\|\boldsymbol{v}_{\leq P}\|_{2L}^{2L} / \|\boldsymbol{v}_{\leq P}\|^2$ and for any $\delta_{\mathbb{P}} \in (0, 1)$, we have with probability at least $1 - \delta_{\mathbb{P}}/T$, that

$$
|\xi_{t+1}| \leq C_L(1 + 4\eta)^t\eta P\left(ma_0 \vee \eta P^2\log^{4L}\left(\frac{T}{\delta_{\mathbb{P}}}\right)\right),
$$

where $C_L > 0$ is a constant that can depend on $L$. By our induction hypothesis $v_p^2 \leq \log^2 d/P$, we

$$
\varepsilon_v = \frac{4L\eta}{\|\boldsymbol{v}_{\leq P}\|^2}\sum_{p=1}^P v_p^{2L} \leq \frac{4L\eta}{\|\boldsymbol{v}_{\leq P}\|^2}\|\boldsymbol{v}_{\leq P}\|_{\infty}^{2L-2}\sum_{p=1}^P v_p^2 \leq \eta\frac{4L\log^{2L-2}(d)}{P^{L-1}} =: \eta\delta_v.
$$

In particular, note that $\delta_v$ does not depend on $t$ and is $o(1)$. For the martingale difference terms, by Lemma 2.2, we have

$$
\mathbb{E}\left[(H_{t+1}^{(1)})^2 \mid \mathcal{F}_t\right] \lesssim_L \eta^2\frac{\|\boldsymbol{v}_{\leq P}\|^4}{\|\boldsymbol{v}_{>P}\|^6}\mathbb{E}\left[\langle\overline{\boldsymbol{v}_{>P}}, \boldsymbol{Z}_{>P}\rangle^2 \mid \mathcal{F}_t\right] \lesssim_L \eta^2 P^2\frac{\|\boldsymbol{v}_{\leq P}\|^4}{\|\boldsymbol{v}_{>P}\|^4},
$$

$$
\mathbb{E}\left[(H_{t+1}^{(2)})^2 \mid \mathcal{F}_t\right] \lesssim_L \eta^2\frac{\|\boldsymbol{v}_{\leq P}\|^2}{\|\boldsymbol{v}_{>P}\|^4}\mathbb{E}\left[\langle\overline{\boldsymbol{v}_{\leq P}}, \boldsymbol{Z}_{\leq P}\rangle^2 \mid \mathcal{F}_t\right] \lesssim_L \eta^2 P^2\frac{\|\boldsymbol{v}_{\leq P}\|^2}{\|\boldsymbol{v}_{>P}\|^2}.
$$

Put $H_{t+1} := H_{t+1}^{(1)} + H_{t+1}^{(2)}$. The above bounds imply that

$$
\mathbb{E}\left[H_{t+1}^2 \mid \mathcal{F}_t\right] \lesssim_L \eta^2 P^2 \frac{\|\boldsymbol{v}_{\leq P}\|^2}{\|\boldsymbol{v}_{>P}\|^2} \lesssim_L \eta^2 P^2 (1+4\eta)^t \frac{\|\boldsymbol{v}_{0,\leq P}\|^2}{\|\boldsymbol{v}_{0,>P}\|^2} \lesssim_L \frac{\eta^2 P^3}{d}(1+4\eta)^t
$$

where the second inequality comes from our induction hypothesis.

For notational simplicity, put $X_t := \|\boldsymbol{v}_{\leq P}\|^2 / \|\boldsymbol{v}_{>P}\|^2$, $x_t^- = (1+4\eta)^t X_0$ and $x_t^+ = (1+4\eta(1+\delta_v))^t X_0$. $x^{\pm}$ will serve as the lower and upper bounds for the deterministic counterpart of $X$, since

$$
(1+4\eta)X_t + \xi_{t+1} + H_{t+1} \leq X_{t+1} \leq (1+4\eta(1+\delta_v))X_t + \xi_{t+1} + H_{t+1}.
$$

Moreover, note that for any $t \leq T$, we have

$$
\frac{x_t^+}{x_t^-} = \left(\frac{1+4\eta(1+\delta_v)}{1+4\eta}\right)^t = \left((1+4\eta(1+\delta_v))\left(1-4\eta \pm 16\eta^2\right)\right)^t
$$

$$
\leq \left(1+4\eta\delta_v \pm 40\eta^2\right)^t
$$

$$
\leq \exp\left(40\eta T\left(\delta_v + \eta\right)\right).
$$

Since $T \leq \log d/\eta$, the above implies

$$
1 \leq \frac{x_t^+}{x_t^-} \leq \exp\left(40\log d\left(\delta_v + \eta\right)\right) \leq 1 + 80\log d\left(\delta_v + \eta\right) = 1 + o(1),
$$

where the last (approximate) identity holds whenever

$$
\delta_v \ll \frac{1}{\log d} \quad \Leftarrow \quad \frac{4L\log^{2L-2}(d)}{P^{L-1}} \ll \frac{1}{\log d} \quad \Leftarrow \quad P \gg (4L)^{1/(L-1)}\log^2 d.
$$

In particular, this implies that the (multiplicative) difference between $x_t^+$ and $x_t^-$ is small.

Now, we apply Lemma F.6 to $X_t$. In our case, we have

$$
\Xi \lesssim_L \eta P\left(ma_0 \vee \eta P^2 \log^{4L}\left(\frac{T}{\delta_{\mathbb{P}}}\right)\right), \quad \sigma_Z^2 \lesssim_L \frac{\eta^2 P^3}{d},
$$

$\alpha = 4(1+o(1))\eta$ and $X_0 = \Theta(P/d)$. Recall that $T \leq O(\log d/\eta)$. Hence, to meet the conditions of Lemma F.6, it suffices to choose

$$
\eta P\left(ma_0 \vee \eta P^2 \log^{4L}\left(\frac{T}{\delta_{\mathbb{P}}}\right)\right) \lesssim_L \frac{X_0}{T} \quad \Leftarrow \quad
\begin{cases}
ma_0 \lesssim_L \dfrac{1}{d\log d}, \\[2mm]
\eta \lesssim_L \dfrac{1}{dP^2 \log^{4L}\left(T/\delta_{\mathbb{P}}\right)\log d}
\end{cases}
$$

$$
\frac{\eta^2 P^3}{d} \lesssim_L \frac{\delta_{\mathbb{P}}\alpha X_0^2}{16} \quad \Leftarrow \quad \eta \lesssim_L \frac{\delta_{\mathbb{P}}}{dP}.
$$

To satisfy the above conditions, it suffices to choose

$$
ma_0 \lesssim_L \frac{1}{d\log d} \quad \text{and} \quad \eta \lesssim_L \frac{\delta_{\mathbb{P}}}{dP^2 \log^{4L+1}\left(d/\delta_{\mathbb{P}}\right)}.
$$

Then, by Lemma F.6, we have, with probability at least $1 - \Theta(\delta_{\mathbb{P}})$, $0.5x_t^- \leq X_t \leq 1.5x_t^+$. Since $x_t^+ = (1+o(1))x_t^-$, this implies $0.5x_t \leq X_t \leq 2x_t$. To complete the proof, it suffices to note that for $x_t$ to grow from $\Theta(P/d)$ to $1$, the number of iterations needed is bounded by $(1+o(1))(4\eta)^{-1}\log\left(d/P\right)$. $\square$

### C.1.2 PRESERVATION OF THE GAP

Now, we show that the gap between the largest coordinate and the second-largest coordinate can be preserved in Stage 1.1. Let $p = \operatorname{argmax}_{i\in[P]} v_i^2(0)$ and consider the ratio $v_p^2/v_q^2$, where $q \in [P]$ is arbitrary. The proof is conceptually very similar to the previous one, except that we will use Lemma F.8 instead of Lemma F.6. However, there is still some technical subtlety that is not involved

in the previous analysis. When $v_q^2$ is close to 0, the dynamics of $v_p^2/v_q^2$ can be unstable, violating the conditions of Lemma F.8. Intuitively, this should not cause any fundamental issue, since we are only interested in the square of largest and second-largest coordinates, both of which should be at least $\Omega(1/d)$ throughout Stage 1.1. To handle this technical issue, we will partition $q \in [P]$ based on the initial value $v_{0,q}^2$. When $v_{0,q}^2 = \Omega(1/d)$, we consider the dynamics of the ratio $v_p^2/v_q^2$ directly. If $v_{0,q}^2$ is small, we will use Lemma C.7 and Lemma C.8, and bound the ratio in a more direct way.

**Lemma C.5** (Gap between large and small coordinates). *Consider $p, q \in [P]$. There exists a universal constant $c_v > 0$ such that if $v_{0,p}^2 \geq 1/d$ and $v_{0,q}^2 \leq c_v/d$, and we choose the hyperparameters according to Lemma C.7 and Lemma C.8, then we have with probability at least $1 - O(\delta_{\mathbb{P}})$, that $v_p^2 \geq 2v_q^2$ throughout Stage 1.1.*

*Proof.* By Lemma C.7, we have

$$v_{t,p}^2 \geq \frac{1}{2}(1 + 4\eta)^t v_{0,p}^2 \geq \frac{1}{2}(1 + 4\eta)^t \frac{1}{d},$$

with probability at least $1 - O(\delta_{\mathbb{P}})$. Meanwhile, by Lemma C.8, we have

$$v_{t,q}^2 \leq 2C(1 + 4\eta)^t \frac{c_v}{d},$$

with probability at least $1 - O(\delta_{\mathbb{P}})$. Hence, as long as $c_v \leq 1/(8C)$, we have $v_{t,q}^2 \leq v_{t,p}^2/2$ throughout Stage 1 with probability at least $1 - O(\delta_{\mathbb{P}})$. $\square$

**Lemma C.6** (Gap between large coordinates). *Consider $p, q \in [P]$ and let $c_v > 0$ be the universal constant in the previous lemma. Suppose that $v_{0,p}^2 \geq v_{0,q}^2 \geq c_v/d$. Let $\varepsilon_R \in (0, 1)$ be given. Suppose that the hyperparameters satisfy the conditions in Lemma C.7 and*

$$ma_0 \lesssim_L \frac{\varepsilon_R}{d \log^3 d}, \quad P \gtrsim_L \frac{\log^3 d}{\varepsilon_R}, \quad \eta \lesssim_L \frac{\varepsilon_R \sqrt{\delta_{\mathbb{P}}}}{dP^2 \log^{2L+2}(d/\delta_{\mathbb{P}})}.$$

*Then, we have $\left| v_p^2/v_q^2 - v_{0,p}^2/v_{0,q}^2 \right| \leq \varepsilon_R$ throughout Stage 1.1 with probability at least $1 - \Theta(\delta_{\mathbb{P}})$.*

*Proof.* First, note that by Lemma C.7, we have $v_{t,q}^2 \geq c_v/(2d)$ throughout Stage 1.1 with probability at least $1 - O(\delta_{\mathbb{P}})$. Recall from Lemma C.1 that for any $k \leq P$, we have

$$\hat{v}_{t+1,k}^2 = \left(1 + 2\eta\left(2Lv_k^{2L-2} + 2 - \rho\right)\right)v_k^2 + 2\eta v_k Z_k \underbrace{\pm 300L^3 \eta ma_0 \pm 300L^3 \eta^2 \left(1 \vee Z_k^2\right)}_{=: \xi_k}.$$

Hence, for any $p, q \in [P]$, we have

$$\begin{aligned}
\frac{v_{p,t+1}^2}{v_{q,t+1}^2} &= \frac{\left(1 + 2\eta\left(2Lv_p^{2L-2} + 2 - \rho\right)\right)v_p^2 + 2\eta v_p Z_p + \xi_p}{\left(1 + 2\eta\left(2Lv_q^{2L-2} + 2 - \rho\right)\right)v_q^2 + 2\eta v_q Z_q + \xi_q} \\
&= \frac{v_p^2}{v_q^2} - \frac{v_p^2}{v_q^2}\frac{2\eta v_q Z_q}{\left(1 + 2\eta\left(2 - \rho\right)\right)v_q^2 + 4L\eta v_q^{2L}} + \frac{2\eta v_p Z_p}{\left(1 + 2\eta\left(2Lv_q^{2L-2} + 2 - \rho\right)\right)v_q^2} \\
&\quad - \frac{2\eta v_p Z_p}{\left(1 + 2\eta\left(2Lv_q^{2L-2} + 2 - \rho\right)\right)v_q^2}\frac{2\eta v_q Z_q + \xi_q}{\hat{v}_{q,t+1}^2} \\
&\quad + \frac{v_p^2}{v_q^2}\frac{2\eta v_q Z_q}{\left(1 + 2\eta\left(2 - \rho\right)\right)v_q^2 + 4L\eta v_q^{2L}}\frac{2\eta v_q Z_q + \xi_q}{\hat{v}_{q,t+1}^2} \\
&\quad + \frac{\xi_p + 4L\eta v_p^{2L}}{\hat{v}_{q,t+1}^2} + \frac{v_p^2}{v_q^2}\frac{4L\eta v_q^{2L} + \xi_q}{\hat{v}_{q,t+1}^2}.
\end{aligned}$$

The first line contains the signal term and the martingale difference terms. The other three lines contain the higher-order error terms. First, for the martingale difference terms, by our induction

hypotheses and the variance bound in Lemma 2.2, we have

$$\mathbb{E}\left[\left(\frac{v_p^2}{v_q^2}\frac{2\eta v_q Z_q}{(1+2\eta(2-\rho))v_q^2+4L\eta v_q^{2L}}\right)^2 \;\middle|\; \mathcal{F}_t\right] \lesssim_L \eta^2 P^2 \frac{v_p^4}{v_q^6} \lesssim_L \eta^2 dP^2 \log^4 d,$$

$$\mathbb{E}\left[\left(\frac{2\eta v_p Z_p}{(1+2\eta(2Lv_q^{2L-2}+2-\rho))v_q^2}\right)^2 \;\middle|\; \mathcal{F}_t\right] \lesssim_L \eta^2 P^2 \frac{v_p^2}{v_q^4} \lesssim_L \eta^2 dP^2 \log^2 d$$

where we have used the induction hypotheses $v_q^2 \geq \Theta(1/d)$ and $v_p^2/v_q^2 = \Theta(v_{0,p}^2/v_{0,q}^2) = O(\log^2 d)$. Using the language of Lemma F.8, these imply

$$\sigma_Z^2 \lesssim_L \eta^2 dP^2 \log^4 d. \tag{11}$$

Then, for the higher-order terms, first by the tail bounds in Lemma 2.2, we have for any $\delta_{\mathbb{P},\xi} \in (0,1)$, that

$$|Z_p| \vee |Z_q| \leq C_L^{2L} P \log^{2L}\left(\frac{C_L}{\delta_{\mathbb{P},\xi}}\right) \quad \text{with probability at least } 1 - 2\delta_{\mathbb{P},\xi}.$$

In particular, this implies that with at least the same probability, we have

$$|\xi_p| \vee |\xi_q| \lesssim_L \eta m a_0 \vee \eta^2 P^2 \log^{4L}\left(\frac{1}{\delta_{\mathbb{P},\xi}}\right).$$

Suppose that $\eta \leq 1/d$. Then, we have

$$\left|\frac{\xi_p+4L\eta v_p^{2L}}{\hat{v}_{q,t+1}^2} + \frac{v_p^2}{v_q^2}\frac{4L\eta v_q^{2L}+\xi_q}{\hat{v}_{q,t+1}^2}\right| \lesssim_L \log^2 d\left(\frac{|\xi_p|+|\xi_q|}{v_q^2} + \eta\left(1+\frac{v_p^2}{v_q^2}\right)\left(v_p^{2L-2}+v_q^{2L-2}\right)\right)$$

$$\lesssim_L \eta m a_0 d\log^2 d + \eta\frac{\log^{2L} d}{P^{L-1}} + \eta^2 dP^2 \log^{4L+2}\left(\frac{d}{\delta_{\mathbb{P}}}\right),$$

and

$$\left|\frac{2\eta v_p Z_p}{(1+2\eta(2Lv_q^{2L-2}+2-\rho))v_q^2}\frac{2\eta v_q Z_q+\xi_q}{\hat{v}_{q,t+1}^2}\right|$$

$$\lesssim_L \frac{\eta^2 |v_p Z_p|}{v_q^3}|Z_q| + \frac{\eta|v_p Z_p|}{v_q^4}|\xi_q|$$

$$\lesssim_L \eta^2 dP^2 \log^{4L+1}\left(\frac{d}{\delta_{\mathbb{P}}}\right) + \eta^3 d^{1.5}P^3 \log^{6L+1}\left(\frac{d}{\delta_{\mathbb{P}}}\right) + \eta^2 d^{1.5}P\log^{2L+1}\left(\frac{d}{\delta_{\mathbb{P}}}\right)m a_0,$$

and, similarly,

$$\left|\frac{v_p^2}{v_q^2}\frac{2\eta v_q Z_q}{(1+2\eta(2-\rho))v_q^2+4L\eta v_q^{2L}}\frac{2\eta v_q Z_q+\xi_q}{\hat{v}_{q,t+1}^2}\right|$$

$$\lesssim_L \frac{v_p^2\eta|Z_q|}{|v_q|^5}\left(\eta|v_q Z_q|+|\xi_q|\right)$$

$$\lesssim_L \eta^2 dP^2 \log^{4L+2}\left(\frac{d}{\delta_{\mathbb{P}}}\right) + \eta^3 d^{1.5}P^3 \log^{6L+2}\left(\frac{d}{\delta_{\mathbb{P}}}\right) + \eta^2 d^{1.5}P\log^{2L+2}\left(\frac{d}{\delta_{\mathbb{P}}}\right)m a_0.$$

Suppose that $\eta \leq 1/(dP^2)$, which is implied by the condition of Lemma C.4. Then, using the language of Lemma F.8, we have

$$\Xi \lesssim_L \eta m a_0 d\log^2 d + \eta\frac{\log^{2L} d}{P^{L-1}} + \eta^2 dP^2 \log^{4L+2}\left(\frac{d}{\delta_{\mathbb{P}}}\right). \tag{12}$$

Combine this with (11), recall $T\eta = O(\log d)$, apply Lemma F.8, and we obtain

$$\left|\frac{v_p^2}{v_q^2} - \frac{v_{0,p}^2}{v_{0,q}^2}\right| \lesssim_L T\eta m a_0 d\log^2 d + T\eta\frac{\log^{2L} d}{P^{L-1}} + T\eta^2 dP^2 \log^{4L+2}\left(\frac{d}{\delta_{\mathbb{P}}}\right)\sqrt{\delta_{\mathbb{P}}^{-1}T\eta^2 dP^2 \log^4 d}$$

$$\lesssim_L m a_0 d\log^3 d + \frac{\log^{2L+1} d}{P^{L-1}} + \eta dP^2 \log^{4L+3}\left(\frac{d}{\delta_{\mathbb{P}}}\right)\sqrt{\delta_{\mathbb{P}}^{-1}\eta dP^2 \log^5 d},$$

throughout Stage 1.1 with probability at least $1 - \Theta(\delta_\mathbb{P})$. For the RHS to be bounded by $\varepsilon_R \in (0, 1)$, it suffices to require

$$ma_0 \lesssim_L \frac{\varepsilon_R}{d \log^3 d}, \quad P \gtrsim_L \frac{\log^3 d}{\varepsilon_R}, \quad \eta \lesssim_L \frac{\varepsilon_R \sqrt{\delta_\mathbb{P}}}{dP^2 \log^{2L+2}(d/\delta_\mathbb{P})}.$$

$\square$

### C.1.3 Other Induction Hypotheses

First, we verify the induction hypothesis: $v_p^2 \leq \log^2 d/P$ for all $p \in [P]$. This condition is used to ensure the influence of the higher-order term is small compared to the influence of the second-order terms.

**Lemma C.7** (Bounds for moderately large $v_p^2$). *Let $\boldsymbol{v}$ be an arbitrary first-layer neuron. Suppose that $p \in [P]$ and $c_v/d \leq v_{0,p}^2 \ll c_v' \log^2 d/d$ for some small $c_v, c_v' > 0$. Then, if we choose*

$$ma_0 \lesssim_L \frac{c_v}{d \log d} \quad and \quad \eta \lesssim_L \frac{c_v(1 \wedge c_v)\delta_\mathbb{P}}{dP^2 \log^{4L+1}(d/\delta_\mathbb{P})},$$

*then there exists a universal constant $C \geq 1$ such that with probability at least $1 - O(\delta_\mathbb{P})$, we have*

$$\frac{1}{2}(1 + 4\eta)^t v_{0,p}^2 \leq v_{t,p}^2 \leq \frac{3C}{2}(1 + 4\eta)^t v_{0,p}^2, \quad \forall t \leq T.$$

*In particular, this implies $v_{t,p}^2 \leq \log^2 d/P$ throughout Stage 1.1.*

*Proof.* First, by Lemma C.1, for any $p \leq P$, we have

$$\hat{v}_{t+1,p}^2 \leq \left(1 + 4\eta + 4L\eta v_p^{2L-2}\right) v_p^2 + 2\eta v_p Z_p + 300L^3 \eta ma_0 + 300L^3 \eta^2 \left(1 \vee Z_p^2\right)$$

$$\leq \left(1 + 4\eta \left(1 + L\left(\frac{\log^2 d}{P}\right)^{L-1}\right)\right) v_p^2 + 2\eta v_p Z_p + 300L^3 \eta ma_0 + 300L^3 \eta^2 \left(1 \vee Z_p^2\right),$$

where the second line comes from the induction hypothesis $v_p^2 \leq \log^2 d/P$. For notational simplicity, put $\delta_v = L\left(\log^2 d/P\right)^{L-1}$ (as in the proof of Lemma C.4) and $\xi_{t+1,p} = 300L^3 \eta ma_0 + 300L^3 \eta^2 \left(1 \vee Z_p^2\right)$, so that the above can be rewritten as

$$v_{t+1,p}^2 \leq \hat{v}_{t+1,p}^2 \leq (1 + 4\eta(1 + \delta_v)) v_p^2 + 2\eta v_p Z_p + \xi_p.$$

By the tail bound in Lemma 2.2, there exists some constant $C_L > 0$ that may depend on $L$ such that for any $\delta_{\mathbb{P},\xi} \in (0, 1)$, we have

$$|Z_p| \leq C_L^{2L} P \log^{2L}\left(\frac{C_L}{\delta_{\mathbb{P},\xi}}\right) \quad \text{with probability at least } 1 - \delta_{\mathbb{P},\xi}.$$

Meanwhile, for the martingale difference term, by our induction hypothesis on $v_p$ and the variance estimate in Lemma 2.2, we have

$$\mathbb{E}\left[(2\eta v_p Z_p)^2 \mid \mathcal{F}_t\right] \leq 4C_L \eta^2 v_p^2 P^2 \lesssim_L (1 + 4\eta(1 + \delta_v))^t \eta^2 v_{0,P}^2 P^2$$

$$\lesssim_L (1 + 4\eta(1 + \delta_v))^t \eta^2 \frac{P^2 \log^2 d}{d}.$$

Using the language of Lemma F.6, these mean

$$\Xi \lesssim_L \eta \left(ma_0 \vee \eta P^2 \log^{4L}\left(\frac{1}{\delta_{\mathbb{P},\xi}}\right)\right), \quad \sigma_Z^2 \lesssim_L \eta^2 \frac{P^2 \log^2 d}{d}.$$

Put $x_t = (1 + 4\eta(1 + \delta_v))^t v_{0,p}^2$ where $x_0 = v_{0,p}^2 \geq c_v/d$. By the proof of Lemma C.4, we know $(1 + 4\eta)^T = \Theta(d/P)$. In particular, this implies $\eta T = \frac{1 + o(1)}{4} \log(d/P)$. Then, by Lemma F.6, we

have $v_p^2 \leq (1 \pm 0.5)x_t$ with probability at least $1 - 2\delta_{\mathbb{P}}$, as long as $ma_0$ and $\eta$ are chosen so that

$$
\eta \left( ma_0 \vee \eta P^2 \log^{4L} \left( \frac{T}{\delta_{\mathbb{P}}} \right) \right) \lesssim_L \frac{x_0}{4T} \quad \Leftarrow \quad
\begin{cases}
ma_0 \lesssim_L \dfrac{c_v}{d \log d}, \\
\eta \lesssim_L \dfrac{c_v}{dP^2 \log^{4L+1}(d/\delta_{\mathbb{P}})}
\end{cases}
$$

$$
\eta^2 \frac{P^2 \log^2 d}{d} \lesssim_L \frac{\delta_{\mathbb{P}} \alpha x_0^2}{16} \quad \Leftarrow \quad \eta \lesssim_L \frac{\delta_{\mathbb{P}} c_v^2}{dP^2 \log^2 d}.
$$

To complete the proof, we now estimate $x_t$. Clear that $x_t \geq (1 + 4\eta)^t x_0$. Meanwhile, we have

$$
\left( \frac{1 + 4\eta(1 + \delta_v)}{1 + 4\eta} \right)^T = \left( 1 + \frac{4\eta\delta_v}{1 + 4\eta} \right)^T \leq (1 + 4\eta\delta_v)^T
$$

$$
\leq \exp\left(4\eta T \delta_v\right) \leq \exp\left( (1 + o(1))\delta_v \log\left( \frac{d}{P} \right) \right) \leq (d/P)^{2\delta_v}.
$$

When $P \geq \log^3 d$, the last term is bounded by a universal constant $C > 0$. As a result, we have

$$
x_t \leq (1 + 4\eta(1 + \delta_v))^t x_0 = \left( \frac{1 + 4\eta(1 + \delta_v)}{1 + 4\eta} \right)^t (1 + 4\eta)^t x_0 \leq C(1 + 4\eta)^t x_0.
$$

$\square$

**Lemma C.8** (Upper bound for small $v_q^2$). *Let $\boldsymbol{v}$ be an arbitrary first-layer neuron. Suppose that $q \in [P]$ and $v_q^2 \leq c_v/d$ for some $c_v > 0$. Then, if we choose*

$$
ma_0 \lesssim_L \frac{c_v}{d \log d} \quad \text{and} \quad \eta \lesssim_L \frac{c_v(1 \wedge c_v)\delta_{\mathbb{P}}}{dP^2 \log^{4L+1}(d/\delta_{\mathbb{P}})},
$$

*then there exists a universal constant $C \geq 1$ such that with probability at least $1 - O(\delta_{\mathbb{P}})$, we have*

$$
v_{t,q}^2 \leq 2Cc_v \frac{(1 + 4\eta)^t}{d}, \quad \forall t \leq T.
$$

*Proof.* The proof is essentially the same as the previous one. It suffices to use Lemma F.7 in place of Lemma F.6. $\square$

The following lemma is not used in our proof. It serves as an example of using Lemma F.10 to obtain $\mathrm{poly}\log$ dependence on $\delta_{\mathbb{P}}$.

**Lemma C.9.** *There exists a constant $C_L > 0$ that may depend on $L$ such that if we choose*

$$
ma_0 \leq \frac{\log d}{C_L d} \quad \text{and} \quad \eta \leq \frac{1}{C_L dP \log^{2L+3}\left( \frac{Tmd}{\delta_{\mathbb{P}}} \right)},
$$

*then with probability at least $1 - \delta_{\mathbb{P}}$, we have*

$$
\sup_{i \in [m]} \sup_{r > P} \sup_{t \leq T} v_{i,t,r}^2 \leq \frac{\log^2 d}{d}.
$$

*Proof.* We will use Lemma F.10. Fix a first-layer neuron $\boldsymbol{v}$ and $r > P$. Assume the induction hypothesis $v_r^2 \leq K_v/d$ where $K_v > 0$ is a parameter to be determined later. Recall from Lemma C.1 that

$$
v_{t+1,r}^2 \leq \hat{v}_{t+1,r}^2 = (1 - 2\eta\rho) v_r^2 + 2\eta v_r Z_r \pm 300L^3 \eta ma_0 \pm 300L^3 \eta^2 \left( 1 \vee Z_r^2 \right).
$$

Let $\xi_{t+1,r}$ denote the last two terms. Then, we can write

$$
\hat{v}_{t+1,r}^2 \leq v_r^2 + 2\eta v_r Z_r + \xi_r.
$$

By the tail bound in Lemma 2.2, for any $\delta_{\mathbb{P}} \in (0, 1)$,

$$
|Z_r| \leq C_L^{2L} P \log^{2L}\left( \frac{T}{C_L \delta_{\mathbb{P}}} \right) \quad \text{with probability at least } 1 - \delta_{\mathbb{P}}/T.
$$

Hence, with probability at least $1 - \delta_\mathbb{P}/T$, we have

$$|\xi_r| \leq 300L^3\eta m a_0 + 300L^3\eta^2 C_L^{2L}P\log^{2L}\left(\frac{T}{C_L\delta_\mathbb{P}}\right)$$

$$\leq 600L^3C_L^{2L}\eta\left(ma_0 \vee \eta P\log^{2L}\left(\frac{T}{C_L\delta_\mathbb{P}}\right)\right) =: \Xi.$$

Meanwhile, for the martingale difference terms, $Z_r$ satisfies the tail bound (15) with $a = C_L$, $b = P^{-1/(2L)}$, $c = 1/(2L)$, and $\sigma_Z^2 = C_L P^2$. Hence, by Lemma F.10, we have

$$\sup_{t\leq T}\left|v_{t,r}^2 - v_{0,r}^2\right| \leq T\Xi + \frac{2K_v\eta C_c}{d}\sqrt{T\left(\sigma_Z^2 + \frac{1}{b^{2/c}} + \frac{\log^{1/c}\left(\frac{aT}{b\sigma_Z\delta_\mathbb{P}}\right)}{b^{1/c}}\right)\log\left(\frac{T}{\delta_\mathbb{P}}\right)}$$

$$\leq C_L'T\eta\left(ma_0 \vee \eta P\log^{2L}\left(\frac{T}{C_L\delta_\mathbb{P}}\right)\right) + \frac{K_v}{d}C_L'\sqrt{\eta^2 T}P\log^{L+1}\left(\frac{C_L PT}{\delta_\mathbb{P}}\right),$$

with probability at least $1 - 2\delta_\mathbb{P}$, for some constant $C_L' > 0$ that may depend on $L$. Recall that $T \leq \eta^{-1}\log d$. Therefore,

$$\sup_{t\leq T}\left|v_{t,r}^2 - v_{0,r}^2\right| \leq C_L'\log d\left(ma_0 \vee \eta P\log^{2L}\left(\frac{T}{\delta_\mathbb{P}}\right)\right) + \frac{K_v}{d}C_L'\sqrt{\eta\log d}P\log^{L+1}\left(\frac{PT}{\delta_\mathbb{P}}\right),$$

with probability at least $1 - 2\delta_\mathbb{P}$. Thus, apply the union bound over all neurons and all $r > P$, replace $\delta_\mathbb{P}$ with $\delta_\mathbb{P}/(2md)$, and we obtain

$$\sup_{i\in[m]}\sup_{r>P}\sup_{t\leq T}\left|v_{i,t,r}^2 - v_{i,0,r}^2\right| \leq C_L''\log d\left(ma_0 \vee \eta P\log^{2L}\left(\frac{Tmd}{\delta_\mathbb{P}}\right)\right)$$

$$+ \frac{K_v}{d}C_L''\sqrt{\eta\log d}P\log^{L+1}\left(\frac{PTmd}{\delta_\mathbb{P}}\right),$$

with probability at least $1 - \delta_\mathbb{P}$. Finally, recall that we assume $\sup_{i\in[m]}\sup_{r>P}\sup_{t\leq T}v_{i,0,r}^2 \leq \log^2/(2d)$. Choose $K_v = \log^2 d$. Then, we have $\sup_{i\in[m]}\sup_{r>P}\sup_{t\leq T}v_{i,t,r}^2 \leq \log^2 d/d$ with probability at least $1 - \delta_\mathbb{P}$ as long as

$$C_L''\log d\left(ma_0 \vee \eta P\log^{2L}\left(\frac{Tmd}{\delta_\mathbb{P}}\right)\right) \leq \frac{\log^2 d}{2d} \quad \Leftarrow \quad \begin{cases} ma_0 \leq \dfrac{\log d}{2C_L''d} \\ \eta \leq \dfrac{\log d}{2C_L''dP\log^{2L}\left(\frac{Tmd}{\delta_\mathbb{P}}\right)} \end{cases}$$

$$\frac{K_v}{d}C_L''\sqrt{\eta\log d}P\log^{L+1}\left(\frac{PTmd}{\delta_\mathbb{P}}\right) \leq \frac{\log^2 d}{2d} \quad \Leftarrow \quad \eta \leq \frac{1}{4(C_L'')^2P^2\log^{2L+3}\left(\frac{PTmd}{\delta_\mathbb{P}}\right)}.$$

$\square$

### C.2 STAGE 1.2: RECOVERY OF THE DIRECTIONS

Let $\boldsymbol{v}$ be an arbitrary first-layer neuron. Assume w.l.o.g. that $v_1^2$ is the largest at initialization and $v_{0,1}^2/\max_{2\leq k\leq P}v_{0,k}^2 \geq 1 + c_g$ for some small constant $c_g > 0$. By Lemma C.2, we know this gap can be approximately preserved. In other words, we may assume that $v_{T_1,1}^2/\max_{2\leq k\leq P}v_{T_1,k}^2 \geq 1 + c_g$ for some small constant $c_g > 0$ that is potentially smaller than the previous $c_g$. In this subsection, we show that $v_1^2$ will grow from $\Omega(1/P)$ to $3/4$ and then to close to 1. Formally, we prove the following lemma.

**Lemma C.10** (Stage 1.2). *Let $\boldsymbol{v} \in \mathbb{S}^{d-1}$ be an arbitrary first-layer neuron satisfying $v_{T_1,1}^2 \geq c/P$ and $v_{T_1,1}^2/\max_{2\leq k\leq P}v_{T_1,k}^2 \geq 1 + c$ for some small universal constant $c > 0$. Let $\delta_\mathbb{P} \in (e^{-\log^C d}, 1)$ and $\varepsilon_v > 0$ be given. Suppose that we choose*

$$ma_0 \lesssim_L \frac{\varepsilon_v}{dP^{2L}\log(1/\varepsilon_v)} \quad \text{and} \quad \eta \lesssim_L \frac{\varepsilon_v^2\delta_\mathbb{P}}{dP^{L+3}\log^{4L}(d/\delta_\mathbb{P})}.$$

*Then, with probability at least $1 - O(\delta_{\mathbb{P}})$, we have $v_1^2 \geq 1 - \varepsilon_v$ within $O_L\left(\left(P^{L-1} + \log(1/\varepsilon_v)\right)/\eta\right)$ iterations.*

*Proof.* It suffices to combine Lemma C.12 and Lemma C.13. $\qquad\square$

**Lemma C.11** (Dynamics of $v_1^2$). *We have*

$$v_{t+1,1}^2 = v_1^2 \left(1 + 4L\eta v_1^{2L-2} - 4L\eta \|\boldsymbol{v}\|_{2L}^{2L}\right) + \frac{2\eta v_1 Z_1 - 2\eta \langle \boldsymbol{v}, \boldsymbol{Z} \rangle}{1 + 2\eta(2-\rho) + 4L\eta \|\boldsymbol{v}\|_{2L}^{2L}} + \xi_{t+1}$$

*where $\xi_t$ satisfies $|\xi_t| \leq C_L \eta d \left(ma_0 \vee \eta P^2 \log^{4L}\left(\frac{d}{\delta_{\mathbb{P},\xi}}\right)\right)$, with probability least $1 - \delta_{\mathbb{P},\xi}$ for some constant $C_L > 0$ that can depend on $L$.*

*Proof.* Recall from Lemma C.1 that

$$\hat{v}_{t+1,1}^2 = \left(1 + 2\eta\left(2Lv_1^{2L-2} + 2 - \rho\right)\right)v_1^2 + 2\eta v_1 Z_1$$
$$\underbrace{\pm 300L^3\eta ma_0 \pm 300L^3\eta^2\left(1 \vee Z_k^2\right)}_{=:\ \xi_{1,t+1}}$$
$$= v_1^2\left(1 + 2\eta(2-\rho) + 4L\eta v_1^{2L-2}\right) + 2\eta v_1 Z_1 + \xi_{1,t+1},$$

where $\rho := 2\|\boldsymbol{v}_{\leq P}\|^2 + 2L\|\boldsymbol{v}_{\leq P}\|_{2L}^{2L}$. Meanwhile, we also have

$$\|\hat{\boldsymbol{v}}_{t+1}\|^2 = \sum_{k=1}^{d}\left(1 + 2\eta\left(2Lv_k^{2L-2} + 2 - \rho\right)\right)v_k^2 + 2\eta\langle \boldsymbol{v}, \boldsymbol{Z} \rangle + \langle \boldsymbol{1}, \boldsymbol{\xi} \rangle$$
$$= 1 + 2\eta(2-\rho) + 4L\eta\|\boldsymbol{v}\|_{2L}^{2L} + 2\eta\langle \boldsymbol{v}, \boldsymbol{Z} \rangle + \langle \boldsymbol{1}, \boldsymbol{\xi} \rangle.$$

Then, we compute

$$v_{t+1,1}^2 = \frac{v_1^2\left(1 + 2\eta(2-\rho) + 4L\eta v_1^{2L-2}\right) + 2\eta v_1 Z_1 + \xi_{1,t+1}}{1 + 2\eta(2-\rho) + 4L\eta\|\boldsymbol{v}\|_{2L}^{2L} + 2\eta\langle \boldsymbol{v}, \boldsymbol{Z} \rangle + \langle \boldsymbol{1}, \boldsymbol{\xi} \rangle}$$
$$= v_1^2\frac{1 + 2\eta(2-\rho) + 4L\eta v_1^{2L-2}}{1 + 2\eta(2-\rho) + 4L\eta\|\boldsymbol{v}\|_{2L}^{2L} + 2\eta\langle \boldsymbol{v}, \boldsymbol{Z} \rangle + \langle \boldsymbol{1}, \boldsymbol{\xi} \rangle}$$
$$+ \frac{2\eta v_1 Z_1}{1 + 2\eta(2-\rho) + 4L\eta\|\boldsymbol{v}\|_{2L}^{2L} + 2\eta\langle \boldsymbol{v}, \boldsymbol{Z} \rangle + \langle \boldsymbol{1}, \boldsymbol{\xi} \rangle} + \frac{\xi_{1,t+1}}{\|\hat{\boldsymbol{v}}_{t+1}\|^2}$$
$$=:\ \mathtt{Tmp}_1 + \mathtt{Tmp}_2 + \mathtt{Tmp}_3.$$

For notational simplicity, we define $N_v^2 := 1 + 2\eta(2-\rho) + 4L\eta\|\boldsymbol{v}\|_{2L}^{2L}$. Meanwhile, by the tail bound in Lemma 2.2, for each $k \in [d]$ and any $\delta_{\mathbb{P},\xi} \in (0,1)$, we have

$$|Z_k| \leq C_{2L}^L P \log^{2L}\left(\frac{C_L}{\delta_{\mathbb{P},\xi}}\right) \quad \text{with probability at least } 1 - \delta_{\mathbb{P},\xi}.$$

Then, by union bound, with at least the same probability, we have

$$|\langle \boldsymbol{v}, \boldsymbol{Z} \rangle| \vee \max_{k \in [d]}|Z_k| \leq C_L^{2L} P \log^{2L}\left(\frac{2C_L d}{\delta_{\mathbb{P},\xi}}\right).$$

As a result, with at least the same probability, we have

$$|\xi_1| \leq 600L^3\eta\left(ma_0 \vee \eta C_L^{4L} P^2 \log^{4L}\left(\frac{2C_L d}{\delta_{\mathbb{P},\xi}}\right)\right),$$

$$|\langle \boldsymbol{1}, \boldsymbol{\xi} \rangle| \leq 600L^3\eta d\left(ma_0 \vee \eta C_L^{4L} P^2 \log^{4L}\left(\frac{2C_L d}{\delta_{\mathbb{P},\xi}}\right)\right).$$

Now, we are ready to analyze each of $\mathtt{Tmp}_i$ $(i \in [3])$.

First, for the signal term $\mathtt{Tmp}_1$, we write

$$\frac{1 + 2\eta\left(2 - \rho\right) + 4L\eta v_1^{2L-2}}{1 + 2\eta\left(2 - \rho\right) + 4L\eta\left\|\boldsymbol{v}\right\|_{2L}^{2L} + 2\eta\left\langle\boldsymbol{v}, \boldsymbol{Z}\right\rangle + \left\langle\boldsymbol{1}, \boldsymbol{\xi}\right\rangle}$$

$$= \frac{1 + 2\eta\left(2 - \rho\right) + 4L\eta v_1^{2L-2}}{1 + 2\eta\left(2 - \rho\right) + 4L\eta\left\|\boldsymbol{v}\right\|_{2L}^{2L}}\left(1 - \frac{2\eta\left\langle\boldsymbol{v}, \boldsymbol{Z}\right\rangle + \left\langle\boldsymbol{1}, \boldsymbol{\xi}\right\rangle}{N_v^2 + 2\eta\left\langle\boldsymbol{v}, \boldsymbol{Z}\right\rangle + \left\langle\boldsymbol{1}, \boldsymbol{\xi}\right\rangle}\right)$$

$$= \frac{1 + 2\eta\left(2 - \rho\right) + 4L\eta v_1^{2L-2}}{1 + 2\eta\left(2 - \rho\right) + 4L\eta\left\|\boldsymbol{v}\right\|_{2L}^{2L}}\left(1 - \frac{2\eta\left\langle\boldsymbol{v}, \boldsymbol{Z}\right\rangle}{N_v^2}\left(1 - \frac{2\eta\left\langle\boldsymbol{v}, \boldsymbol{Z}\right\rangle + \left\langle\boldsymbol{1}, \boldsymbol{\xi}\right\rangle}{\left\|\hat{\boldsymbol{v}}_{t+1}\right\|^2}\right) - \frac{\left\langle\boldsymbol{1}, \boldsymbol{\xi}\right\rangle}{\left\|\hat{\boldsymbol{v}}_{t+1}\right\|^2}\right)$$

$$= \frac{1 + 2\eta\left(2 - \rho\right) + 4L\eta v_1^{2L-2}}{1 + 2\eta\left(2 - \rho\right) + 4L\eta\left\|\boldsymbol{v}\right\|_{2L}^{2L}}\left(1 - \frac{2\eta\left\langle\boldsymbol{v}, \boldsymbol{Z}\right\rangle}{N_v^2} \pm 4\eta^2\left\langle\boldsymbol{v}, \boldsymbol{Z}\right\rangle^2 \pm \left|\left\langle\boldsymbol{1}, \boldsymbol{\xi}\right\rangle\right|\right).$$

For the first factor, by (10), we have

$$\frac{1 + 2\eta\left(2 - \rho\right) + 4L\eta v_1^{2L-2}}{1 + 2\eta\left(2 - \rho\right) + 4L\eta\left\|\boldsymbol{v}\right\|_{2L}^{2L}}$$

$$= \left(1 + 2\eta\left(2 - \rho\right) + 4L\eta v_1^{2L-2}\right)\left(1 - 2\eta\left(2 - \rho\right) - 4L\eta\left\|\boldsymbol{v}\right\|_{2L}^{2L} \pm 160L^2\eta^2\right)$$

$$= 1 + 4L\eta v_1^{2L-2} - 4L\eta\left\|\boldsymbol{v}\right\|_{2L}^{2L} \pm 300L^2\eta^2.$$

As a result, we have

$$\frac{\mathtt{Tmp}_1}{v_1^2} = \left(1 + 4L\eta v_1^{2L-2} - 4L\eta\left\|\boldsymbol{v}\right\|_{2L}^{2L} \pm 300L^2\eta^2\right)\left(1 - \frac{2\eta\left\langle\boldsymbol{v}, \boldsymbol{Z}\right\rangle}{N_v^2} \pm 4\eta^2\left\langle\boldsymbol{v}, \boldsymbol{Z}\right\rangle^2 \pm \left|\left\langle\boldsymbol{1}, \boldsymbol{\xi}\right\rangle\right|\right)$$

$$= 1 + 4L\eta v_1^{2L-2} - 4L\eta\left\|\boldsymbol{v}\right\|_{2L}^{2L} - \frac{2\eta\left\langle\boldsymbol{v}, \boldsymbol{Z}\right\rangle}{N_v^2}$$

$$\pm O_L(1)\eta d\left(ma_0 \vee \eta P^2 \log^{4L}\left(\frac{d}{\delta_{\mathbb{P}, \xi}}\right)\right).$$

Then, we consider the (approximate) martingale difference term $\mathtt{Tmp}_2$. We have

$$\mathtt{Tmp}_2 = \frac{2\eta v_1 Z_1}{N_v^2}\left(1 - \frac{2\eta\left\langle\boldsymbol{v}, \boldsymbol{Z}\right\rangle + \left\langle\boldsymbol{1}, \boldsymbol{\xi}\right\rangle}{\left\|\hat{\boldsymbol{v}}_{t+1}\right\|^2}\right)$$

$$= \frac{2\eta v_1 Z_1}{N_v^2} \pm O_L(1)\eta d\left(ma_0 \vee \eta P^2 \log^{4L}\left(\frac{d}{\delta_{\mathbb{P}, \xi}}\right)\right).$$

Thus, we have

$$v_{t+1,1}^2 = v_1^2\left(1 + 4L\eta v_1^{2L-2} - 4L\eta\left\|\boldsymbol{v}\right\|_{2L}^{2L}\right) - \frac{2\eta\left\langle\boldsymbol{v}, \boldsymbol{Z}\right\rangle}{N_v^2} + \frac{2\eta v_1 Z_1}{N_v^2}$$

$$\pm O_L(1)\eta d\left(ma_0 \vee \eta P^2 \log^{4L}\left(\frac{d}{\delta_{\mathbb{P}, \xi}}\right)\right).$$

$\square$

**Lemma C.12** (Weak recovery of directions). *Suppose that we choose*

$$ma_0 \le \frac{c_{g,L}}{dP^{2L}} \quad \text{and} \quad \eta \le \frac{c_{g,L}\delta_{\mathbb{P}}}{dP^{L+3}\log^{4L}\left(d/\delta_{\mathbb{P}}\right)}.$$

*Then within* $O_L\left(\frac{P^{L-1}}{\eta c_{g,L}}\right)$ *iterations, we will have* $v_1^2 \ge 3/4$ *with probability at least* $1 - O(\delta_{\mathbb{P}})$.

*Proof.* By Lemma C.11, we have

$$v_{t+1,1}^2 = v_1^2\left(1 + 4L\eta v_1^{2L-2} - 4L\eta\left\|\boldsymbol{v}\right\|_{2L}^{2L}\right) + \frac{2\eta v_1 Z_1 - 2\eta\left\langle\boldsymbol{v}, \boldsymbol{Z}\right\rangle}{1 + 2\eta\left(2 - \rho\right) + 4L\eta\left\|\boldsymbol{v}\right\|_{2L}^{2L}} + \xi_{t+1}$$

where $\xi_t$ satisfies $|\xi_t| \leq C_L \eta d \left( m a_0 \vee \eta P^2 \log^{4L} \left( \frac{d}{\delta_{\mathbb{P},\xi}} \right) \right)$, with probability least $1 - \delta_{\mathbb{P},\xi}$ for some constant $C_L > 0$ that can depend on $L$. Meanwhile, by the variance bound in Lemma 2.2, we have

$$\mathbb{E} \left[ \left( \frac{2\eta v_1 Z_1 - 2\eta \langle \boldsymbol{v}, \boldsymbol{Z} \rangle}{1 + 2\eta (2 - \rho) + 4L\eta \|\boldsymbol{v}\|_{2L}^{2L}} \right)^2 \;\middle|\; \mathcal{F}_t \right] \lesssim_L \eta^2 P^2.$$

For the signal term, we write

$$v_1^{2L-2} - \|\boldsymbol{v}_{\leq P}\|_{2L}^{2L} = v_1^{2L-2} - v_1^{2L} - \sum_{k=2}^{P} v_k^{2L}$$

$$= v_1^{2L-2} \left( 1 - v_1^2 \right) - \left( \|\boldsymbol{v}_{\leq P}\|^2 - v_1^2 \right) \sum_{k=2}^{P} \frac{v_k^2}{\|\boldsymbol{v}_{\leq P}\|^2 - v_1^2} v_k^{2L-2}.$$

Note that the last summation is a weighted average of $\{v_k^{2L-2}\}_{2 \leq k \leq P}$. Similar to the proof in Section C.1.2, we can maintain the induction hypothesis $v_1^2 / \max_{2 \leq k \leq P} v_k^2 \geq 1 + c_g/2$[7], which gives

$$\sum_{k=2}^{P} \frac{v_k^2}{\|\boldsymbol{v}_{\leq P}\|^2 - v_1^2} v_k^{2L-2} \leq \left( \max_{2 \leq k \leq P} v_k^2 \right)^{L-1} \leq \left( \frac{v_1^2}{1 + c_g/2} \right)^{L-1} = \frac{v_1^{2L-2}}{1 + c_{g,L}},$$

where $c_{g,L} > 0$ is a constant that depend on $L$ and $c_g$. Therefore,

$$v_1^{2L-2} - \|\boldsymbol{v}_{\leq P}\|_{2L}^{2L} \geq v_1^{2L-2} \left( 1 - v_1^2 \right) - \left( \|\boldsymbol{v}_{\leq P}\|^2 - v_1^2 \right) \frac{v_1^{2L-2}}{1 + c_{g,L}}$$

$$= \frac{v_1^{2L-2}}{1 + c_{g,L}} \left( 1 - \|\boldsymbol{v}_{\leq P}\|^2 + c_{g,L} \left( 1 - v_1^2 \right) \right)$$

$$\geq \frac{c_{g,L}}{1 + c_{g,L}} v_1^{2L-2} \left( 1 - v_1^2 \right).$$

As a result, for the signal term, we have

$$v_1^2 \left( 1 + 4L\eta v_1^{2L-2} - 4L\eta \|\boldsymbol{v}\|_{2L}^{2L} \right) \geq v_1^2 \left( 1 + 4L\eta \frac{c_{g,L}}{1 + c_{g,L}} v_1^{2L-2} \left( 1 - v_1^2 \right) \right)$$

$$= v_1^2 + 4L\eta \frac{c_{g,L}}{1 + c_{g,L}} v_1^{2L} \left( 1 - v_1^2 \right)$$

$$\geq v_1^2 + \eta \frac{c_{g,L} L}{1 + c_{g,L}} v_1^{2L},$$

where the last line comes from the induction hypothesis $v_1^2 \leq 3/4$. Thus, using the notations of Lemma F.11, we have

$$\alpha = \eta \frac{c_{g,L} L}{1 + c_{g,L}}, \quad \Xi = C_L \eta d \left( m a_0 \vee \eta P^2 \log^{4L} \left( \frac{d}{\delta_{\mathbb{P},\xi}} \right) \right), \quad \sigma_Z^2 = C_L \eta^2 P^2,$$

for some large constant $C_L > 0$ that may differ from the previous one. Meanwhile, by Lemma F.12 and the assumption $x_0 = v_1^2 \geq \Omega(1/P)$, we have

$$T \lesssim \frac{1}{x_0^{L-1} \alpha} \leq \frac{P^{L-1}}{\alpha} \lesssim_L \frac{P^{L-1}}{\eta c_{g,L}}.$$

Thus, to meet the conditions of Lemma F.11, it suffices to choose

$$\Xi \leq \frac{x_0}{4T} \quad \Leftarrow \quad m a_0 \leq \frac{c_{g,L}}{dP^L}, \quad \eta \leq \frac{c_{g,L}}{dP^{L+2} \log^{4L} (d/\delta_{\mathbb{P}})},$$

$$\sigma_Z^2 \leq \frac{x_0^2 \delta_{\mathbb{P}}}{16T} \quad \Leftarrow \quad \eta \lesssim_L \frac{\delta_{\mathbb{P}} c_{g,L}}{P^{L+3}}.$$

$\square$

---

[7]The only difference is that now the $2L$-th order terms cannot be simply ignored as we no longer have the induction hypothesis $v_p^2 \leq \log^2 d/P$. To handle them, it suffices to note that if $v_1^2 \geq v_q^2$, then those $2L$-th order terms are also larger for $v_1^2$, which will even lead to an amplification of the gap. In fact, this is why we can recover the directions using them.

**Lemma C.13** (Strong recovery of directions). *Let $v \in \mathbb{S}^{d-1}$ be an arbitrary first-layer neuron. Let $\delta_{\mathbb{P}}$ and $\varepsilon_*$ be given. Suppose that we choose*

$$ma_0 \lesssim_L \frac{\varepsilon_*}{d \log(1/\varepsilon_*)} \quad and \quad \eta \lesssim_L \frac{\varepsilon_*^2 \delta_{\mathbb{P}}}{dP^2 \log^{4L}(d/\delta_{\mathbb{P}})}.$$

*Then, with probability at least $1 - O(\delta_{\mathbb{P}})$, we have $v_1^2 \geq 1 - \varepsilon_*$ within $O_L(\log(1/\varepsilon_*)/\eta)$ iterations.*

*Proof.* Again, by Lemma C.11, we have

$$v_{t+1,1}^2 = v_1^2 \left(1 + 4L\eta v_1^{2L-2} - 4L\eta \|v\|_{2L}^{2L}\right) + \frac{2\eta v_1 Z_1 - 2\eta \langle v, Z \rangle}{1 + 2\eta(2 - \rho) + 4L\eta \|v\|_{2L}^{2L}} + \xi_{t+1}$$

where $\xi_t$ satisfies $|\xi_t| \leq C_L \eta d \left(ma_0 \vee \eta P^2 \log^{4L}\left(\frac{d}{\delta_{\mathbb{P},\xi}}\right)\right)$, with probability least $1 - \delta_{\mathbb{P},\xi}$ for some constant $C_L > 0$ that can depend on $L$. Meanwhile, by the proof of the previous lemma, we have

$$v_1^2 \left(1 + 4L\eta v_1^{2L-2} - 4L\eta \|v\|_{2L}^{2L}\right) \geq v_1^2 \left(1 + 4L\eta \frac{c_{g,L}}{1 + c_{g,L}} v_1^{2L-2}\left(1 - v_1^2\right)\right)$$

$$= v_1^2 + 4L\eta \frac{c_{g,L}}{1 + c_{g,L}} v_1^{2L}\left(1 - v_1^2\right)$$

$$\geq v_1^2 + 4L\eta \frac{c_{g,L}}{1 + c_{g,L}} \left(\frac{3}{4}\right)^{2L}\left(1 - v_1^2\right).$$

This implies

$$1 - v_{t+1,1}^2 \leq \left(1 - v_1^2\right)\left(1 - 4L\eta \frac{c_{g,L}}{1 + c_{g,L}} \left(\frac{3}{4}\right)^{2L}\right) - \frac{2\eta v_1 Z_1 - 2\eta \langle v, Z \rangle}{1 + 2\eta(2 - \rho) + 4L\eta \|v\|_{2L}^{2L}} - \xi_{t+1}$$

For the martingale difference term, also by the previous proof, we have

$$\mathbb{E}\left[\left(\frac{2\eta v_1 Z_1 - 2\eta \langle v, Z \rangle}{1 + 2\eta(2 - \rho) + 4L\eta \|v\|_{2L}^{2L}}\right)^2 \middle| \mathcal{F}_t\right] \lesssim_L \eta^2 P^2.$$

Let $\varepsilon_* > 0$ denote our target accuracy. Hence, in the language of Lemma F.6,[8] we have

$$\alpha = -4L\eta \frac{c_{g,L}}{1 + c_{g,L}} \left(\frac{3}{4}\right)^{2L}, \qquad \eta T = O_L(\log(1/\varepsilon_*)),$$

$$\sigma_Z^2 = O_L(1)\eta^2 P^2, \qquad \Xi = O_L(1)\eta d\left(ma_0 \vee \eta P^2 \log^{4L}\left(\frac{Td}{\delta_{\mathbb{P}}}\right)\right).$$

To meet the conditions of Lemma F.6, it suffices to choose

$$\Xi \leq \frac{\varepsilon_*}{4T} \quad \Leftarrow \quad ma_0 \lesssim_L \frac{\varepsilon_*}{d \log(1/\varepsilon_*)}, \quad \eta \lesssim_L \frac{\varepsilon_*}{dP^2 \log(1/\varepsilon_*) \log^{4L}(d/\delta_{\mathbb{P}})},$$

$$\sigma_Z^2 \leq \frac{\delta_{\mathbb{P}}|\alpha|\varepsilon_*^2}{16} \quad \Leftarrow \quad \eta \lesssim_L \frac{\delta_{\mathbb{P}} c_{g,L} \varepsilon_*^2}{P^2}.$$

Then, with probability at least $1 - O(\delta_{\mathbb{P}})$, we have $v_1^2 \geq 1 - \varepsilon$ within $T = O_L(\log(1/\varepsilon_*)/\eta)$ iterations. $\qquad \square$

## C.3 DEFERRED PROOFS IN THIS SECTION

*Proof of Lemma C.1.* Recall that

$$\hat{v}_{t+1,k} = v_{t,k} + \eta\left(\mathbb{1}\{k \leq P\}\left(2 + 2Lv_k^{2L-2}\right) - \rho\right)v_k + \eta Z_{t+1,k} + \eta O_{t+1,k},$$

---

[8]When $\alpha$ is negative, it suffices to replace $x_0$ with our target $\varepsilon_*$.

where $|O_{t+1,k}| \le 2Lma_0$. Then, we compute

$$\hat{v}_{t+1,k}^2 = \left(\left(1 + \eta\left(\mathbb{1}\{k \le P\}\left(2 + 2Lv_k^{2L-2}\right) - \rho\right)\right)v_k + \eta O_k + \eta Z_k\right)^2$$

$$= \left(1 + 2\eta\left(\mathbb{1}\{k \le P\}\left(2 + 2Lv_k^{2L-2}\right) - \rho\right)\right)v_k^2 + 2\eta v_k Z_k + 2\eta v_k O_k$$

$$+ \eta^2\left(\mathbb{1}\{k \le P\}\left(2 + 2Lv_k^{2L-2}\right) - \rho\right)^2 v_k^2$$

$$+ 2\eta^2\left(\mathbb{1}\{k \le P\}\left(2 + 2Lv_k^{2L-2}\right) - \rho\right)v_k Z_k$$

$$+ 2\eta^2\left(\mathbb{1}\{k \le P\}\left(2 + 2Lv_k^{2L-2}\right) - \rho\right)v_k O_k$$

$$+ \eta^2 O_k^2 + \eta^2 Z_k^2 + 2\eta^2 Z_k O_k.$$

The last four lines, which we denote by $\mathrm{Tmp}^{(2)}$ for notational simplicity, contain terms that are quadratic in $\eta$. The first term is the second line is the "signal term" that corresponds to the GD update, the second term forms a martingale difference sequence and the second term captures the influence of other neuron and shrinks with $a_0$.

First, we bound the second-order terms. For $\rho$, we have the following naïve upper bound:

$$\rho = 2\sum_{i=1}^{P} v_i^2 + 2L\sum_{i=1}^{P} v_i^{2L} \le \left(2 + 2L\max_{j \le P} v_j^{2L-2}\right)\|\boldsymbol{v}_{\le P}\|^2 \le 2 + 2L\max_{j \le P} v_j^{2L-2} \le 4L, \quad (13)$$

where the last inequality comes from the fact $L \ge 2$. Similarly, we also have $2 + 2Lv_k^{2L-2} \le 4L$. Hence, we have

$$\left|\mathbb{1}\{k \le P\}\left(2 + 2Lv_k^{2L-2}\right) - \rho\right| \le 2 + 2Lv_k^{2L-2} + \rho \le 8L.$$

Thus, for the second-order terms (last four lines), we have

$$|\mathrm{Tmp}^{(2)}| \le 64L^2\eta^2 v_k^2 + 16L\eta^2|v_k Z_k| + 16L\eta^2|v_k O_k| + \eta^2 O_k^2 + \eta^2 Z_k^2 + 2\eta^2 Z_k O_k$$

$$\le 100L^2\eta^2 v_k^2 + 10L\eta^2 Z_k^2 + 10L\eta^2 O_k^2$$

$$\le 300L^3\eta^2\left(v_k^2 \vee Z_k^2 \vee m^2 a_0^2\right),$$

where we use the inequality $ab \le a^2/2 + b^2/2$ in the second line to handle the cross terms. In other words, we have

$$\hat{v}_{t+1,k}^2 = \left(1 + 2\eta\left(\mathbb{1}\{k \le P\}\left(2 + 2Lv_k^{2L-2}\right) - \rho\right)\right)v_k^2 + 2\eta v_k Z_k + 2\eta v_k O_k$$

$$\pm 300L^3\eta^2\left(v_k^2 \vee Z_k^2 \vee m^2 a_0^2\right).$$

Meanwhile, for the last term in the first line, we have $|2\eta v_k O_k| \le 4L\eta v_k ma_0$. Thus,

$$\hat{v}_{t+1,k}^2 = \left(1 + 2\eta\left(\mathbb{1}\{k \le P\}\left(2 + 2Lv_k^{2L-2}\right) - \rho\right)\right)v_k^2 + 2\eta v_k Z_k$$

$$\pm 4L\eta v_k ma_0 \pm 300L^3\eta^2 m^2 a_0^2 \pm 300L^3\eta^2\left(v_k^2 \vee Z_k^2\right)$$

$$= \left(1 + 2\eta\left(\mathbb{1}\{k \le P\}\left(2 + 2Lv_k^{2L-2}\right) - \rho\right)\right)v_k^2 + 2\eta v_k Z_k$$

$$\pm 300L^3\eta ma_0 \pm 300L^3\eta^2\left(1 \vee Z_k^2\right).$$

$\square$

# D    STAGE 2: TRAINING THE SECOND LAYER

**Lemma D.1.** *Suppose that for each $p \in [P]$, there exists a first-layer neuron $\boldsymbol{v}_{i_p}$ with $v_{i_p,p}^2 \ge 1 - \varepsilon_v$ for some small positive $\varepsilon_v = O(1/P)$, then we can choose $\boldsymbol{a}_* \in \mathbb{R}^m$ with $\|\boldsymbol{a}_*\| = \sqrt{P}$ such that*

$$\mathcal{L}(\boldsymbol{a}_*, \boldsymbol{V}) := \mathbb{E}\left(f_*(\boldsymbol{x}) - f(\boldsymbol{x}; \boldsymbol{a}_*, \boldsymbol{V})\right)^2 \le 10LP^2\varepsilon_v.$$

*Proof.* Choose one $\boldsymbol{v}_{i_p}$ for each $p \in [P]$. Then, we set the $i_p$-th entries of $\boldsymbol{a}_*$ to be 1 and all other entries 0. Then, we write

$$\left(f_*(\boldsymbol{x}) - f(\boldsymbol{x}; \boldsymbol{a}_*, \boldsymbol{V})\right)^2 = \left(\sum_{k=1}^{P}\left(\phi(x_k) - \phi(\boldsymbol{v}_{i_k} \cdot \boldsymbol{x})\right)\right)^2$$

$$= \sum_{k,l=1}^{P}\left(\phi(x_k) - \phi(\boldsymbol{v}_{i_k} \cdot \boldsymbol{x})\right)\left(\phi(x_l) - \phi(\boldsymbol{v}_{i_l} \cdot \boldsymbol{x})\right).$$

Recall from the proof of Lemma 2.1 (cf. Section A) that for any $\boldsymbol{v}, \boldsymbol{v}' \in \mathbb{S}^{d-1}$, we have

$$\mathbb{E}_{\boldsymbol{x} \sim \mathcal{N}(0,\boldsymbol{I})} [\phi(\boldsymbol{v} \cdot \boldsymbol{x})\phi(\boldsymbol{v}' \cdot \boldsymbol{x})] = \langle \boldsymbol{v}, \boldsymbol{v}' \rangle^2 + \langle \boldsymbol{v}, \boldsymbol{v}' \rangle^{2L}.$$

Hence, for $k = l$, we have

$$\begin{aligned}
\mathbb{E} \left(\phi(x_k) - \phi(\boldsymbol{v}_{i_k} \cdot \boldsymbol{x})\right)^2 &= \mathbb{E}\, \phi^2(x_k) + \mathbb{E}\, \phi^2(\boldsymbol{v}_{i_k} \cdot \boldsymbol{x}) - 2\, \mathbb{E}\, \phi(x_k)\phi(\boldsymbol{v}_{i_k} \cdot \boldsymbol{x}) \\
&= 4 - 2\left(v_{i_k,k}^2 + v_{i_k,k}^{2L}\right) \\
&\leq 4L\varepsilon_v.
\end{aligned}$$

Meanwhile, for $k \neq l$, we have

$$\begin{aligned}
&\mathbb{E} \left(\phi(x_k) - \phi(\boldsymbol{v}_{i_k} \cdot \boldsymbol{x})\right)\left(\phi(x_l) - \phi(\boldsymbol{v}_{i_l} \cdot \boldsymbol{x})\right) \\
&= \mathbb{E}\, \phi(x_k)\phi(x_l) + \mathbb{E}\, \phi(\boldsymbol{v}_{i_k} \cdot \boldsymbol{x})\phi(\boldsymbol{v}_{i_l} \cdot \boldsymbol{x}) - \mathbb{E}\, \phi(x_k)\phi(\boldsymbol{v}_{i_l} \cdot \boldsymbol{x}) - \mathbb{E}\, \phi(\boldsymbol{v}_{i_k} \cdot \boldsymbol{x})\phi(x_l) \\
&\leq \langle \boldsymbol{v}_{i_k}, \boldsymbol{v}_{i_l} \rangle^2 + \langle \boldsymbol{v}_{i_k}, \boldsymbol{v}_{i_l} \rangle^{2L}.
\end{aligned}$$

Note that

$$\langle \boldsymbol{v}_{i_k}, \boldsymbol{v}_{i_l} \rangle^2 \leq 2v_{i_l,k}^2 + 2\langle \boldsymbol{v}_{i_k} - \boldsymbol{e}_k, \boldsymbol{v}_{i_l} \rangle^2 \leq 2\varepsilon_v + 2\|\boldsymbol{v}_{i_k} - \boldsymbol{e}_k\|^2 = 2\varepsilon_v + 4\left(1 - v_{i_k,k}\right) \leq 6\varepsilon_v.$$

As a result, $\langle \boldsymbol{v}_{i_k}, \boldsymbol{v}_{i_l} \rangle^2 + \langle \boldsymbol{v}_{i_k}, \boldsymbol{v}_{i_l} \rangle^{2L} \leq 10\varepsilon_v$. Combining these two cases, we obtain

$$\mathbb{E} \left(f_*(\boldsymbol{x}) - f(\boldsymbol{x}; \boldsymbol{a}_*, \boldsymbol{V})\right)^2 \leq 4PL\varepsilon_v + 10P^2\varepsilon_v \leq 10LP^2\varepsilon_v.$$

$\square$

Now, we are ready to prove the following generalization bound for Stage 2. The proof of it is adapted from Section B.8 of Oko et al. (2024), which in turn is based on (Damian et al. (2022); Abbe et al. (2022); Ba et al. (2022)).

**Lemma D.2.** *Suppose that for each $p \in [P]$, there exists a first-layer neuron $\boldsymbol{v}_{i_p}$ with $v_{i_p,p}^2 \geq 1 - \varepsilon_v$ for some small positive $\varepsilon_v = O(1/P)$. Then, there exists some $\lambda > 0$ such that the ridge estimator $\hat{\boldsymbol{a}}$ we obtain in Stage 2 satisfies*

$$\|f(\cdot; \hat{\boldsymbol{a}}, \boldsymbol{V}) - f_*\|_{L^1(D)} \leq \frac{8\|\boldsymbol{a}_*\|\sqrt{m}}{\sqrt{N}\delta_{\mathbb{P}}} + \sqrt{10LP^2\varepsilon_v},$$

*with probability at least $1 - 2\delta_{\mathbb{P}}$.*

*Proof.* For notational simplicity, let $D = \mathcal{N}(0,1)$ and $\hat{D} = \frac{1}{N}\sum_{n=1}^{N} \delta_{\boldsymbol{x}_{T+n}}$ denote the empirical distribution of the samples we use in Stage 2. In addition, we write $f_{\boldsymbol{a}}$ for $f(\cdot; \boldsymbol{a}, \boldsymbol{V})$ where $\boldsymbol{V}$ is the first-layer weights we have obtained in Stage 1 and $\boldsymbol{X} = (\boldsymbol{x}_{T+n})_{n=1}^{N}$.

Let $\boldsymbol{a}_* \in \mathbb{R}^m$ denote the second-layer weights we constructed in Lemma D.1 and $\hat{\boldsymbol{a}} \in \mathbb{R}^m$ denote the ridge estimator obtained via minimizing $\boldsymbol{a} \mapsto \|f_* - f_{\boldsymbol{a}}\|_{L^2(\hat{D})}^2 + \lambda\|\boldsymbol{a}\|^2$. By the equivalence between norm-constrained linear regression and ridge regression, there exists $\lambda > 0$ such that

$$\|f_* - f_{\hat{\boldsymbol{a}}}\|_{L^2(\hat{D})}^2 \leq \|f_* - f_{\boldsymbol{a}_*}\|_{L^2(\hat{D})}^2 \quad \text{and} \quad \|\hat{\boldsymbol{a}}\| \leq \|\boldsymbol{a}_*\|.$$

Choose this $\lambda$ and let $\mathcal{F} := \{f(\cdot; \boldsymbol{a}) : \|\boldsymbol{a}\| \leq \|\boldsymbol{a}_*\|\}$ be our hypothesis class. Note that $f_{\hat{\boldsymbol{a}}} \in \mathcal{F}$. Moreover, we have

$$\begin{aligned}
\|f_{\hat{\boldsymbol{a}}} - f_*\|_{L^1(D)} &= \left(\|f_{\hat{\boldsymbol{a}}} - f_*\|_{L^1(D)} - \|f_{\hat{\boldsymbol{a}}} - f_*\|_{L^1(\hat{D})}\right) + \|f_{\hat{\boldsymbol{a}}} - f_*\|_{L^1(\hat{D})} \\
&\leq \sup_{\boldsymbol{a} : \|\boldsymbol{a}\| \leq \|\boldsymbol{a}_*\|} \left(\|f_{\boldsymbol{a}} - f_*\|_{L^1(D)} - \|f_{\boldsymbol{a}} - f_*\|_{L^1(\hat{D})}\right) + \|f_{\hat{\boldsymbol{a}}} - f_*\|_{L^1(\hat{D})} \\
&\leq \sup_{\boldsymbol{a} : \|\boldsymbol{a}\| \leq \|\boldsymbol{a}_*\|} \left(\|f_{\boldsymbol{a}} - f_*\|_{L^1(D)} - \|f_{\boldsymbol{a}} - f_*\|_{L^1(\hat{D})}\right) + \|f_{\boldsymbol{a}_*} - f_*\|_{L^2(\hat{D})},
\end{aligned}$$

where we used the fact that $\|f_{\hat{\boldsymbol{a}}} - f_*\|_{L^1(\hat{D})} \leq \|f_{\hat{\boldsymbol{a}}} - f_*\|_{L^2(\hat{D})} \leq \|f_{\boldsymbol{a}_*} - f_*\|_{L^1(\hat{D})}$ in the last line.

Now, we bound the first term. Let $\boldsymbol{\sigma} := (\sigma_n)_{n=1}^N$ be i.i.d. Rademacher variables that are also independent of everything else. By symmetrization and Theorem 7 of Meir & Zhang (2003), we have

$$
\mathbb{E}_{\boldsymbol{X}} \left[ \sup_{\boldsymbol{a}\,:\,\|\boldsymbol{a}\| \leq \|\boldsymbol{a}_*\|} \left( \|f_{\boldsymbol{a}} - f_*\|_{L^1(D)} - \|f_{\boldsymbol{a}} - f_*\|_{L^1(\hat{D})} \right) \right]
$$

$$
\leq 2 \, \mathbb{E}_{\boldsymbol{X},\boldsymbol{\sigma}} \sup_{\boldsymbol{a}\,:\,\|\boldsymbol{a}\| \leq \|\boldsymbol{a}_*\|} \frac{1}{N} \sum_{t=1}^N \sigma_t \left| f_a(\boldsymbol{x}_{T+n}) - f_*(\boldsymbol{x}_{T+n}) \right|
$$

$$
\leq 2 \, \mathbb{E}_{\boldsymbol{X},\boldsymbol{\sigma}} \sup_{\boldsymbol{a}\,:\,\|\boldsymbol{a}\| \leq \|\boldsymbol{a}_*\|} \frac{1}{N} \sum_{t=1}^N \sigma_t \left( f_a(\boldsymbol{x}_{T+n}) - f_*(\boldsymbol{x}_{T+n}) \right)
$$

$$
\leq \frac{2}{N} \, \mathbb{E}_{\boldsymbol{X},\boldsymbol{\sigma}} \sup_{\boldsymbol{a}\,:\,\|\boldsymbol{a}\| \leq \|\boldsymbol{a}_*\|} \sum_{t=1}^N \sigma_t f_a(\boldsymbol{x}_{T+n}) + 2 \, \mathbb{E}_{\boldsymbol{X},\boldsymbol{\sigma}} \frac{1}{N} \sum_{t=1}^N \sigma_t f_*(\boldsymbol{x}_{T+n}).^{\nearrow 0}
$$

Note that the first term is two times the Rademacher complexity $\mathrm{Rad}_N(\mathcal{F})$ of $\mathcal{F}$ (see, for example, Chapter 4 of Wainwright (2019)). By (the proof of) Lemma 48 of Damian et al. (2022), we have

$$
\mathrm{Rad}_N(\mathcal{F}) \leq \frac{\|\boldsymbol{a}_*\|}{\sqrt{N}} \sqrt{\mathbb{E}_{\boldsymbol{x} \sim \mathcal{N}(0,\boldsymbol{I}_d)} \|\phi(\boldsymbol{V}\boldsymbol{x})\|^2} = \frac{\|\boldsymbol{a}_*\|}{\sqrt{N}} \sqrt{\sum_{k=1}^m \mathbb{E}_{\boldsymbol{x} \sim \mathcal{N}(0,\boldsymbol{I}_d)} \phi^2(\boldsymbol{v}_k \cdot \boldsymbol{x})}
$$

$$
= \frac{\|\boldsymbol{a}_*\| \sqrt{m}}{\sqrt{N}} \sqrt{\mathbb{E}_{x_1 \sim \mathcal{N}(0,1)} \phi^2(x_1)}
$$

$$
= \frac{2 \|\boldsymbol{a}_*\| \sqrt{m}}{\sqrt{N}}.
$$

In other words, we have

$$
\mathbb{E} \sup_{\boldsymbol{a}\,:\,\|\boldsymbol{a}\| \leq \|\boldsymbol{a}_*\|} \left( \|f_{\boldsymbol{a}} - f_*\|_{L^1(D)} - \|f_{\boldsymbol{a}} - f_*\|_{L^1(\hat{D})} \right) \leq \frac{4 \|\boldsymbol{a}_*\| \sqrt{m}}{\sqrt{N}}.
$$

Hence, for any $\delta_{\mathbb{P}} \in (0,1)$, by Markov's inequality, we have

$$
\sup_{\boldsymbol{a}\,:\,\|\boldsymbol{a}\| \leq \|\boldsymbol{a}_*\|} \left( \|f_{\boldsymbol{a}} - f_*\|_{L^1(D)} - \|f_{\boldsymbol{a}} - f_*\|_{L^1(\hat{D})} \right) \leq \frac{4 \|\boldsymbol{a}_*\| \sqrt{m}}{\sqrt{N}\delta_{\mathbb{P}}},
$$

with probability at least $1 - \delta_{\mathbb{P}}$. Apply the same argument to $\|f_{\boldsymbol{a}_*} - f_*\|_{L^2(\hat{D})}$ and recall from Lemma D.1 that $\|f_{\boldsymbol{a}_*} - f_*\|_{L^2(D)}^2 \leq 10LP^2\varepsilon_v$, and we obtain

$$
\|f_{\hat{\boldsymbol{a}}} - f_*\|_{L^1(D)} \leq \frac{8 \|\boldsymbol{a}_*\| \sqrt{m}}{\sqrt{N}\delta_{\mathbb{P}}} + \sqrt{10LP^2\varepsilon_v},
$$

with probability at least $1 - 2\delta_{\mathbb{P}}$. $\qquad\square$

# E    PROOF OF THE MAIN THEOREM

**Theorem 2.1** (Main Theorem). *Consider the setting and algorithm described above. Let $C > 0$ be a large universal constant. Suppose that $\log^C d \leq P \leq d$ and $\{\boldsymbol{v}_k^*\}_{k=1}^P$ are orthonormal. Let $\delta_{\mathbb{P}} \in (\exp(-\log^C d), 1)$ and $\varepsilon_* > 0$ be given. Suppose that we choose $a_0, \eta, T, N$ satisfying*

$$
m = \Omega\left(P^8 \log^{1.5}(P \vee 1/\delta_{\mathbb{P}})\right), \quad a_0 = O_L\left(\frac{\varepsilon_*^2}{mdP^{2L+2}\log^3 d \log(1/\varepsilon_*)}\right), \quad N = \Omega_L\left(\frac{Pm}{\varepsilon_*^2\delta_{\mathbb{P}}^2}\right),
$$

$$
\eta = O_L\left(\frac{\varepsilon_*^4\delta_{\mathbb{P}}}{dP^{L+8}\log^{4L+1}(d/\delta_{\mathbb{P}})}\right) = \tilde{O}_L\left(\frac{\varepsilon_*^4\delta_{\mathbb{P}}}{dP^{L+8}}\right),
$$

$$
T = O_L\left(\frac{\log d + P^{L-1} + \log(P/\varepsilon_*)}{\eta}\right) = \tilde{O}_L\left(\frac{dP^{2L+7}}{\delta_{\mathbb{P}}\varepsilon_*^4}\right).
$$

*Then, there exists some $\lambda > 0$ such that at the end of training, we have $\mathcal{L}(\boldsymbol{a}, \boldsymbol{V}) \leq \varepsilon_*$ with probability at least $1 - O(\delta_{\mathbb{P}})$.*

*Proof.* First, by Lemma B.3, we should choose $m = 400P^8 \log^{1.5}(P \vee 1/\delta_{\mathbb{P}})$. Meanwhile, by Lemma D.2, to achieve target $L^1$-error $\varepsilon_*$ with probability at least $1 - O(\delta_{\mathbb{P}})$, we need

$$N \gtrsim \frac{Pm}{\varepsilon_*^2 \delta_{\mathbb{P}}^2}, \quad \varepsilon_v = O_L\left(\frac{\varepsilon_*^2}{P^2}\right).$$

Then, to meet the conditions of Lemma C.2 and Lemma C.10 (uniformly over those $P$ good neurons), we choose

$$a_0 = O_L\left(\frac{\varepsilon_*^2}{mdP^{2L+2}\log^3 d \log(1/\varepsilon_*)}\right), \quad \eta = O_L\left(\frac{\varepsilon_*^4 \delta_{\mathbb{P}}}{dP^{L+8}\log^{4L+1}(d/\delta_{\mathbb{P}})}\right).$$

By Lemma C.2 and Lemma C.10, the numbers of iterations needed for Stage 1.1 and Stage 1.2 are $O_L(\log(d/P)/\eta)$ and $O_L\left(\left(P^{L-1} + \log(1/\varepsilon_v)\right)/\eta\right)$, respectively. Thus, the total number of iterations is bounded by

$$T = O_L\left(\frac{\log d + P^{L-1} + \log(P/\varepsilon_*)}{\eta}\right) = \tilde{O}_L\left(\frac{d\operatorname{poly}(P)}{\varepsilon_*^4 \delta_{\mathbb{P}}}\right).$$

$\square$

# F    TECHNICAL LEMMAS

## F.1    CONCENTRATION AND ANTI-CONCENTRATION OF GAUSSIAN VARIABLES

In this subsection, we first present several concentration and anti-concentration results for Gaussian variables. While almost all of them have been proved in the past in different papers and textbooks such as (van Handel (2016); Wainwright (2019)), we provide proofs of most of them for easier reference.

**Lemma F.1** (Concentration of norm). *Let $\boldsymbol{Z} \sim \mathcal{N}(0, \boldsymbol{I}_d)$. Then, we have*

$$\mathbb{P}\left(|\|\boldsymbol{Z}\| - \mathbb{E}\|\boldsymbol{Z}\|| \geq s\right) \leq 2e^{-s^2/2}.$$

**Remark.** $\|\boldsymbol{Z}\|$ follows the chi distribution $\chi_d$, whose expectation is $\sqrt{2}\Gamma((d+1)/2)/\Gamma(d/2)$. With Stirling's formula, one can show that for any large $d$,

$$\sqrt{d} \geq \mathbb{E}\|\boldsymbol{Z}\| = \sqrt{d-1}\left(1 - \frac{1}{4d} + \frac{O(1)}{d^2}\right) = \sqrt{d}\left(1 - \frac{2}{d}\right).$$

♣

*Proof.* We will use without proof the following result: if $\boldsymbol{Z} \sim \mathcal{N}(0, \boldsymbol{I}_d)$ and $f : \mathbb{R}^d \to \mathbb{R}$ is 1-Lipschitz, then $f(\boldsymbol{Z})$ is 1-subgaussian. We apply this result to the 1-Lipschitz function $\|\cdot\|$. This gives $\mathbb{P}\left(\|\boldsymbol{Z}\| - \mathbb{E}\|\boldsymbol{Z}\| \geq s\right) \leq e^{-s^2/2}$. Apply the same result to $-\|\cdot\|$ yields the lower tails.    $\square$

**Lemma F.2** (Upper tail for the maximum). *Let $Z_1, \ldots, Z_d \sim \mathcal{N}(0, 1)$ be independent. We have the upper tail*

$$\mathbb{P}\left(\max_{i \in [d]} |Z_i| \geq \sqrt{2\log d} + s\right) \leq 2e^{-s^2/2}, \quad \forall s \geq 0.$$

*Proof.* For notational simplicity, put $Z^* = \max_{i \in [d]} Z_i$. By union bound and the Chernoff bound, we have for each $s, \theta > 0$,

$$\mathbb{P}(Z^* \geq s) = \mathbb{P}\left(\bigvee_{i=1}^d Z_i \geq s\right) \leq d\,\mathbb{P}(Z_1 \geq s) \leq d\frac{\mathbb{E}\,e^{\theta Z_1}}{e^{\theta s}} = de^{\theta^2/2 - \theta s}.$$

Choose $\theta = s$ to minimize the RHS, and we obtain $\mathbb{P}(Z^* \geq s) \leq e^{\log d - s^2/2}$. Replace $s$ with $\sqrt{2\log d + s^2}$ and this becomes

$$\mathbb{P}\left(Z^* \geq \sqrt{2\log d} + s\right) \leq \mathbb{P}\left(Z^* \geq \sqrt{2\log d + s^2}\right) \leq e^{-s^2/2}.$$

Use the fact $-\min_{i \in [d]} Z_i \stackrel{d}{=} \max_{i \in [d]} Z_i$ and we complete the proof.    $\square$

**Lemma F.3** (Lower tail for the maximum). *Let $Z_1, \ldots, Z_d \sim \mathcal{N}(0, 1)$ be independent. Let $c > 0$ be any universal constant. We have*

$$\mathbb{P}\left[\max_{i \in [d]} Z_i \geq (1 + c)\sqrt{2 \log d}\right] \geq \frac{1}{8\pi(1 + c)} \frac{1}{d^{(1+c)^2 - 1}\sqrt{\log d}}.$$

*Proof.* First, we prove a general result on the integral $I(x) = \int_x^\infty e^{-y^2/2}\mathrm{d}y$. Make the change of variable $y = x\tau$ to obtain $I(x) = x \int_1^\infty e^{-x^2\tau^2/2}\mathrm{d}\tau$. Since the integrand decays very fast as $\tau$ grows, we expand $\tau^2/2$ around as $\tau^2/2 = 1/2 + (\tau - 1) + (\tau - 1)^2/2$. This gives

$$I(x) = xe^{-x^2/2}\int_1^\infty e^{-x^2(\tau-1)}e^{-x^2(\tau-1)^2/2}\mathrm{d}\tau = xe^{-x^2/2}\int_0^\infty e^{-x^2\tau}e^{-x^2\tau^2/2}\mathrm{d}\tau$$

For the second factor, we have

$$\int_0^\infty e^{-x^2\tau}e^{-x^2\tau^2/2}\mathrm{d}\tau \leq \int_0^\infty e^{-x^2\tau}\mathrm{d}\tau = \frac{1}{x^2},$$

$$\int_0^\infty e^{-x^2\tau}e^{-x^2\tau^2/2}\mathrm{d}\tau \geq \int_0^\infty e^{-x^2\tau}\left(1 - \frac{x^2\tau^2}{2}\right)\mathrm{d}\tau = \frac{1}{x^2}\left(1 - \frac{1}{x^2}\right).$$

Combining these bounds together, we obtain

$$\frac{e^{-x^2/2}}{x}\left(1 - \frac{1}{x^2}\right) \leq I(x) \leq \frac{e^{-x^2/2}}{x}. \tag{14}$$

With this estimation, we are ready to prove this lemma. Let $c > 0$ be a constant. Note that by our previous tail bound, $\max_{i \in [d]} Z_i \geq (1 + c)\sqrt{2 \log d} =: \theta$ is a rare event. We have

$$\mathbb{P}\left[\max_{i \in [d]} Z_i \geq \theta\right] = 1 - \left(1 - \frac{I(\theta)}{\sqrt{2\pi}}\right)^d \geq \frac{d}{2}\frac{I(\theta)}{\sqrt{2\pi}}$$

$$\geq \frac{d}{4\sqrt{2\pi}}\frac{e^{-\theta^2/2}}{\theta} = \frac{1}{8\pi(1 + c)}\frac{1}{d^{(1+c)^2 - 1}\sqrt{\log d}}.$$

$\square$

**Lemma F.4** (Gap between the largest and the second largest). *Let $Z_1, \ldots, Z_d \sim \mathcal{N}(0, 1)$ be independent. Consider an arbitrary universal constant $c \geq 1/\sqrt{2}$. Define the good and bad events as*

$$G := \left\{\max_{i \in [d]} |Z_i| \geq (1 + 2c)\sqrt{2 \log d}\right\},$$

$$B := \left\{\exists i \neq j \in [d], \min\{|Z_i|, |Z_j|\} \geq (1 + c)\sqrt{2 \log d}\right\}.$$

*We have*

$$\frac{\mathbb{P}(B)}{\mathbb{P}(G)} \leq \frac{8\pi(1 + 2c)\sqrt{\log d}}{d^{1 - 2c^2}} \to 0 \quad \text{as } d \to \infty.$$

*Let $|Z|_{(1)}$ and $|Z|_{(2)}$ be the largest and second-largest among $|Z_1|, \ldots, |Z_d|$. We have*

$$\mathbb{P}\left[\frac{|Z|_{(1)}}{|Z|_{(2)}} \geq \frac{1 + 2c}{1 + c}\right] \geq \mathbb{P}[G \wedge \neg B] \geq (1 - o(1))\mathbb{P}(G) \geq \frac{1}{5\pi(1 + 2c)}\frac{1}{d^{4c+4c^2}\sqrt{\log d}}.$$

*Proof.* Let $0 < c_1 < c_2$ be two universal constants to be determined later. By Lemma F.3, we have

$$\mathbb{P}(G) := 2\mathbb{P}\left[\max_{i \in [d]} Z_i \geq (1 + c_2)\sqrt{2 \log d}\right] \geq \frac{1}{4\pi(1 + c_2)}\frac{1}{d^{(1+c_2)^2 - 1}\sqrt{\log d}}.$$

Meanwhile, we have

$$\mathbb{P}(B) := \mathbb{P}\left[\exists i \neq j \in [d], \min\{|Z_i|, |Z_j|\} \geq (1 + c_1)\sqrt{2 \log d}\right]$$

$$\leq 2\binom{d}{2}\left(\mathbb{P}\left[Z_1 \geq (1 + c_1)\sqrt{2 \log d}\right]\right)^2$$

$$\leq d^2 \exp\left(-2(1 + c_1)^2 \log d\right)$$

$$= d^{-2(1+c_1)^2 + 2}.$$

Combine these bounds together, we obtain

$$\frac{\mathbb{P}(B)}{\mathbb{P}(G)} \leq \frac{4\pi(1 + c_2)d^{(1+c_2)^2 - 1}\sqrt{\log d}}{d^{2(1+c_1)^2 - 2}} = \frac{4\pi(1 + c_2)\sqrt{\log d}}{d^{2(1+c_1)^2 - 1 - (1+c_2)^2}}.$$

Suppose that $c_1^2 = c^2 > 1/2$ and choose $c_2 = 2c_1$. Then, the above becomes

$$\frac{\mathbb{P}(B)}{\mathbb{P}(G)} \leq \frac{4\pi(1 + 2c)\sqrt{\log d}}{d^{1 - 2c^2}}.$$

$\square$

## F.2 STOCHASTIC INDUCTION

Our proof is essentially a large induction. When certain properties hold, we know how to analyze the dynamics and can show certain quantities are bounded with high probability. Meanwhile, certain properties hold as long as those quantities are still well-controlled. In the deterministic setting, this seemingly looped argument can be made formal by either mathematical induction (in discrete time) or the continuity argument (in continuous time). In this subsection, we show the same can also be done in the presence of randomness and derive a stochastic version of Gronwall's lemma and its generalizations.

We start with an example where Doob's submartingale inequality can be directly used. Let $(\Omega, \mathcal{F}, (\mathcal{F}_t)_t, \mathbb{P})$ be our filtered probability space and $(Z_t)_t$ be a martingale difference sequence. Suppose that $\mathbb{E}[Z_{t+1}^2 \mid \mathcal{F}_t]$ is uniformly bounded by $\sigma_Z^2$. Then, by Doob's submartingale inequality, for any $M > 0$ and $T > 0$, we have

$$\mathbb{P}\left[\sup_{t \leq T}\left|\sum_{s=1}^t Z_s\right| \geq M\right] \leq M^{-2}\mathbb{E}\left(\sum_{s=1}^T Z_s\right)^2 = \frac{T\sigma_Z^2}{M^2}.$$

In particular, this implies that when $M = \omega(\sigma_Z\sqrt{T})$, we have $\sup_{t \leq T}\left|\sum_{s=1}^t Z_s\right| \leq M$ with high probability.

Note that there is no need to any kind of "induction" in the above example. However, things become subtle if instead of assuming $\mathbb{E}[Z_{t+1}^2 \mid \mathcal{F}_t]$ is bounded by $\sigma_Z^2$, we assume it is bounded by $\sigma_Z^2$ as long as $\sup_{s \leq t}|\sum_{r=1}^s Z_r| \leq M$. Intuitively, since $M$ is chosen so that $\sup_{t \leq T}\left|\sum_{s=1}^t Z_s\right| \leq M$ holds with high probability, the bounds $\mathbb{E}[Z_{t+1}^2 \mid \mathcal{F}_t] \leq \sigma_Z^2$ should also hold with high probability and we can still use Doob's submartingale inequality as before. Now, we formalize this argument.

**Lemma F.5.** *Let $(Z_t)_t$ be a martingale difference sequence. Suppose that there exists $M, \sigma_Z > 0$ such that if $\sup_{s \leq t}|\sum_{r=1}^s Z_s| \leq M$, then we have $\mathbb{E}[Z_{t+1}^2 \mid \mathcal{F}_t] \leq \sigma_Z^2$. Then, we have*

$$\mathbb{P}\left[\sup_{t \leq T}\left|\sum_{s=1}^t Z_s\right| > M\right] \leq \frac{T\sigma_Z^2}{M^2}.$$

*Note that this bound is the same as the one we obtained with the assumption that $\mathbb{E}[Z_{t+1}^2 \mid \mathcal{F}_t] \leq \sigma_Z^2$ always holds.*

*Proof.* Consider the stopping time $\tau := \inf\{t \geq 0 : \left|\sum_{s=1}^t Z_s\right| > M\}$. By definition, we have $\sup_{s \leq t}|\sum_{r=1}^s Z_s| \leq M$ for all $t \leq \tau$. Then, we define $Y_{t+1} = Z_{t+1}\mathbb{1}\{t < \tau\}$. Note that $(Y_t)$ is

a martingale difference sequence with $\mathbb{E}[Y_{t+1}^2 \mid \mathcal{F}_t] \leq \sigma_Z^2$. As a result, by Doob's submartingale inequality, we have $\mathbb{P}\left[\sup_{t \leq T} \left| \sum_{s=1}^t Y_s \right| > M \right] \leq T\sigma_Z^2/M^2$. To relate it to $(Z_t)_t$, we compute

$$\mathbb{P}\left[\sup_{t \leq T} \left| \sum_{s=1}^t Z_s \right| > M \right] = \mathbb{P}\left[\sup_{t \leq T} \left| \sum_{s=1}^t Z_s \right| > M \wedge \tau \leq T \right] = \mathbb{P}\left[\left| \sum_{s=1}^\tau Z_s \right| > M \wedge \tau \leq T \right]$$

$$= \mathbb{P}\left[\left| \sum_{s=1}^\tau Y_s \right| > M \wedge \tau \leq T \right]$$

$$\leq \frac{T\sigma_Z^2}{M^2},$$

where the first and second identities comes from the definition of $\tau$ and the third from the fact $Z_t = Y_t$ for all $t \leq \tau$. $\qquad\square$

Now, we consider a more complicated case, where is process of interest is not a pure martingale. Suppose that the process $(X_t)_t$ satisfies

$$X_{t+1} = (1 + \alpha)X_t + \xi_{t+1} + Z_{t+1}, \quad X_0 = x_0 > 0,$$

where the signal growth rate $\alpha > 0$ and initialization $x_0 > 0$ are given and fixed, $(\xi_t)_t$ is an adapted process, and $(Z_t)_t$ is a martingale difference sequence. In most cases, $(\xi_t)_t$ will represent the higher-order error terms.

Our goal is control the difference between $X_t$ and its deterministic counterpart $x_t = (1 + \alpha)^t x_0$. To this end, we recursively expand the RHS to obtain

$$X_{t+1} = (1 + \alpha)^2 X_{t-1} + (1 + \alpha)\xi_t + \xi_{t+1} + (1 + \alpha)Z_t + Z_{t+1}$$

$$= (1 + \alpha)^{t+1} x_0 + \sum_{s=1}^t (1 + \alpha)^{t-s} \xi_{s+1} + \sum_{s=1}^t (1 + \alpha)^{t-s} Z_{s+1}.$$

Divide both sides with $(1 + \alpha)^{t+1}$ and replace $t + 1$ with $t$. Then, the above becomes

$$X_t (1 + \alpha)^{-t} = x_0 + \sum_{s=1}^t (1 + \alpha)^{-s} \xi_s + \sum_{s=1}^t (1 + \alpha)^{-s} Z_s.$$

Note that $((1 + \alpha)^{-t} Z_t)_t$ is still a martingale difference sequence. Ideally, $|\xi_t|$ should be small as it represents the higher-order error terms, and we have bounds on the conditional variance of $Z_t$ so that we can apply Doob's submartingale inequality to the last term. Unfortunately, in many cases, since $\xi_{t+1}$ and $Z_{t+1}$, particularly their maximum and (conditional) variance, can potentially depend on $(X_s)_{s \leq t}$, we may only be able to assume $|\xi_{t+1}| \leq (1 + \alpha)^t \Xi$ with probability at least $1 - \delta_{\mathbb{P},\xi}$ (for each $t$) and $\mathbb{E}[Z_{t+1}^2 \mid \mathcal{F}_t] \leq (1 + \alpha)^t \sigma_Z^2$ for some $\xi_{\mathbb{P},\xi}$, $\Xi$ and $\sigma_Z^2$ when, say, $X_t = (1 \pm 0.5)x_t$. Still, we can use the previous argument to estimate the probability that $X_t \notin (1 \pm 0.5)x_t$ for some $t \leq T$.

Let $\tau := \inf\{t \geq 0 : X_t \notin (1 \pm 0.5)x_t\}$ and then $\hat{\xi}_{t+1} = \xi_{t+1}\mathbb{1}\{t \leq \tau\}$, and $\hat{Z}_{t+1} = Z_{t+1}\mathbb{1}\{t \leq \tau\}$. Clear that $\tau$ is a stopping time, $\hat{\xi}$ is adapted, and $\hat{Z}$ is still a martingale difference sequence. Moreover, we have $|\hat{\xi}_t| \leq (1+\alpha)^t \Xi$ with probability at least $1 - \delta_{\mathbb{P},\xi}$ and $\mathbb{E}\left[\hat{Z}_{t+1}^2 \mid \mathcal{F}_t\right] \leq (1 + \alpha)^t \sigma_Z^2$ for all $t \geq 0$. As a result,

$$\left| \sum_{s=1}^t (1 + \alpha)^{-s} \hat{\xi}_s \right| \leq \Xi t \leq T\Xi \quad \text{with probability at least } 1 - T\delta_{\mathbb{P},\xi},$$

$$\mathbb{E}\left( \sum_{s=1}^t (1 + \alpha)^{-s} \hat{Z}_s \right)^2 = \sum_{s=1}^t (1 + \alpha)^{-2s} \mathbb{E}\,\mathbb{E}\left[\hat{Z}_s^2 \mid \mathcal{F}_{s-1}\right] \leq \sum_{s=1}^t (1 + \alpha)^{-s} \sigma_Z^2 \leq \frac{\sigma_Z^2}{\alpha}.$$

Then, by Doob's submartingale inequality, we have

$$\mathbb{P}\left[\sup_{t \leq T} \left| \sum_{s=1}^T (1 + \alpha)^{-s} \hat{Z}_s \right| \geq \frac{x_0}{4} \right] \leq \frac{16\sigma_Z^2}{\alpha x_0^2}.$$

Hence, for any $\delta_{\mathbb{P}} \in (0, 1)$, if we assume

$$\Xi \le \frac{x_0}{4T} \quad \text{and} \quad \sigma_Z^2 \le \frac{\delta_{\mathbb{P}} \alpha x_0^2}{16},$$

then with probability at least $1 - \delta_{\mathbb{P}} - T\delta_{\mathbb{P},\xi}$, we have

$$\left| \sum_{s=1}^{t} (1+\alpha)^{-s} \hat{\xi}_s + \sum_{s=1}^{t} (1+\alpha)^{-s} \hat{Z}_s \right| \le \frac{x_0}{2}, \quad \forall t \in [T].$$

Then, similar to the previous argument, we have

$$\begin{aligned}
\mathbb{P}\left[\exists t \in [T], \ X_t \notin (1 \pm 0.5)x_t\right] &= \mathbb{P}\left[\exists t \in [T], \ X_t \notin (1 \pm 0.5)x_t \wedge \tau \le T\right] \\
&= \mathbb{P}\left[X_\tau \notin (1 \pm 0.5)x_\tau \wedge \tau \le T\right] \\
&= \mathbb{P}\left[\left| \sum_{s=1}^{\tau} (1+\alpha)^{-s} \xi_s + \sum_{s=1}^{\tau} (1+\alpha)^{-s} Z_s \right| \ge \frac{x_0}{2} \wedge \tau \le T\right] \\
&= \mathbb{P}\left[\left| \sum_{s=1}^{T} (1+\alpha)^{-s} \hat{\xi}_s + \sum_{s=1}^{T} (1+\alpha)^{-s} \hat{Z}_s \right| \ge \frac{x_0}{2} \wedge \tau \le T\right] \\
&\le 1 - \delta_{\mathbb{P}} - T\delta_{\mathbb{P},\xi}.
\end{aligned}$$

Namely, we have proved the following discrete-time stochastic Gronwall's lemma.

**Lemma F.6** (Stochastic Gronwall's lemma). *Suppose that $(X_t)_t$ satisfies*

$$X_{t+1} = (1+\alpha)X_t + \xi_{t+1} + Z_{t+1}, \quad X_0 = x_0 > 0,$$

*where the signal growth rate $\alpha > 0$ and initialization $x_0 > 0$ are given and fixed, $(\xi_t)_t$ is an adapted process, and $(Z_t)_t$ is a martingale difference sequence. Define $x_t = (1+\alpha)^t x_0$.*

*Let $T > 0$ and $\delta_{\mathbb{P}} \in (0, 1)$ be given. Suppose that there exists some $\delta_{\mathbb{P},\xi} \in (0, 1)$ and $\Xi, \sigma_Z > 0$ such that for every $t \ge 0$, if $X_t = (1 \pm 0.5)x_t$, then we have $|\xi_{t+1}| \le (1+\alpha)^t \Xi$ with probability at least $1 - \delta_{\mathbb{P},\xi}$ and $\mathbb{E}[Z_{t+1}^2 \mid \mathcal{F}_t] \le (1+\alpha)^t \sigma_Z^2$. Then, if*

$$\Xi \le \frac{x_0}{4T} \quad \text{and} \quad \sigma_Z^2 \le \frac{\delta_{\mathbb{P}} \alpha x_0^2}{16},$$

*we have $X_t = (1 \pm 0.5)x_t$ for all $t \in [T]$ with probability at least $1 - \delta_{\mathbb{P}} - T\delta_{\mathbb{P},\xi}$.*

**Remark**. With only the dependence on $\alpha$ and $x_0$ kept, then conditions become $\Xi \le O(\alpha x_0)$ and $\sigma_Z \le O(\sqrt{\alpha} x_0)$. When $\alpha$ is small, the second condition is much weaker than the first one. ♣

**Remark**. The above argument can be easily generalized to cases where we have multiple induction hypotheses. For example, if we have another process $X'_{t+1} = (1+\alpha')X'_t + \xi'_{t+1} + Z'_{t+1}$ and we need both $X_t = (1 \pm 0.5)x_t$ and $X'_t = (1 \pm 0.5)x'_t$ for the bounds on $|\xi_{t+1}|, |\xi'_{t+1}|, \mathbb{E}[Z_{t+1}^2 \mid \mathcal{F}_t]$, $\mathbb{E}[(Z'_{t+1})^2 \mid \mathcal{F}_t]$ to hold. In this case, the final failure probability will be bounded by $T(\delta_{\mathbb{P},\xi} + \delta_{\mathbb{P},\xi'}) + 2\delta_{\mathbb{P}}$. ♣

If we are interested only in the upper bound, the above lemma can be used instead. In this lemma, the dependence on the initial value is more lenient.

**Lemma F.7.** *Suppose that $(X_t)_t$ satisfies*

$$X_{t+1} = (1+\alpha)X_t + \xi_{t+1} + Z_{t+1}, \quad X_0 = x_0 > 0,$$

*where the signal growth rate $\alpha > 0$ and initialization $x_0 > 0$ are given and fixed, $(\xi_t)_t$ is an adapted process, and $(Z_t)_t$ is a martingale difference sequence. Define $x_t^+ = (1+\alpha)^t x_0^+$, where $x_0^+$ is any value that is at least $x_0$.*

*Let $T > 0$ and $\delta_{\mathbb{P}} \in (0, 1)$ be given. Suppose that there exists some $\delta_{\mathbb{P},\xi} \in (0, 1)$ and $\Xi, \sigma_Z > 0$ such that for every $t \ge 0$, if $X_t = (1 \pm 0.5)x_t$, then we have $|\xi_{t+1}| \le (1+\alpha)^t \Xi$ with probability at least $1 - \delta_{\mathbb{P},\xi}$ and $\mathbb{E}[Z_{t+1}^2 \mid \mathcal{F}_t] \le (1+\alpha)^t \sigma_Z^2$. Then, if*

$$\Xi \le \frac{x_0^+}{4T} \quad \text{and} \quad \sigma_Z^2 \le \frac{\delta_{\mathbb{P}} \alpha (x_0^+)^2}{16},$$

*we have $X_t \le 2x_t^+$ for all $t \in [T]$ with probability at least $1 - \delta_{\mathbb{P}} - T\delta_{\mathbb{P},\xi}$.*

*Proof.* Similar to the previous proof, we still have

$$X_t(1+\alpha)^{-t} = x_0 + \sum_{s=1}^{t}(1+\alpha)^{-s}\xi_s + \sum_{s=1}^{t}(1+\alpha)^{-s}Z_s.$$

Instead of requiring the last two terms to be bounded by $x_0/2$, we can simply require them to be bounded by $x_0^+/2$ where $x_0^+$ is any value that is at least $x_0$. Then, to complete the proof, it suffices to repeat the previous argument. $\square$

The above lemmas will be used in Stage 1.1 to estimate the growth rate of the signals. The next lemma considers the case where $\alpha$ is 0 and will be used to show the gap between the largest and the second-largest coordinates can be preserved during Stage 1.1.

**Lemma F.8.** *Suppose that $(X_t)_t$ satisfies*

$$X_{t+1} = X_t + \xi_{t+1} + Z_{t+1}, \quad X_0 = x_0 > 0,$$

*where the signal growth rate $\alpha > 0$ and initialization $x_0 > 0$ are given and fixed, $(\xi_t)_t$ is an adapted process, and $(Z_t)_t$ is a martingale difference sequence.*

*Let $T > 0$ and $\delta_{\mathbb{P}} \in (0,1)$ be given. Suppose that there exists some $\delta_{\mathbb{P},\xi} \in (0,1)$ and $\Xi, \sigma_Z > 0$ such that for every $t \leq T$, if $|X_t - x_0| \leq T\Xi + \sqrt{T\sigma_Z^2/\delta_{\mathbb{P}}}$, then $|\xi_t| \leq \Xi$ with probability at least $1 - \delta_{\mathbb{P},\xi}$ and $\mathbb{E}[Z_{t+1}^2 \mid \mathcal{F}_t] \leq \sigma_Z^2$. Then, we have*

$$\sup_{t \leq T}|X_t - x_0| \leq T\Xi + \sqrt{\frac{T\sigma_Z^2}{\delta_{\mathbb{P}}}} \quad \text{with probability at least } 1 - T\delta_{\mathbb{P},\xi} - \delta_{\mathbb{P}}.$$

*Proof.* Recursively expand the RHS, and we obtain

$$X_t = x_0 + \sum_{s=1}^{t}\xi_s + \sum_{s=1}^{t}Z_s.$$

Consider the stopping time $\tau := \inf\left\{t \geq 0 \ : \ |X_t - x_0| > T\Xi + \sqrt{T\sigma_Z^2/\delta_{\mathbb{P}}}\right\}$. Define $\hat{\xi}_{t+1} = \mathbb{1}\{t < \tau\}\xi_{t+1}$ and $\hat{Z}_{t+1} = \mathbb{1}\{t < \tau\}Z_{t+1}$. Clear that

$$\sup_{t \leq T}\left|\sum_{s=1}^{t}\hat{\xi}_t\right| \leq T\Xi \quad \text{with probability at least } 1 - T\delta_{\mathbb{P},\xi}.$$

Meanwhile, by Doob's submartingale inequality, we have

$$\mathbb{P}\left[\sup_{t \leq T}\left|\sum_{s=1}^{t}\hat{Z}_s\right| \geq M\right] \leq \frac{T\sigma_Z^2}{M^2}.$$

In other words,

$$\sup_{t \leq T}\left|\sum_{s=1}^{t}\hat{\xi}_t + \sum_{s=1}^{t}\hat{Z}_t\right| \leq T\Xi + \sqrt{\frac{T\sigma_Z^2}{\delta_{\mathbb{P}}}} \quad \text{with probability at least } 1 - T\delta_{\mathbb{P},\xi} - \delta_{\mathbb{P}}.$$

Finally, we compute

$$\mathbb{P}\left[\sup_{t \leq T}|X_t - x_0| > T\Xi + \sqrt{\frac{T\sigma_Z^2}{\delta_{\mathbb{P}}}}\right] = \mathbb{P}\left[\sup_{t \leq T}|X_t - x_0| > T\Xi + \sqrt{\frac{T\sigma_Z^2}{\delta_{\mathbb{P}}}} \wedge T \geq \tau\right]$$

$$= \mathbb{P}\left[\left|\sum_{s=1}^{\tau}\xi_t + \sum_{s=1}^{\tau}Z_t\right| > T\Xi + \sqrt{\frac{T\sigma_Z^2}{\delta_{\mathbb{P}}}} \wedge T \geq \tau\right]$$

$$= \mathbb{P}\left[\left|\sum_{s=1}^{\tau}\hat{\xi}_t + \sum_{s=1}^{\tau}\hat{Z}_t\right| > T\Xi + \sqrt{\frac{T\sigma_Z^2}{\delta_{\mathbb{P}}}} \wedge T \geq \tau\right]$$

$$\leq 1 - T\delta_{\mathbb{P},\xi} - \delta_{\mathbb{P}}.$$

$\square$

The above proofs are all based on Doob's $L^2$-submartingale inequality. In other words, it only uses the information about the conditional variance, whence the dependence on $\delta_{\mathbb{P}}$ is $\sqrt{\delta_{\mathbb{P}}}$. It is possible to get a better dependence (of form $\operatorname{poly}\log(1/\delta_{\mathbb{P}})$) if we have a full tail bound similar to the ones in Lemma 2.2. This can be useful when we need to use the union bound. To this end, we need the following generalization of Freedman's inequality. The proof of it is deferred to the end of this section. In short, we truncate $Z_t$ at $M$, apply Freedman's inequality to the truncated sequence, and estimate the error introduced by the truncation. This and the next lemmas will not be used in the proof of our main results. We include them here to explain a possible strategy to improve the dependence on $\delta_{\mathbb{P}}$.

**Lemma F.9** (Freedman's inequality with unbounded variables). *Let $(Z_t)_t$ be martingale difference sequence with $\mathbb{E}[Z_t^2 \mid \mathcal{F}_{t-1}] \leq \sigma_Z^2$. Suppose that $Z_t$ satisfies the tail bound*

$$\mathbb{P}\left[|Z_t| \geq s \mid \mathcal{F}_{t-1}\right] \leq a \exp\left(-bs^c\right), \quad \forall s > 0, \tag{15}$$

*for some $a \geq 1$ and $b, c \in (0, 1]$. Then, there exists a constant $C_c$ that may depend on $c$ such that for any $\delta_{\mathbb{P}} \in (0, 1)$, we have, with probability at least $1 - \delta_{\mathbb{P}}$ that*

$$\left|\sum_{t=1}^{T} Z_t\right| \leq C_c \sqrt{T\left(\sigma_Z^2 + \frac{1}{b^{2/c}} + \frac{\log^{1/c}\left(\frac{aT}{b\sigma_Z\delta_{\mathbb{P}}}\right)}{b^{1/c}}\right) \log\left(\frac{1}{\delta_{\mathbb{P}}}\right)}.$$

**Remark**. Similar bounds hold for a wider range of parameters. We will only use lemma in the proof of Lemma C.9, where the martingale difference sequence is $(Z_t)_t$ satisfies the tail bound in Lemma 2.2 (without the $\log m$ introduced by the union bound). In other words, we have $a = C_L$, $b = P^{-1/(2L)}$, $c = 1/(2L)$, and $\sigma_Z^2 = C_L P^2$. In particular, note that both $1/b^{2/c}$ and $\sigma_Z^2$ have order $P^2$. ♣

With this lemma, we can obtain the following variant of Lemma F.8. Our goal here is to replace $\sqrt{T\sigma_Z^2/\delta_{\mathbb{P}}}$ with $\sqrt{T\sigma_Z^2/\operatorname{poly}\log\delta_{\mathbb{P}}}$. The proof is essentially the same as the proof of Lemma F.8, and is therefore deferred to the end of this section. An example of applying is lemma can be found in the proof of Lemma C.9.

**Lemma F.10.** *Suppose that $(X_t)_t$ satisfies[9]*

$$X_{t+1} = X_t + \xi_{t+1} + h_t Z_{t+1}, \quad X_0 = x_0 > 0,$$

*where the signal growth rate $\alpha > 0$ and initialization $x_0 > 0$ are given and fixed, $(\xi_t)_t$, $(h_t)_t$ are adapted processes, and $(Z_t)_t$ is a martingale difference sequence.*

*Let $T > 0$ and $\delta_{\mathbb{P}} \in (0, 1)$ be given. Suppose that there exists some $\delta_{\mathbb{P},\xi} \in (0, 1)$ and $\Xi, \sigma_Z, h^* > 0$ such that for every $t \leq T$, if*

$$|X_t - x_0| \leq T\Xi + C_c h^* \sqrt{T\left(\sigma_Z^2 + \frac{1}{b^{2/c}} + \frac{\log^{1/c}\left(\frac{aT}{b\sigma_Z\delta_{\mathbb{P}}}\right)}{b^{1/c}}\right) \log\left(\frac{T}{\delta_{\mathbb{P}}}\right)}, \tag{16}$$

*then $|\xi_t| \leq \Xi$ with probability at least $1 - \delta_{\mathbb{P},\xi}$, $|h_t| \leq h^*$, $\mathbb{E}[Z_{t+1}^2 \mid \mathcal{F}_t] \leq \sigma_Z^2$, and $Z_{t+1}^2$ satisfies the tail bound (15). Then, with probability at least $1 - T\delta_{\mathbb{P},\xi} - \delta_{\mathbb{P}}$, (16) holds for all $t \in [T]$.*

Now, we consider the case where the signal grows at a polynomial instead of linear rate. This lemma will be used in Stage 1.2, where the $2L$-th order terms dominate.

**Lemma F.11.** *Suppose that $(X_t)_t$ satisfies*

$$X_{t+1} = X_t + \alpha X_t^p + \xi_{t+1} + Z_{t+1}, \quad X_0 = x_0 > 0, \tag{17}$$

*where $p > 1$, the signal growth rate $\alpha > 0$ and initialization $x_0 > 0$ are given and fixed, $(\xi_t)_t$ is an adapted process, and $(Z_t)_t$ is a martingale difference sequence. Let $\hat{x}_t$ be the solution to the deterministic relationship*

$$\hat{x}_{t+1} = \hat{x}_t + \alpha \hat{x}_t^p, \quad \hat{x}_0 = x_0/2.$$

---

[9]Since we require $b \leq 1$ in (15), we need to "normalize" $Z_{t+1}$ here and use $h_t$ to keep its size.

*Fix $T > 0, \delta_{\mathbb{P}} \in (0,1)$. Suppose that there exist $\Xi, \sigma_Z > 0$ and $\delta_{\mathbb{P},\xi} \in (0,1)$ such that when $X_t \geq \hat{x}_t$, we have $|\xi_t| \leq \Xi$ with probability at least $1 - \delta_{\mathbb{P},\xi}$ and $\mathbb{E}[Z_{t+1} \mid \mathcal{F}_t] \leq \sigma_Z^2$. Then, if*

$$\Xi \leq \frac{x_0}{4T} \quad and \quad \sigma_Z^2 \leq \frac{x_0^2 \delta_{\mathbb{P}}}{16T},$$

*we have $X_t \geq \hat{x}_t$ for all $t \leq T$.*

*Proof.* Similar to our previous argument, we can assume w.l.o.g. that the bounds on $|x_t|$ and the conditional variance of $Z_{t+1}$ always hold.

Note that we can rewrite (17) as $X_{t+1} = X_t(1 + \alpha X_t^{p-1}) + \xi_t + Z_t$ and view it as the linear recurrence relationship in Lemma F.6 with a non-constant growth rate. This suggests defining the counterpart of $(1 + \alpha)^t$ as

$$P_{s,t} := \begin{cases} \prod_{r=s}^{t-1}(1 + \alpha X_r^{p-1}), & t > s, \\ 1, & t = s. \end{cases}$$

Then, we can inductively write (17) as

$$X_1 = X_0 \left(1 + \alpha X_0^{p-1}\right) + \xi_0 + Z_0,$$

$$X_2 = \left(X_0 \left(1 + \alpha X_0^{p-1}\right) + \xi_0 + Z_0\right) \left(1 + \alpha X_1^{p-1}\right) + \xi_1 + Z_1$$

$$= X_0 \left(1 + \alpha X_0^{p-1}\right) \left(1 + \alpha X_1^{p-1}\right) + \left(1 + \alpha X_1^{p-1}\right)(\xi_0 + Z_0) + \xi_1 + Z_1$$

$$= X_0 P_{0,2} + P_{1,2}(\xi_0 + Z_0) + \xi_1 + Z_1,$$

$$X_3 = X_2 \left(1 + \alpha X_2^{p-1}\right) + \xi_2 + Z_2$$

$$= (X_0 P_{0,2} + P_{1,2}(\xi_0 + Z_0) + \xi_1 + Z_1) \left(1 + \alpha X_2^{p-1}\right) + \xi_2 + Z_2$$

$$= X_0 P_{0,3} + P_{1,3}(\xi_0 + Z_0) + P_{2,3}(\xi_1 + Z_1) + \xi_2 + Z_2.$$

Continue the above expansion, and eventually we obtain

$$X_t = X_0 P_{0,t} + \sum_{s=1}^{t} P_{s,t}(\xi_{s-1} + Z_{s-1}).$$

By our induction hypothesis, we have $P_{0,s} \geq 1$. Hence, we can divide both sides with $P_{0,t}$ and then the above becomes

$$P_{0,t}^{-1} X_t = X_0 + \sum_{s=1}^{t} P_{0,t}^{-1} P_{s,t}(\xi_{s-1} + Z_{s-1}) = X_0 + \sum_{s=1}^{t} P_{0,s}^{-1} \xi_{s-1} + \sum_{s=1}^{t} P_{0,s}^{-1} Z_{s-1}.$$

For the second term, we have

$$\left| \sum_{s=1}^{t} P_{0,s} \xi_{s-1} \right| \leq \sum_{s=1}^{t} P_{0,s} |\xi_{s-1}| \leq T\Xi,$$

for all $t \leq T$ with probability at least $1 - T\delta_{\mathbb{P},\xi}$. By our assumption on $\Xi$, this is bounded by $x_0/4$. For the last term, by Doob's submartingale inequality, for any $M > 0$, we have

$$\mathbb{P}\left[\sup_{r \leq t} \left| \sum_{s=1}^{t} P_{0,s}^{-1} Z_{s-1} \right| \geq M\right] \leq M^{-2} \sum_{s=1}^{t} \mathbb{E}\left[P_{0,s}^{-2} Z_{s-1}^2\right] \leq \frac{\sigma_Z^2 T}{M^2}.$$

Choose $M = x_0/4$ and the RHS becomes $16\sigma_Z^2 T/x_0^2$, which is bounded by $\delta_{\mathbb{P}}$ by our assumption on $\sigma_Z$. Thus, with probability at least $1 - T\delta_{\mathbb{P},\xi} - \delta_{\mathbb{P}}$, we have $X_t \geq P_{0,t}(x_0/2)$ for all $t$. In particular, this implies $X_t \geq \hat{x}_t$ with at least the same probability. $\square$

The above coupling lemma, when combined with the following estimation on the growth rate of the deterministic process $\hat{x}_t$, gives an upper bound on the time needed for $X_t$ to grow from a small value to $\Theta(1)$.

**Lemma F.12.** *Suppose that $(x_t)_t$ satisfies $x_{t+1} = x_t + \alpha x_t^p$ for some $x_0 \in (0, 1)$ and $p > 2$ and $\alpha \ll 1/p$. Then, we have $x_t$ must reach $0.9$ within $O(1/(x_0^{p-1}\alpha))$ iterations.*

*Proof.* Consider the continuous-time process $\dot{y}_\tau = (1 - \delta)y_\tau^p$ where $y_0 = x_0$ and $\delta > 0$ is a parameter to be determined later. For $y$, we have the closed-form formula

$$y_\tau = \left( \frac{1}{x_0^{p-1}} - (p-1)(1-\delta)\tau \right)^{-1/(p-1)}.$$

Now, we show by induction that $x_t \geq y_{t\alpha}$. Clear that this holds when $t = 0$. In addition, we have

$$x_{t+1} - y_{(y+1)\alpha} = x_t - y_t + \int_0^\alpha \left( x_t^p - (1-\delta)y_{t\alpha+\beta}^p \right) d\beta.$$

Note that since $x_t \geq y_{t\alpha}$ and $y_{t\alpha+\beta} \leq y_{(t+1)\alpha}$, it suffices to ensure $y_{t\alpha} \geq (1-\delta)y_{(t+1)\alpha}$. By our closed-form formula for $y_\tau$, we have

$$y_{t\alpha} \geq (1-\delta)y_{(t+1)\alpha}$$

$$\Leftrightarrow \quad \frac{1}{x_0^{p-1}} - (p-1)(1-\delta)t\alpha \leq (1-\delta)^{1-p}\left( \frac{1}{x_0^{p-1}} - (p-1)(1-\delta)(t+1)\alpha \right)$$

$$\Leftrightarrow \quad (1-\delta)^{p-1} \leq 1 - \frac{(p-1)(1-\delta)\alpha}{\frac{1}{x_0^{p-1}} - (p-1)(1-\delta)t\alpha}.$$

We are interested in the regime where $\frac{1}{x_0^{p-1}} - (p-1)(1-\delta)t\alpha \geq c_p$ for some small constant $c_p > 0$ that may depend on $p$. In this regime, we have

$$\frac{(p-1)(1-\delta)\alpha}{\frac{1}{x_0^{p-1}} - (p-1)(1-\delta)t\alpha} \leq c_p p\alpha.$$

As a result, if $c_p p\alpha \leq 0.1$, then in order for $y_{t\alpha} \geq (1-\delta)y_{(t+1)\alpha}$ in this regime, it suffices to choose

$$(1-\delta)^{p-1} \leq 1 - c_p p\alpha \quad \Leftarrow \quad (1-\delta)^{p-1} \leq e^{-2c_p p\alpha}$$
$$\Leftarrow \quad 1 - \delta \leq e^{-4c_p\alpha} \quad \Leftarrow \quad \delta \geq 8c_p\alpha.$$

Let 1 be our target value for $x_t$. To reach $C_*$, we need $\frac{1}{x_0^{p-1}} - (p-1)t\alpha \leq 1$. Choose $c_p = 1$. Then the above implies that $x_t \geq y_{t\alpha}$ with $\dot{y}_\tau = (1 - 8\alpha)y_\tau^p$ when $x_t \leq 1$. Combine this with the closed formula for $y_\tau$, and we conclude that $x_\tau$ must reach $1/2$ within $O(1/(x_0^{p-1}\alpha))$ iterations. □

### F.3 DEFERRED PROOFS OF THIS SECTION

*Proof of Lemma F.9.* In this proof, $C_c > 0$ will be a constant that can depend on $c$ and may change across lines. Let $M > 0$ be a parameter to be determined later. Write

$$Z_t = Z_t \mathbb{1}\{|Z_t| \leq M\} - \mathbb{E}\left[ Z_t \mathbb{1}\{|Z_t| \leq M\} \mid \mathcal{F}_{t-1} \right]$$
$$+ \mathbb{E}\left[ Z_t \mathbb{1}\{|Z_t| \leq M\} \mid \mathcal{F}_{t-1} \right] + Z_t \mathbb{1}\{|Z_t| > M\}.$$

Let $\hat{Z}_t$ denote the two terms in RHS of the first line. Note that $(\hat{Z}_t)_t$ is a martingale difference sequence with conditional variance bounded by $\sigma_Z^2$. Moreover, every $\hat{Z}_t$ is bounded by $2M$. Thus, by Freedman's inequality, we have

$$\mathbb{P}\left[ \left| \sum_{t=1}^T \hat{Z}_t \right| \geq s \right] \leq 2\exp\left( -\frac{s^2}{2T(\sigma_Z^2 + M)} \right), \quad \forall s \geq 0. \tag{18}$$

Now, we estimate the expectation $\mathbb{E}\left[ Z_t \mathbb{1}\{|Z_t| \leq M\} \mid \mathcal{F}_{t-1} \right]$. Since $\mathbb{E}\left[ Z_t \mid \mathcal{F}_{t-1} \right] = 0$, it is equal to $\mathbb{E}\left[ Z_t \mathbb{1}\{|Z_t| > M\} \mid \mathcal{F}_{t-1} \right]$, for which we have

$$|\mathbb{E}\left[ Z_t \mathbb{1}\{|Z_t| > M\} \mid \mathcal{F}_{t-1} \right]| \leq \mathbb{E}\left[ |Z_t| \mathbb{1}\{|Z_t| > M\} \mid \mathcal{F}_{t-1} \right]$$
$$= \int_M^\infty \mathbb{P}\left[ |Z_t| \geq s \right] ds \leq a \int_M^\infty \exp\left( -bs^c \right) ds.$$

Apply the change-of-variables $y = s/M$ and then $z = y^c$. Then, the above becomes

$$|\mathbb{E}\left[Z_t \mathbb{1}\{|Z_t| > M\} \mid \mathcal{F}_{t-1}\right]| \leq \frac{aM}{c} \int_1^\infty \exp\left(-bM^c z\right) z^{1/c-1} \, \mathrm{d}z$$

$$\leq \frac{aM}{c} \int_1^\infty \exp\left(-bM^c z + \left(\frac{1}{c} - 1\right) \log z\right) \, \mathrm{d}z.$$

Note that $\log z \leq \sqrt{z} \leq z$ for all $z \geq 1$. Hence, as long as $M^c \geq 2(1/c - 1)/b$, we will have

$$|\mathbb{E}\left[Z_t \mathbb{1}\{|Z_t| > M\} \mid \mathcal{F}_{t-1}\right]| \leq \frac{aM}{c} \int_1^\infty \exp\left(-bM^c z/2\right) \, \mathrm{d}z$$

$$\leq \frac{2a}{bc} \exp\left((1-c)\log M - bM^c/2\right).$$

Note that there exists some constant $C_c > 0$ that depends on $c$ such that $\log M \leq M^{c/2}$ for all $M^c \geq C_c$. Suppose that $M$ is at least $C_c$. Then, as long as $M^{c/2} \geq 4(1-c)/b$, we will have

$$|\mathbb{E}\left[Z_t \mathbb{1}\{|Z_t| > M\} \mid \mathcal{F}_{t-1}\right]| \leq \frac{2a}{bc} \exp\left(-bM^c/4\right).$$

In other words, for any $\varepsilon_0 > 0$, we have $|\mathbb{E}\left[Z_t \mathbb{1}\{|Z_t| > M\} \mid \mathcal{F}_{t-1}\right]| \leq \varepsilon_0/T$ if

$$M^c \geq C_c \vee \frac{2(1/c-1)}{b} \vee \frac{16(1-c)^2}{b^2} \vee \frac{4}{b} \log\left(\frac{2aT}{\varepsilon_0 bc}\right)$$

$$= C_c \left(\frac{1}{b^2} \vee \frac{1}{b} \log\left(\frac{aT}{\varepsilon_0 b}\right)\right).$$

Meanwhile, by union bound and our tail bound on $Z_t$, we have

$$\mathbb{P}\left[\exists t \in [T], Z_t \mathbb{1}\{|Z_t| > M\} \neq 0\right] \leq \sum_{t=1}^T \mathbb{P}\left[|Z_t| > M\right] \leq Ta \exp\left(-bM^c\right).$$

Combine the above bounds with (18), and we obtain

$$\mathbb{P}\left[\left|\sum_{t=1}^T Z_t\right| \geq \varepsilon_0 + s\right] \leq \mathbb{P}\left[\left|\sum_{t=1}^T Z_t\right| \geq s\right] + \mathbb{P}\left[\exists t \in [T], Z_t \mathbb{1}\{|Z_t| > M\} \neq 0\right]$$

$$\leq 2 \exp\left(-\frac{s^2}{2T(\sigma_Z^2 + M)}\right) + Ta \exp\left(-bM^c\right),$$

where $M > 0$ satisfies

$$M^c \geq C_c \left(\frac{1}{b^2} \vee \frac{1}{b} \log\left(\frac{aT}{\varepsilon_0 b}\right)\right).$$

Let $\delta_{\mathbb{P}} \in (0, 1)$ be our target failure probability. We have

$$Ta \exp\left(-bM^c\right) \leq \frac{\delta_{\mathbb{P}}}{2} \quad \Leftarrow \quad M^c \geq \frac{1}{b} \log\left(\frac{2Ta}{\delta_{\mathbb{P}}}\right),$$

$$2 \exp\left(-\frac{s^2}{2T(\sigma_Z^2 + M)}\right) \leq \frac{\delta_{\mathbb{P}}}{2} \quad \Leftarrow \quad s^2 \geq 2T(\sigma_Z^2 + M) \log\left(\frac{4}{\delta_{\mathbb{P}}}\right).$$

Thus, for any $\delta_{\mathbb{P}} \in (0, 1)$, we have with probability at least $1 - \delta_{\mathbb{P}}$, we have

$$\left|\sum_{t=1}^T Z_t\right| \leq \varepsilon_0 + \sqrt{2T(\sigma_Z^2 + M) \log\left(\frac{4}{\delta_{\mathbb{P}}}\right)} \quad \text{where} \quad M^c \geq C_c \left(\frac{1}{b^2} \vee \frac{1}{b} \log\left(\frac{aT}{\varepsilon_0 b \delta_{\mathbb{P}}}\right)\right).$$

To remove the parameter $\varepsilon_0$, we choose $\varepsilon_0 = \sqrt{2T\sigma_Z^2 \log\left(\frac{4}{\delta_{\mathbb{P}}}\right)}$. Then, the above becomes, with probability at least $1 - \delta_{\mathbb{P}}$, we have

$$\left|\sum_{t=1}^T Z_t\right| \leq 2\sqrt{2T(\sigma_Z^2 + M) \log\left(\frac{4}{\delta_{\mathbb{P}}}\right)} \quad \text{where} \quad M^c \geq C_c \left(\frac{1}{b^2} \vee \frac{1}{b} \log\left(\frac{aT}{b\sigma_Z \delta_{\mathbb{P}}}\right)\right).$$

$\square$

*Proof of Lemma F.10.* As in the proof of Lemma F.8, we write $X_t = x_0 + \sum_{s=1}^t \xi_s + \sum_{s=1}^t h_{s-1} Z_s$, define

$$\tau := \inf \left\{ t \geq 0 \, : \, |X_t - x_0| > T\Xi + C_c \sqrt{T \left( \sigma_Z^2 + \frac{1}{b^{2/c}} + \frac{\log^{1/c} \left( \frac{aT}{b\sigma_Z \delta_{\mathbb{P}}} \right)}{b^{1/c}} \right) \log \left( \frac{T}{\delta_{\mathbb{P}}} \right)} \right\},$$

and $\hat{\xi}_{t+1} = \xi_{t+1} \mathbb{1}\{t < \tau\}$, $\hat{Z}_{t+1} = \mathbb{1}\{t < \tau\} Z_{t+1}$. By construction, we have

$$\sup_{t \leq T} \left| \sum_{s=1}^t \hat{\xi}_t \right| \leq T\Xi \quad \text{with probability at least } 1 - T\delta_{\mathbb{P},\xi}.$$

For the martingale difference term, first note that $h_t \hat{Z}_{t+1}/h_*$ satisfies (15). Hence, by Lemma F.9, with probability at least $1 - \delta_{\mathbb{P}}$, we have

$$\left| \sum_{s=1}^t h_t \hat{Z}_t \right| \leq C_c h_* \sqrt{T \left( \sigma_Z^2 + \frac{1}{b^{2/c}} + \frac{\log^{1/c} \left( \frac{aT}{b\sigma_Z \delta_{\mathbb{P}}} \right)}{b^{1/c}} \right) \log \left( \frac{1}{\delta_{\mathbb{P}}} \right)}.$$

Replace $\delta_{\mathbb{P}}$ with $\delta_{\mathbb{P}}/T$, apply the union bound, and we obtain

$$\sup_{t \leq T} \left| \sum_{s=1}^t h_t \hat{Z}_t \right| \leq C_c h_* \sqrt{T \left( \sigma_Z^2 + \frac{1}{b^{2/c}} + \frac{\log^{1/c} \left( \frac{aT}{b\sigma_Z \delta_{\mathbb{P}}} \right)}{b^{1/c}} \right) \log \left( \frac{T}{\delta_{\mathbb{P}}} \right)},$$

with probability at least $1 - \delta_{\mathbb{P}}$. In other words, we have

$$\sup_{t \in [T]} \left| \sum_{s=1}^t \hat{\xi}_t + \sum_{s=1}^t h_t \hat{Z}_t \right| \leq T\Xi + C_c h^* \sqrt{T \left( \sigma_Z^2 + \frac{1}{b^{2/c}} + \frac{\log^{1/c} \left( \frac{aT}{b\sigma_Z \delta_{\mathbb{P}}} \right)}{b^{1/c}} \right) \log \left( \frac{T}{\delta_{\mathbb{P}}} \right)},$$

with probability at least $1 - T\delta_{\mathbb{P},\xi} - \delta_{\mathbb{P}}$. To complete the proof, it suffices to repeat the final part of the proof of Lemma F.8. $\qquad\square$

## G  SIMULATION

We include simulation results for Stage 1 in this section. The goal here is to provide empirical evidence that (i) if we have both the second- and $2L$-th order terms, then the sample complexity of online SGD scales linearly with $d$, (ii) the same also holds for the absolute function (which is a special case of the setting in Li et al. (2020)) and (iii) without the higher-order terms, online SGD cannot recovery the exact directions.

The setting is the same as the one we have described in Section 2. We choose the hyperparameters roughly according to Theorem 2.1. To reduce the needed computational resources, we choose $m = \Theta(P^2)$ instead of $\tilde{\Omega}(P^8)$. Note that by the Coupon Collector problem, we need $m = \Omega(P \log P)$ to ensure that for each $p \in [P]$, there exists at least one neuron $\boldsymbol{v}$ with $v_p^2 \geq \max_{q \leq P} v_q^2$. Since we are mostly interested in the dependence on $d$, for the learning rate, we choose $\eta = c/d$, where $c$ is a tunable constant that is independent of $d$ but can depend on everything else. $T$ is chosen according to Theorem 2.1 and we early-stop the training when for all $p \in [P]$, there exists a neuron with $v_p^2 \geq 0.95$ (in the moving average sense).

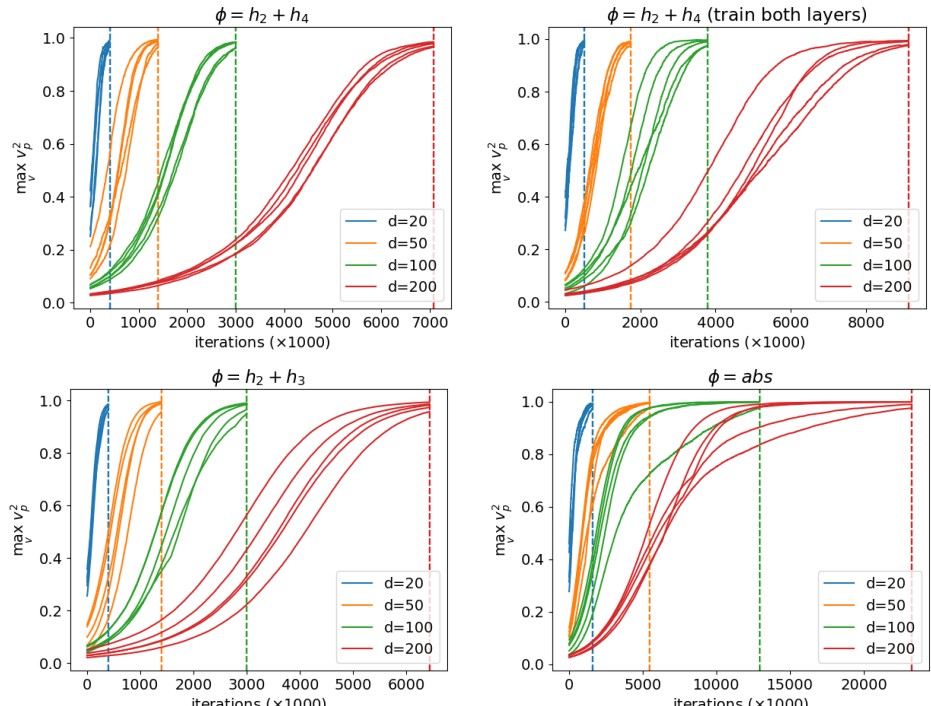

Figure 1: Recovery of directions. The above plots show the evolution of the correlation with each of the ground-truth directions. We fix the relevant dimension $P = 5$ and vary the ambient dimension $d$. Different colors represent different $d$. For each color, one curve represents $\max_{\boldsymbol{v}} v_p^2$ for one $p \in [P]$. In the first row, the link function is $\phi = h_2 + h_4$, a function that is covered by our theoretical results. In the left plot, we use the algorithm (3), while in the right plot, we train both layers simultaneously. We claimed that our theoretical results can be extended to other link functions with reasonably regular Hermite coefficients. The plots in the second row, where the link functions are $h_2 + h_4$ and the absolute value function, respectively, provides an empirical evidence for this. We can see that in all cases, online SGD successfully recover all ground-truth directions, and the number of steps/samples it needs scales approximately linearly with $d$.

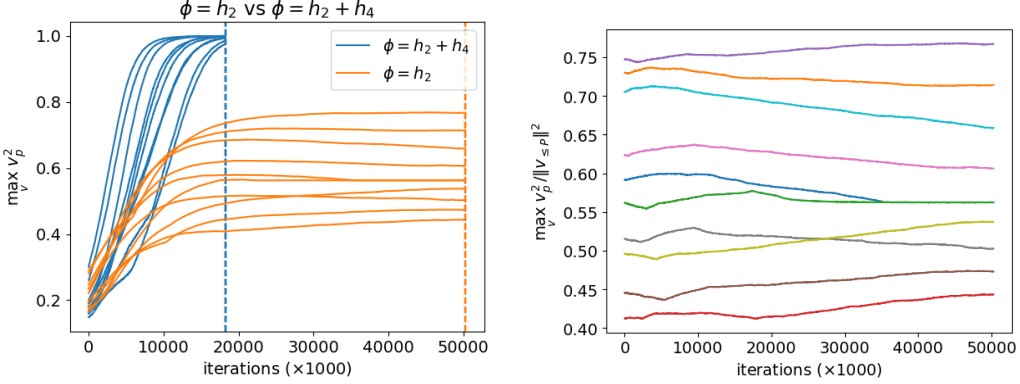

Figure 2: Necessity of the higher order terms. In these two figures, we choose $P = 10$ and $d = 100$. The left plot shows the maximum correlation each of the ground-truth directions (also see Figure 1). We can see that in the isotropic case, whether online SGD can recover the ground-truth directions is determined by the presence/absence of the higher-order terms. The right plot shows the change of $\max_{\boldsymbol{v}} v_p^2 / \|\boldsymbol{v}_{\leq P}\|^2$ for each $p \in [P]$ in Stage 1 when the link function is $h_2$. One can observe that they are almost unchanged throughout training. This, together with the left plot, shows that the increase of the correlation is caused by learning the subspace instead of the actual directions.

