# OpenReview forum: "Learning Orthogonal Multi-Index Models: A Fine-Grained Information Exponent Analysis"
_ICLR.cc/2025/Conference — Submitted to ICLR 2025_

### Official Review · Reviewer_Jcgf · 2024-10-31

**Soundness:** 4
**Presentation:** 3
**Contribution:** 3
**Rating:** 5
**Confidence:** 3

**Summary:**

This work investigates the sample complexity of learning Gaussian multi-index functions of the type $f_{\star}(x) = \sum\limits_{k=1}^{P}\left(h_{2}(\langle v_{k}^{\star},x\rangle)+h_{2L}(\langle v_{k}^{\star},x\rangle)\right)$ with two-layer neural networks trained under a particular two-step spherical one-pass SGD scheme.

The main result is to show that this target can be strongly learned with this procedure with $\tilde{O}({\rm poly}(P))$ hidden-units and $\tilde{O}(d{\rm poly}(P))$ samples / steps, in contrast with the purely quadratic case where the target can only be weakly learned. This result stems from an asymptotic description of the training dynamics in two phases, the first where only a subspace is recovered, followed by a second phase where the exact directions $\{v_{k}^{\star}\}_{k=1}^{P}$ are retrieved.

**Strengths:**

The description of the time scales for the dynamics is interesting. The informal discussion of how the SGD argument can be derived from the GF analysis and "user-friendly" discussion of the martingale argument from (Ben Arous 2021) is a nice addition to the literature.

**Weaknesses:**

- The setting studied is quite specific, and the focus on the information exponent without a broader discussion of the Gaussian multi-index literature can be misleading to the reader (see *Questions*).
- Overall, the manuscript is quite technical, which hinders clarity.

**Questions:**

As the authors discuss in the introduction, the information exponent (IE) was introduced by Ben Arous et al., 2021 specifically in the context of characterizing the sample complexity of one-pass SGD for well-specified single-index models (a.k.a. generalized linear models). While this concept was shown to be also relevant to the sample complexity in non-parametric settings (e.g. learning a single-index target with an infinite width 2-layer neural network, e.g. [Bietti et al. 2022] or [1,2]), there are a couple of works that already highlight that the information exponent is not a relevant concept beyond the rather specific scope for which it was introduced. For instance, Abbe et al., 2022 already points that is not sufficient to characterize strong learnability of multi-index functions, and other works show that the IE is not a robust concept to the change of algorithm even for weak learnability of single- and multi-index models [3, 4, 5, 6, Dandi et al., 2024, Damien et al., 2024, Lee et al., 2024]. In other words, it is already a consensus in the community that the IE is not a fundamental notion of computational complexity. For this reason, I find it is quite a stretch to motivate the result in this manuscript around this concept (e.g. as in the first bullet point in "*1.2 Our Contributions*").

In my reading, one contribution of this work is to provide an explicit example of a non-staircase multi-index target where strong learnability is achieved with one-pass SGD in $\tilde{O}(d)$ steps, showing that the staircase structure of Abbe et al. (2022, 2023) is not a necessary property. This hierarchical "weak-to-strong" learning structure is interesting per se.

Below are some specific questions / suggestions:

- In line 280, is $k\in[d]$ a typo?
- I would suggest the authors to avoid using the same index for quantities which are in different ranges, e.g. $k\in [P]$ or $k\in[m]$ for target and model neurons. This could help readability of the technical parts.
- In this work, you consider a noiseless multi-index target. Would the strong recovery result holds true in the presence of label noise?
- The second sentence in the discussion in L99-103 on the differences with respect to Ben Arous et al., (2024) is confusing, specially in light of Appendix A. If the analysis rely on a reduction to tensor decomposition, the settings do not look so different after all. Maybe be more precise on what you mean by "not comparable".

Some comments on the related literature:

- Reference [Lee et al., 2024] is concurrent to [4], please acknowledge both when referring to this result.
- The statistical and computational sample complexity results for single-index models in Damien et al. (2024) generalize the ones from [3]. Please acknowledge this in the discussion in L44-47.
- In L47-48: beyond SGD, computational weak-learnability results for multi-index functions were derived in [5] for polynomials and in [6] general link functions but linear sample complexity.
- Convergence rates for gradient flow on multi-index targets were derived in [7]. The setting studied in this manuscript (which assumes a small learning rate with respect to the initial conditions) should be close to that. Can you comment the relationship to this?

**References**

- [1] Raphaël Berthier, Andrea Montanari, Kangjie Zhou. [Learning time-scales in two-layers neural networks](https://link.springer.com/article/10.1007/s10208-024-09664-9). Foundations of Computational Mathematics. 2024 Aug 22:1-84.
- [2] Yatin Dandi, Florent Krzakala, Bruno Loureiro, Luca Pesce, Ludovic Stephan. [How Two-Layer Neural Networks Learn, One (Giant) Step at a Time](https://arxiv.org/abs/2305.18270). arXiv:2305.18270 [stat.ML]
- [3] Jean Barbier, Florent Krzakala, Nicolas Macris, Léo Miolane, Lenka Zdeborová. [Optimal Errors and Phase Transitions in High-Dimensional Generalized Linear Models](https://www.pnas.org/doi/10.1073/pnas.1802705116). PNAS 2019
- [4] Luca Arnaboldi, Yatin Dandi, Florent Krzakala, Luca Pesce, Ludovic Stephan. [Repetita Iuvant: Data Repetition Allows SGD to Learn High-Dimensional Multi-Index Functions](https://arxiv.org/abs/2405.15459). arXiv:2405.15459 [stat.ML]
- [5] Sitan Chen, Raghu Meka. [Learning Polynomials of Few Relevant Dimensions](http://proceedings.mlr.press/v125/chen20a/chen20a.pdf). ICML 2020
- [6] Emanuele Troiani, Yatin Dandi, Leonardo Defilippis, Lenka Zdeborová, Bruno Loureiro, Florent Krzakala. [Fundamental computational limits of weak learnability in high-dimensional multi-index models](https://arxiv.org/abs/2405.15480). arXiv:2405.15480 [cs.LG]
- [7] Alberto Bietti, Joan Bruna, Loucas Pillaud-Vivien. [On Learning Gaussian Multi-index Models with Gradient Flow](https://arxiv.org/pdf/2310.19793). arXiv:2310.19793 [stat.ML]

---

> ### Author Response · Authors · 2024-11-18
>
> Thank you for your constructive review and suggestions! We'd like to address your concerns as follows.
>
>
> ### Comparison with the generative exponent results
>
> We agree that for single-index models, the information exponent may not be the ideal way to characterize the
> sample complexity of this task outside the context of one-pass SGD. However, we'd like to emphasize that our results are
> more about the difference between single- and multi-index models, rather than showing information exponent is not a good
> way to analyze single-index models. In particular, the argument in our paper can also be generalized to generative
> exponents (see bellow). We choose to use one-pass SGD and information exponent only because the algorithm and the examples
> are simpler in this setting.
>
> First, note that even if we replace the information exponent with the generative exponent, we still can't distinguish
> the cases $\phi = h_2$ and $\phi = h_2 + h_4$. In both cases, the generative exponents are $2$. However, in the first
> case, due to the rotational invariance, any vector that lies in the relevant subspace is a minimizer of the loss,
> and it is impossible to recover the exact directions, while in the second case, we can recover the ground-truth
> directions using the higher-order terms. In other words, in certain multi-index cases, it is still insufficient to
> consider only the "lowest order", even when the "order" is described using the generative exponent.
>
> On the other hand, we believe that it should also be possible to generalize our positive results to generative exponents.
> Namely, if the link function is $\phi = h_2 + g$ where $g$ is a reasonable function with generative exponent
> larger than $2$, then we can recover the exact directions and the sample complexity is still $\tilde{O}(d)$.
> Similar to the information exponent setting, the intuition here is that we can recover the relevant subspace using
> $h_2$, and use that as a warm start to learn the exact directions with $g$. (Formally establishing this
> might be more challenging since the algorithm needed in Stage 2 would be more complicated.)
>
> In summary, our work highlights the difference between singe- and multi-index models, whereas the generative exponent
> and batch-reusing approaches focus on a different aspect. In particular, both information exponent and generative
> exponent can fail to capture the richer structure of multi-index models, and in certain cases, this can be fixed by
> considering the higher order terms.
>
> ### Responses to other questions/suggestions
>
> * Line 280: No, $k \in [d]$ is not a typo. Here, $v$ refers to an arbitrary first layer neuron and $v_k$ denotes its $k$-th
>   coordinate. We use the indicator $1\{ k \le P \}$ in the formula to differentiate between the cases $k \le P$ and $k > P$.
> * Indexing: We will use a different symbol to index the neurons in the revision for clarity.
> * Label noise: We believe that our argument still works in the presence of label noise. Similar to situations of
>   the information exponent paper and its follow-ups, the gradient noise is already fairly large, and we have chosen
>   $\eta = \tilde{O}(1/d)$ to make the dynamics stable. Hence, the label noise can usually be merged into the gradient
>   noise as long as it is not too crazy.
> * Difference with Ben Arous et al. (2024): The main difference between our results is that in their paper, still only
>   lowest order is considered, while we exploit both the lower and higher order terms.
>   In addition, they run SGD on the Stiefel manifold, which explicitly encodes the
>   non-degeneracy condition (while we need to prove it in Stage 1). The advantage of the Stiefel SGD is that they can
>   handle the case where the second-order terms are not isotropic while we need them to be isotropic.
> * Relationship to [7]: Similar to Ben Arous et al. (2024), they run GF on the Stiefel manifold (which prevents collapse)
>   and consider only the lowest order (though the lowest orders can be different for different directions in their
>   setting).
>
> ### Other related literatures
>
> Thank you very much for pointing out the related literature! We have included them in the revision.

---

> ### Comment · Reviewer_Jcgf · 2024-11-26
>
> I thank the authors for their rebuttal and for addressing my specific questions.
>
> On the general comment: I understand that the focus of this work is on the difference between single- and multi-index models (and not IE vs. GE).
>
> However, the point I tried to convey is that, in any case, the IE is not a robust notion of complexity. It fails as soon as you change model (single- to multi-index) or algorithm (one pass to full batch / multi-pass SGD), as shown by many works in the literature at this point in time.
>
> Note I am not saying your result is not interesting (I think it is). I am just pointing out that the way you pitch it in the abstract and introduction puts too much weight on the IE as fundamental, while it is already well understood it is a rather restrictive notion. Your result is yet another case example of this.

---

> > ### Author Response · Authors · 2024-11-26
> >
> > Thank you for clarifying your point! We'll change the way we introduce the setting from single- vs multi-index IE to lowest-order type analysis (including both IE and GE) vs analysis using both lower- and higher-order terms.
> >
> > Still, we wish to emphasize that, as we have discussed in the rebuttal, our counterexample also works for GE, so it is not only a restriction of IE, but also a restriction of other "lowest-order" measures including GE.

---

### Official Review · Reviewer_VZRT · 2024-11-03

**Soundness:** 3
**Presentation:** 4
**Contribution:** 4
**Rating:** 8
**Confidence:** 2

**Summary:**

The authors study the problem of learning multi-index models of the form $\sum_{k=1}^P {\rm He}_2 (z_k) + {\rm He}_L(z_k)$ where $L$ is even and greater than $2$ with a two layer neural network of width $m$. The training in the first layer is performed using online SGD and keeping the second layer fixed and small, while the second layer is learned at the end of training using linear regression.

Interestingly, the authors discover that in the training of the first layer there are two phases: first the subspaces are recovered, and then the individual directions are recovered.

**Strengths:**

The paper is exceptionally well written and easy to follow. The outline of the proof in sections 3 and 4 is incredibly clear, and it gives an insightful and intuitive picture of the different stages of learning.

The problem of studying this class of multi-index models is in my opinion incredibly important as they are instrumental in developing a theory of two layer networks, and studying the interaction between different terms in the Hermite expansion is an important piece that needs to be figured out.

**Weaknesses:**

I think the paper could greatly benefit from a set of numerical experiments implementing the algorithm in 2.3. The learning algorithm described in the paper is different from a standard gradient descent iteration in which one updates the weights in both the first and second layer together, and it's not clear to me if such a case will have qualitatively the same behaviour as what is presented. Additionally, I believe the assumption that the weights in the first layer evolve independently is not true in practice, and I would expect such interactions to be considered in future works. All of these points could have been addressed by a set of numerical experiments, which could have at least offered a justification for the assumptions taken in this work.

**Questions:**

1. Is it necessary to keep the weights in the first layer as a collection of orthonormal vectors? Would it be sufficient for them to be extracted from a Normal distribution?
2. Do you have some intuition of what happens if $a_0$ is not so small? As you state the dynamics of different weight vectors in the first layer are no more independent, but do you expect this to impact the qualitative behaviour you describe in your theorems? This could be explored numerically.
3. What would change if we were studying the model $\sum_{k=1}^P {\rm He}_2 (z_k) + {\rm He}_3(z_k)$?
4. What is the role of the regularisation $\lambda$ in learning the second layer? It's common not to regularise when training a two layer network. Could you comment on its role here?

---

> ### Author Response · Authors · 2024-11-18
>
> Thank you for your positive review! We'd like to answer your questions as follows.
>
> ### Numerical experiments
>
> We have included experimental results in the revision (Appendix G) for link functions $h_2 + h_4$, $h_2 + h_3$ and the
> absolute value function, with $d$ ranging from $20$ to $200$. For the link function $h_2 + h_4$, we also include
> experiments where $a$ and $V$ are trained simultaneously. In all cases, the sample/time complexity scales approximately
> linearly with $d$, which matches our theoretical analysis.
>
> ### Other questions
>
> * Orthogonal weights: We believe that strict orthogonality is not necessary, but having at least approximately
>   orthogonal weights is important for the argument to work. The approximate orthogonality can be formalized as
>   an incoherent condition, which is also used in Oko et al., 2024 and is also commonly assumed in the tensor decomposition
>   literature. In particular, if the weights are Gaussian vectors, then with high probability, they are approximately
>   orthogonal to each other (and therefore incoherent). In the incoherent case, the dynamics are similar to the orthogonal
>   case, but the proof will be more complicated due to some technical details.
> * It is true that when $a_0$ is not small, the dynamics of the weights will no longer be approximately independent.
>   However, in the orthogonal case, the influence of the interaction between neurons can usually be upper bounded.
>   Similar arguments have appeared in, for example, [1] and [2] (though the settings are different). In addition,
>   if we initialize $a_0$ to be small, but do not fix them during training, one can show that $a_k$ will grow faster
>   when $v_k$ is close to one of the ground-truth, as its negative gradient is roughly proportional to $f_*(x) \phi(v_k \cdot x)$.
>   In some sense, the behavior is similar to our layer-wise training strategy. See Appendix G for related experiments.
>   We choose to use layer-wise training only because the proof will be cleaner.
> * The only potential issue $h_2 + h_3$ can cause is the initial correlation $v \cdot v^*_k$ might be negative (which is not
>   an issue if the link function is even). This can be easily fixed by replacing each neuron $v$ with two neurons $v$ and
>   $-v$ at initialization. Experimental results supporting this are provided in the Appendix G of the revision.
> * The purpose of $l_2$ regularization is mostly technical, as we wish to use the generalization results for ridge
>   regression in Stage 2. We believe that with some effort, one can remove the $l_2$ regularization or replace
>   ridge regression with SGD since after all, the optimization task in Stage 2 (training the second layer with the first
>   layer fixed) is convex.
>
>
> [1] Yuanzhi Li, Tengyu Ma, and Hongyang R. Zhang. Learning Over-Parametrized Two-Layer Neural Networks beyond NTK. 2020
>
> [2] Gérard Ben Arous, Cédric Gerbelot, and Vanessa Piccolo. High-dimensional optimization for multi-spiked tensor. 2024.

---

> > ### Comment · Reviewer_VZRT · 2024-11-25
> >
> > Thanks for your detailed reply and your additional experiments.
> >
> > I happily keep my positive evaluation.

---

### Official Review · Reviewer_2HZx · 2024-11-03

**Soundness:** 3
**Presentation:** 3
**Contribution:** 2
**Rating:** 5
**Confidence:** 4

**Summary:**

This paper studies the problem of learning an additive Gaussian multi-index model with know link function, where the directions in the multi-index model are orthonormal and the non-linearity only includes the Hermite polynomials of degree 2 and $2L$ for some $L \geq 2$. It is shown that when using online SGD on the squared loss, while looking at the 2nd degree Hermite polynomial is not sufficient for strong recovery of all directions, one can leverage the 2nd degree Hermite polynomial to weakly recover the subspace with $\tilde{O}(d)$ samples. From this weak recovery, once can use the higher-order Hermite term to strongly recover the individual directions in the multi-index model, while maintaining the $\tilde{O}(d)$ sample complexity.

**Strengths:**

The majority of theoretical work on learning high-dimensional Gaussian single/multi-index models focus on the importance of the first non-zero term in the Hermite expansion of the link function. It is nice to show that sometimes the higher order terms can be useful in the analysis.

**Weaknesses:**

My main concern with the current submission is its scope. Specifically, the picture presented here will never be relevant to functions with information exponent larger than 2, as Oko et al., 2024 show that only looking at the first non-zero term in the Hermite expansion is sufficient. Even if the information exponent is exactly 2, as Oko et al., 2024 argue, after the subspace is recovered with online SGD, it is possible to subtract it with quadratic activation from the multi-index model, thus obtaining a new model that has information exponent larger than 2.

I think while the current analysis is interesting, this submission could be more strengthened if we also had a story for other types of link functions with information exponent 2, e.g. $h_2 + h_3$. Also, moving from exactly orthogonal to approximately orthogonal weights and characterizing the effect of this approximate orthogonality can be another valuable contribution.

**Questions:**

Some possible typos:

* What is $\epsilon_*$ and how does it relate to $\epsilon$ in Theorem 2.1?

* Is there a normalization step missing from Stage 1 of the algorithm in Section 2.3? Assuming there is a normalization step, then it seems the ODE in Line 278 is only considering the gradient step and not the projection. It is mentioned that the second term on the RHS comes from the projection, but it seems that the second term on the RHS is only coming from the higher degree Hermite term.

* I think $v_k = {v^*}_k^\top v$ has not been formally introduced before the equation in Line 278.

* $\Vert v_{\leq P}\Vert^2$ is missing from the numerator of the last in term in Line 304.

* It seems like the RHS of the equation in Line 352 must be $4L(1 - \Vert v_{\leq P}\Vert^2 + c_L(\Vert v_{\leq P}\Vert^2 - v_1^2))v_1^{2L} \geq 4Lc_L(\Vert v_{\leq P}\Vert^2 - v_1^2)v_1^{2L}$.

---

> ### Author Response · Authors · 2024-11-18
>
> Thank you for your review and suggestions! We'd like to address your concerns as follows.
>
>
> ### More comparison with Oko et al., 2024
>
> We wish to emphasize that our result is not a direct extension of Oko et al., 2024 (even when they learn the quadratic part
> of the link function first). Our sample/time complexity scales linear with $d$ (up to some logarithmic factors), while
> their results scale with $d^p$. Even if they use the strategy in [DLS22] to learn the quadratic part first and
> then subtract it from the target function (as they discussed in their paper) to make the information exponent larger than $2$,
> the final complexity will still be $d^p$ or at least $d^2$, depending on how you implement the algorithm.
>
> The most straightforward implementation of their argument is first learning the quadratic part $f_2$ of $f_*$ and then
> fitting the function $f_* - f_2$ using their argument. In this case, the complexity is still $d^p$ as this algorithm
> does not leverage the subspace information learned from the quadratic part.
>
> To utilize the subspace information, they would need to (re)initialize the neurons in the learned subspace (which itself is not
> very natural) in the second part of the algorithm. In addition, for the one-large-step argument in [DLS22] to work,
> $d^2$ samples are required. In contrast, our method only requires $d$ samples.
>
> [DLS22] Alex Damian, Jason D. Lee, Mahdi Soltanolkotabi. Neural Networks can Learn Representations with Gradient Descent. 2022
>
> ### On the scope of this paper
>
> It's true that our results focus on functions with information exponent $2$. However, one goal of our paper is
> to highlight that information exponent (or generative exponent, see the discussion with reviewer Jcgf) is too coarse to
> capture the richer structure of multi-index models.
>
> We also believe this type of analysis (starting with lower-order terms and then moving to higher-order terms) can
> potentially be useful in the general non-orthogonal case. Intuitively, when the order is high, the influence of each ground-truth neuron
> will be more localized, allowing the nearby neurons to converge to a single direction rather than a mixture of them.
> However, this localization will also make finding those local attraction regime more difficult. In the ideal case, we
> should be able to utilize the lower order terms first to find a not-so-local attraction regime and gradually move to
> higher order terms to localize the learner neurons. This mirrors our approach of using the 2nd order terms to find a
> right subspace and then the higher order terms to recover the exact directions.
>
> ### More general link functions
>
> As we have discussed in the paper, we believe that our results can be readily generalized to more general even link
> functions. In addition, for link functions like $h_2 + h_3$, the only issue here is that the initial $v$ can be negatively
> aligned with a ground-truth directions. This can be easily fixed by replacing each neuron $v$ with two neurons $v$ and $-v$
> at initialization. We include experimental results for link functions $h_2 + h_4$, $h_2 + h_3$ and the absolute value
> function in the revision (Appendix G). In all cases, the complexity scales roughly linearly with $d$.
>
> ### Typos and other clarifications
>
> Thank you for identifying the typos! We have corrected them in the revision.
>
> * $\epsilon_*$ and $\epsilon$: In Theorem 2.1, $\epsilon$ and $\epsilon_*$ are the same. This is a typo.
> * Normalization step: Indeed, we forget the normalization step in the Eq. (3). However, in the proof, we do have the
>   normalization step (cf. line 1006). For gradient flow, the normalization step is not necessary as long as we project
>   away the radial component of the gradient using $I - vv^T$ (as in Section 3). In GF, if $v_0$ is on the unit sphere,
>   $v_t$ will remain on the unit sphere.
> * Introduction of $v_k$: We assume w.l.o.g. that $v^*_k = e_k$ in line 274 and here $v_k$ refers to the $k$th coordinate
>   of $v$.
> * Missing $\| v_{\le P} \|^2$ in line 304: Thank you for pointing this out!
> * Line 352: This term is correct. It is indeed $c_L(1 - v_1^2)$ instead of $c_L ( \| v_{\le P} \|^2 - v_1^2 )$.
>   We have $v_1^2 \ge (1 + c) v_k^2$ for all $k \ge 2$ and therefore $v_k^2 \lesssim (1 - c) v_1^2$. Plug this into line
>   345 and then line 340, and we obtain the middle term of line 352. Then, replacing $\| v_{\le P} \|^2$ with $1$
>   leads to the final inequality.

---

> > ### Comment · Reviewer_2HZx · 2024-11-25
> >
> > I thank the authors for their detailed response. My impression is that if one trains the neurons with a quadratic activation, they will learn a predictor of the form $x^\top Q x$ where $Q = \sum_k v^*_k{v^*_k}^\top$. While this does not recover each individual direction, the $P$ dimensional subspace would be strongly recovered. By projecting onto this subspace, the dimension would be reduced and there will no longer be a dependence on $d$. In that case, there would be no point in looking at higher-order terms for high-dimensional recovery, it will only be relevant for fine-tuning the learning of the target function, which has not been the focus of the information exponent type of analysis. Can the authors perhaps provide an argument why such strong recovery of $Q$ with only quadratic activation functions would be impossible?

---

> > > ### Author Response · Authors · 2024-11-25
> > >
> > > Thank you for the additional comments! I'm not sure if your question is about why we want exact recovery of directions instead of just subspace recovery, or about the difference between our analysis and learning the subspace first and going from there, so I'll try to address both of them.
> > >
> > > First, it's true that we can recover the subspace using only the quadratic terms and we don't need to use the higher-order term if that's our only goal. Even so, our analysis still improves the previous $d^2$ bound in [DLS22] to $d$ in the isotropic case.
> > >
> > > ## Subspace recovery vs directional recovery
> > > While the earlier information exponent analysis cares mostly about the subspace recovery (or the recovery of a single direction in the single-index case), there is an increasing amount of recent results that concern the exact recovery of directions ([Oko et al, 2024, Ben Arous et al., 2024]). One important reason of directional recovery being preferable is that it essentially reduces the learning task to an 1-dimensional one, and the number of neurons needed for it will be $O(P)$ when the link function is known and $O(P \times \text{regularity of the link function})$ in the misspecified case. On the other hand, if we only know the subspace, the number of neurons needed can be exponential in $P$ even if the link function is known (unless it is quadratic). Though our focus is mostly on the dependence on $d$, $\mathrm{poly}(P)$ vs $\exp(P)$ is a big difference. Thus, being able to recover the exact directions is generally better than only being able to recover the subspace.
> > >
> > > ## Our analysis vs learning the subspace first and going from there
> > > I would say this really depends on what you do after you learn the subspace.
> > > * If you use general learning results for a generic function (i.e., ignoring the higher-order terms), then the complexity would be $e^P$.
> > > * If you use [Oko et al, 2024]'s result directly with the quadratic part subtracted (i.e., discarding the lower-order terms), then the complexity would be $d^{\mathrm{IE}}$ .
> > > * If you project everything onto the learned subspace and then use an argument similar to [Oko et al, 2024]'s or ours in that subspace (i.e., using both the lower- and higher-order terms), then the complexity would indeed be linear in $d$. However, I would say this supports our point that using both lower- and higher-order terms is beneficial in the multi-index case. Moreover, our analysis shows that this process can occur naturally in online SGD.

---

> > > > ### Comment · Reviewer_2HZx · 2024-12-02
> > > >
> > > > Thank you for providing further clarifications. Indeed, I was referring to the last bullet point above, i.e. if we first use quadratic activations with online SGD, we can recover the subspace (not the individual directions) with an almost linear in $d$ sample complexity, after which $d$ is removed from complexities by projecting onto this subspace.
> > > >
> > > > I think the point raised about the potentially exponential in $P$ gap between subspace and direction recovery is indeed interesting, but perhaps it requires a significant revision to rigorously develop this idea on concrete examples. Therefore, while I maintain my score, I encourage the authors to further work on this direction which I believe can turn into a very strong result.

---

### Meta-Review · Area_Chair_n2Lj · 2024-12-21

**Metareview:**

(a) **Summary of the scientific claims and findings**
The paper studies the learning dynamics of multi-index Gaussian functions through the lens of stochastic gradient descent (SGD). It demonstrates that a two-phase approach—using lower-order Hermite terms for subspace recovery followed by higher-order terms for exact directional recovery—can lead to better sample complexities than existing methods for multi-index models. This hierarchical approach emphasizes structural properties of multi-index models, extending beyond traditional information exponent analyses.

(b) **Strengths**
- The paper identifies key theoretical insights into the interplay of lower- and higher-order terms in Hermite expansions, advancing understanding of multi-index learning.
- It highlights a novel hierarchical learning dynamic that could inform future algorithm designs.
- The theoretical results are supported by clear mathematical arguments and connections to existing literature.

(c) **Weaknesses**
- **Lack of generality**: The method applies to specific link functions and relies on exact orthogonality assumptions, limiting broader applicability.
- **Experimental Validation**: The lack of numerical experiments weakens empirical support for the theoretical claims, leaving important practical questions unanswered.
- **Clarity in Comparisons**: The relationship between this work and prior results, such as those of Oko et al. (2024), needs more precise articulation.
- **Unaddressed Scenarios**: Important extensions, such as approximate orthogonality and noisy label scenarios, are not thoroughly explored.

(d) **Decision Rationale**
The paper offers valuable insights but falls short of ICLR's standard due to limited scope, incomplete empirical validation, and lack of generalizability. These issues, coupled with specific points raised by reviewers, warrant a rejection. However, the authors are strongly encouraged to address these limitations and resubmit.

**Additional Comments On Reviewer Discussion:**

The discussion among reviewers and authors touched on multiple key points:

1. **Scope and Generalizability (Reviewer 2HZx)**
   - **Concern**: The method may not generalize to functions with information exponents beyond 2 or approximate orthogonality settings.
   - **Author Response**: Clarified the scope as focusing on specific cases but acknowledged potential extensions.
   - **Final Weigh-In**: The limited scope reduces impact and relevance.

2. **Numerical Experiments (Reviewer VZRT)**
   - **Concern**: Lack of numerical validation for the theoretical claims and insights into practical scenarios.
   - **Author Response**: Promised inclusion of experiments in a revision but did not provide sufficient results in the current submission.
   - **Final Weigh-In**: Experimental validation is essential for a strong submission.

3. **Information Exponent Relevance (Reviewer Jcgf)**
   - **Concern**: Overemphasis on information exponent as a fundamental complexity measure, which is already shown to be restrictive.
   - **Author Response**: Acknowledged the critique and adjusted the narrative to focus on "lowest-order type analysis."
   - **Final Weigh-In**: The adjustment is appreciated but insufficient to address broader conceptual critiques.

4. **Technical and Notational Clarifications**
   - **Concern**: Reviewers flagged several typographical and conceptual ambiguities.
   - **Author Response**: Promised corrections in a revision.

### Final Recommendation
Reject but strongly encourage resubmission. Addressing generalizability, adding comprehensive experiments, and refining the narrative can elevate the paper’s contribution to ICLR standards.

---

### Decision · Program_Chairs · 2025-01-22

Reject